# Cost-aware Bayesian Optimization
# via the Pandora's Box Gittins Index

**Qian Xie**[1]      **Raul Astudillo**[2]      **Peter I. Frazier**[1]      **Ziv Scully**[1]      **Alexander Terenin**[1]

[1]Cornell University      [2]Caltech

## Abstract

Bayesian optimization is a technique for efficiently optimizing unknown functions in a black-box manner. To handle practical settings where gathering data requires use of finite resources, it is desirable to explicitly incorporate function evaluation costs into Bayesian optimization policies. To understand how to do so, we develop a previously-unexplored connection between cost-aware Bayesian optimization and the Pandora's Box problem, a decision problem from economics. The Pandora's Box problem admits a Bayesian-optimal solution based on an expression called the Gittins index, which can be reinterpreted as an acquisition function. We study the use of this acquisition function for cost-aware Bayesian optimization, and demonstrate empirically that it performs well, particularly in medium-high dimensions. We further show that this performance carries over to classical Bayesian optimization without explicit evaluation costs. Our work constitutes a first step towards integrating techniques from Gittins index theory into Bayesian optimization.

## 1   Introduction

Bayesian optimization is a framework for optimizing functions whose evaluation is time-consuming or expensive. It is widely used for hyperparameter tuning of machine learning algorithms [38], robot control [29], material design [47], and other areas. Bayesian optimization works by forming a probabilistic model for the objective function, and then chooses where to sample via an acquisition function that balances the explore-exploit trade-offs arising from uncertainty in this model.

We study *cost-aware* Bayesian optimization, where one must pay a cost to acquire another sample and this cost may vary with where the function is evaluated. Costs are an important factor in practical scenarios. For instance, in hyperparameter tuning using GPUs rented from a cloud provider, training a neural network for twice as many epochs may carry twice the financial cost.

Despite its practical relevance, cost-aware Bayesian optimization is less-studied than standard Bayesian optimization, where budgets are framed in terms of the number of function evaluations and costs are not explicitly considered. Many existing theoretically-principled cost-aware approaches [46, 24, 26, 4, 7] rely on multi-step lookahead computations that are computationally expensive and can be numerically brittle, limiting their applicability. Other approaches lack a theoretical foundation and risk having poor performance on certain problems. For example, one of the most popular cost-aware acquisition functions used in practice, expected improvement per unit cost [38], has recently been theoretically shown by Astudillo et al. [4] to perform arbitrarily worse than the optimal policy. Thus, in the cost-aware setting, there is a need for theoretically-principled and computationally-straightforward acquisition functions with good empirical performance.

In this work, we develop such an approach. To do so, we introduce a novel link between cost-aware Bayesian optimization and a discrete-space decision problem from economics called the *Pandora's*

38th Conference on Neural Information Processing Systems (NeurIPS 2024).

*Box* problem [44, 13, 36]. The Pandora's Box problem admits an explicit Bayesian-optimal solution. We show how this solution can be used to develop a novel acquisition function class for two cost-aware Bayesian optimization settings: (i) *expected budget-constrained* Bayesian optimization, where there is a constraint on the expected cost of the samples taken, and (ii) *cost-per-sample* Bayesian optimization where the total costs incurred are subtracted from the final objective function value. The resulting acquisition functions are closely connected to expected improvement variants, but incorporate costs in a different, non-multiplicative way.

We evaluate the proposed acquisition function, termed the *Pandora's Box Gittins Index (PBGI)*, on a comprehensive set of experiments to understand its strengths and weaknesses. On both sufficiently-easy low-dimensional problems and too-difficult high-dimensional ones, performance is comparable to baselines. On most medium-hard problems of moderate dimension, the proposed acquisition function either matches or outperforms the strongest baselines. Surprisingly, we find this performance carries over to the classical setting with uniform costs. We also discuss limitations, including behavior on problems where baselines are stronger.

The Pandora's Box Gittins Index is a version of the *Gittins index* [20], a general framework for deriving optimal policies for a variety of bandit-like decision problems [43, 14, 22] which is widely-used in queueing theory and related areas [21, 1, 34]. Other versions of the Gittins index have recently been proposed for Bayesian optimization with an infinite-horizon discounted cumulative regret objective [32]. Our Pandora's-Box-based formulation is instead designed for the simple regret objective more commo n in the literature [18]. Our work thus opens a novel angle of attack for designing acquisition functions specialized to specific practical settings of interest.

**Contributions.**    In this work, we (i) connect the Pandora's Box problem with a variant of cost-aware Bayesian optimization over a discrete search space. Using this connection, we (ii) explore the use of Gittins indices, which are Bayesian-optimal for the Pandora's Box problem, as an acquisition function for general cost-aware Bayesian optimization where data is incorporated via the posterior distribution. We (iii) demonstrate the resulting acquisition function has strong empirical performance on a variety of problems of moderate-to-high dimension, including the varying-cost problems it was designed for, as well as classical cost-unaware problems.

## 2    Cost-aware Bayesian optimization

In black-box optimization, we are interested in finding the global optimum of an unknown (potentially stochastic) function $f : X \to \mathbb{R}$ defined on some compact domain, using pointwise function evaluations of $f$ at locations $x_1, .., x_T \in X$ that we select sequentially. We are interested in policies that achieve low *simple regret*—see Garnett [18], Sec. 10.1—namely

$$\mathbb{E} \sup_{x \in X} f(x) - \mathbb{E} \max_{1 \le t \le T} f(x_t) \tag{1}$$

where the expectation is taken over all sources of randomness in the function $f$ and the policy, including the stopping time $T$ and sequential selections $x_1, .., x_T$. In our setup, obtaining a new function evaluation at a point $x$ carries a non-zero *cost* $c(x) \in \mathbb{R}_+$, assumed automatically-differentiable unless discussed otherwise. We consider settings that integrate costs into the problem in different ways:

 (a) In the *expected budget-constrained* setting, there is a budget $B \in \mathbb{R}_+$, and the algorithm is not allowed to exceed this budget in expectation.

 (b) In the *cost-per-sample* setting, at each time the algorithm must choose whether to pay a cost and obtain a new function evaluation, or to stop and return some previously-observed point. In this setting, we add the total sum of costs at termination time to the regret.

Note that the cost function $c : X \to \mathbb{R}_+$ can be constant, which we term *uniform costs*. In this case, (a) reduces to standard black-box optimization with a finite time horizon, and (b) reduces to a variant of stopping-aware Bayesian optimization. These are not the only possible settings: one can also consider other variants including the almost-sure budget-constrained setting where the algorithm is not allowed to exceed the budget in a strict manner. Since we are interested primarily in the role of costs rather than stopping times in this work, we mostly work with budget constraints throughout this paper, especially almost-sure budget constraints for our empirical results. We will use the cost-per-sample setting as a conceptual framework with which to study budget-constrained settings.

## 2.1 Probabilistic models and acquisition functions

*Bayesian optimization* algorithms for solving various black-box optimization problems work by (i) building a *probabilistic model* of $f$—that is, a probability distribution which quantifies what is known about $f$ given the data points $(x_t, y_t)_{t=1}^T$ seen so far—where $y_t = f(x_t)$ are previous function evaluations—then (ii) using the model and its uncertainty to decide where to evaluate the unknown function next. For an introduction, see Frazier [16] and Garnett [18]. Following standard practice, we work with Gaussian process models [33]. Let $f \mid x_{1:t}, y_{1:t}$ be the respective posterior distribution.

To decide where to evaluate $f$ next, one uses the model to define a (potentially random) *acquisition function* $\alpha_t : X \to \mathbb{R}$, which quantifies how promising a particular location is given what is known so far. We then evaluate $f$ at

$$x_{t+1} = \arg\max_{x \in X} \alpha_t(x), \qquad (2)$$

obtaining an additional data point that is used to reduce uncertainty and further improve the model.

## 2.2 Expected improvement per unit cost

The most popular cost-aware acquisition function is *expected improvement per unit cost (EIPC)* [38], defined via

$$\alpha_t^{\mathrm{EIPC}}(x) = \frac{\mathrm{EI}_{f|x_{1:t},y_{1:t}}(x; \max_{1 \le \tau \le t} y_\tau)}{c(x)} \qquad \mathrm{EI}_\psi(x; y) = \mathbb{E} \max(0, \psi(x) - y) \qquad (3)$$

where we have written $\alpha_t^{\mathrm{EIPC}}(\cdot)$ in terms of the general *expected improvement function* $\mathrm{EI}_\psi$, defined with respect to some random function $\psi : X \to \mathbb{R}$, and a comparator point $y$. With this notation, EIPC can be interpreted as the ratio of the expected improvement, with respect to the current posterior and using the best point seen so far as the comparator, to the cost.

In the uniform-cost case, where the costs $c(x) = C \in \mathbb{R}_+$ are constant for all $x$, this acquisition function reduces to the classical *expected improvement (EI)* acquisition function $\alpha_t^{\mathrm{EI}}(x) = \mathrm{EI}_{f|x_{1:t},y_{1:t}}(x; \max_{1 \le \tau \le t} y_\tau)$. In turn, expected improvement can be derived by considering the setup where the unknown function $f$ is randomly sampled from the model's prior. If we imagine that the optimization process continues for only one more time step before stopping, maximizing expected improvement is the optimal strategy in this one-step-lookahead scenario.

Since EIPC reduces to EI in the uniform-cost case where $c(x) = C$, it follows that it chooses the same points whether $C = 0.0001$ or $C = 1\,000\,000$. This is somewhat peculiar: one might expect that a cost-aware acquisition function should be more risk-averse if costs are high, and vice versa if they are low. Thus, EIPC is perhaps best suited to settings where heterogeneity of costs is the main factor at play: without heterogeneity, one can simply apply standard acquisition function variants [11]. However, even with heterogeneous costs, Astudillo et al. [4] show there exist reasonable problems where EIPC performs arbitrarily worse than the optimal policy in an approximation-ratio sense, due to over-sampling low-value low-cost points. Analogous behavior can occur in multi-fidelity settings, which Wu et al. [45] argue can be especially undesirable in the presence of model-misspecification.

In spite of this rather negative outlook, EIPC has been shown to work well on many practical problems, is computationally efficient and reliable, and is the standard cost-aware choice in BoTorch [5]. We therefore ask: *can one develop a technically-principled and computationally-straightforward alternative with at-least-comparable empirical performance?*

## 3 The Pandora's Box Gittins index for Bayesian optimization

To develop a cost-aware acquisition function, we study a simplified decision problem that captures key difficulties of the main problem but is tractable enough to yield analytic insights. An analogous strategy is used classically to derive expected improvement, by exactly solving a simplified one-step decision problem. We study a different simplified decision problem, which can also be solved exactly, but where the simplification is spatial rather than temporal in nature. Specifically, we connect Bayesian optimization with the *Pandora's Box* problem from economics. To do so, we describe Pandora's Box in Section 3.1 and its solution in Section 3.2, showing along the way how these ideas can be reinterpreted from the view of Bayesian optimization. We illustrate this in Figure 1. Then, in Section 3.3, we use Pandora's Box to derive a novel class of cost-aware acquisition functions.

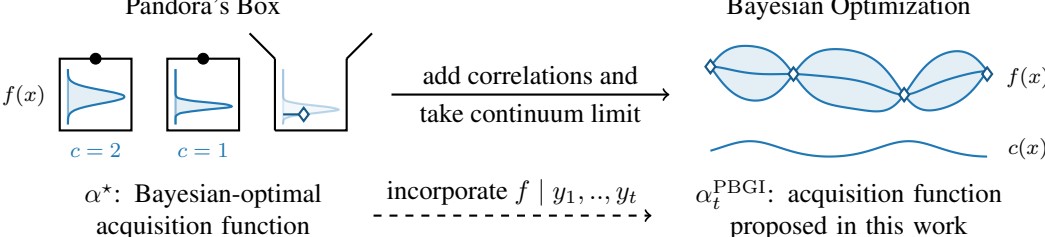

Figure 1: An illustration of this work's key idea. We view cost-aware Bayesian optimization as an extension of the Pandora's Box problem, and derive the cost-aware acquisition function $\alpha_t^{\text{PBGI}}$ by incorporating the posterior into the Bayesian-optimal Pandora's Box acquisition function $\alpha^\star$.

## 3.1 The Pandora's Box problem

The *Pandora's Box* problem [44, 20] is a sequential decision-making problem. It begins with a finite set of boxes, which we collect into a set and label $X = \{1, .., N\}$. Each box has a *hidden reward*, denoted by $f(x)$, and an *inspection cost*, denoted by $c(x)$. The rewards are given by mutually independent random variables whose distributions are known and vary between different boxes.

The decision-making process starts with a set of closed boxes, and proceeds in discrete time steps. At time $t$, one can choose to do one of two things:

1. *Open a box* $x_t \in X$. This incurs cost $c(x_t)$, but reveals the exact value $f(x_t)$ of the reward inside the box, which is drawn using the box's respective reward distribution.

2. *Stop opening new boxes, and take the reward from the best opened box.* This ends the decision-making process, and yields a terminal reward equal to the maximum value among the boxes opened so far, with the convention that at least one box must be opened.

The policy's goal is to maximize the expected net utility, which is the reward of the best open box minus the expected total costs of all boxes opened so far, and is written

$$\mathbb{E} \max_{1 \leq t \leq T} f(x_t) - \mathbb{E} \sum_{t=1}^{T} c(x_t) \tag{4}$$

where $T$ is a random variable that denotes the number of opened boxes, indicating that the policy terminates at time $T + 1$.

If we subtract the objective (4) from $\mathbb{E} \sup_{x \in X} f(x)$, which is constant with respect to the policy, we obtain the sum of the simple regret objective (1) defined in Section 2 and expected total costs. *The Pandora's Box problem is therefore equivalent to a special case of cost-aware black-box optimization*, specifically the cost-per-sample variant of Section 2, where (a) the domain $X$ is a finite set, and (b) the objective function $f$ is random, with independent $f(x)$ and $f(x')$ for $x \neq x'$. We will return to this point in the sequel, but first study the Pandora's Box problem in more detail.

## 3.2 Optimally solving Pandora's Box

The Pandora's Box problem gives rise to an explore-exploit tradeoff: a policy must balance the opportunity gained from learning the value of the reward contained inside the box with the cost of opening it. Since the reward distributions are known, this tradeoff is captured within a Markov decision process (MDP). By general MDP theory, there exists an optimal policy describing which box, if any, one should open for a given configuration—we call such a policy *Bayesian-optimal*.

This MDP can be solved explicitly, with a remarkably simple solution, first derived in an economics setting by Weitzman [44]. We start by associating with each box $x \in X$ a number $\alpha^\star(x)$ known as the *Gittins index* [20]. Define

$$\alpha^\star(x) = g \qquad \text{where } g \text{ solves} \qquad \text{EI}_f(x; g) = c(x) \tag{5}$$

where $\text{EI}_f(x; y)$, previously defined in (3) of Section 2, is the *expected improvement of $x$ relative to $y$*—the same expression which appeared in the expected improvement acquisition function variants

$\alpha_t^{\text{EI}}$ and $\alpha_t^{\text{EIPC}}$. Note that, unlike in those cases, $\alpha^\star$ is *not time-dependent* due to the lack of correlations or conditioning. Since $\text{EI}_f(x;g)$ is strictly decreasing in $g$, the root-finding problem (5) admits a unique solution for every value of $c(x)$.

To understand what $\alpha^\star(x)$ represents, consider a single closed box $x$, and suppose there is a second, open box with reward $f^*$. Is opening box $x$ better than taking the reward $f^*$ from the open box? This amounts to whether the expected improvement from opening $x$ balances out the opening cost $c(x)$: one can show that opening $x$ is better if and only if $\text{EI}_f(x;f^*) > c(x)$. The value $\alpha^\star(x)$ tells us *how large does the alternative reward $f^*$ need to be, for stopping and taking it to be at least as good as opening box $x$*—a kind of *fair value* which makes different boxes directly comparable to one another.

If we decide which box to open via the aforementioned fair values, we obtain the *Gittins index policy*, which proceeds as follows. At each time $t$, let $f_t^* = \max_{1 \le \tau \le t} f(x_\tau)$ be the maximum reward among all open boxes, and let $x_t^*$ be the box of maximum Gittins index value $\alpha^\star(x)$ among unopened boxes, with ties broken according to a given ordering. With this notation, using a tie-breaking rule that stops as early as possible—but noting that other tie-breaking rules, including stopping as late as possible, or stopping with some probability, are also valid—we get:

- If $f_t^* < \alpha^\star(x_t^*)$, the policy opens box $x^*$.
- If $f_t^* \ge \alpha^\star(x_t^*)$, the policy stops and receives terminal reward $f_t^*$.

It turns out that opening boxes according to the order determined by their fair values, in the sense above, is not only a good idea, but is outright Bayesian-optimal. We state this formally as follows.

**Theorem 1** (Weitzman [44]). *Let $X$ be a finite set, let $f : X \to \mathbb{R}$ be a finite-mean random function for which $f(x)$ is independent of $f(x')$ for $x \ne x'$, and let $c : X \to \mathbb{R}_+$, without loss of generality, be deterministic. Then, for the cost-per-sample problem, the policy defined by maximizing the Gittins index acquisition function $\alpha^\star$ with its associated stopping rule is Bayesian-optimal.*

In the language of Bayesian optimization, this means that not only is there an explicit Bayesian-optimal policy for the Pandora's Box setting, but this policy also *takes the form of maximizing an acquisition function*. This gives an explicit solution for the cost-per-sample setting, thereby showing Pandora's Box fits our original goal of finding a simplified decision problem that sheds insights on cost-aware Bayesian optimization. For an alternative proof, see Kleinberg et al. [25], Theorem 1. Using Lagrangian relaxation, we show that the obtained solution carries over to the expected budget-constrained setting.

**Theorem 2.** *Consider the expected budget-constrained problem, with the assumptions of Theorem 1. Assume the problem is feasible and the constraint is active, namely $\min_{x \in X} c(x) < B < \sum_{x \in X} c(x)$. Then there exists a $\lambda > 0$ and a tie-breaking rule such that the policy defined by maximizing the Gittins index acquisition function $\alpha^\star(\cdot)$, defined using costs $\lambda c(x)$, is Bayesian-optimal.*

A proof is given in Appendix B. This result extends a special case of Aminian et al. [3], Theorem 1. Compared to that work, we consider only Pandora's Box, but allow general reward distributions—including those with infinite support, such as Gaussian rewards. The optimal $\lambda$ depends on the budget constraint $B$ implicitly via a convex optimization problem given in Appendix B. In budget-constrained problems, we therefore view $\lambda$ as a hyperparameter, which controls the degree to which the algorithm is risk-averse vs. risk-seeking—precisely what we argued was missing from EIPC in Section 2.

One can intuitively understand $\lambda$ using a needle-in-haystack metaphor. Suppose one wants to find the best needle, but can only search a fraction of the haystack in expectation, represented by the budget $B$. The key phrase is *in expectation*: if the search is not promising, one can stop early and avoid wasting budget, otherwise one can search more of the haystack. This prompts the question: in what situations should one continue searching? The answer depends on the interplay between the budget $B$, costs $c$, and best value $f_t^*$ seen so far, and is encoded by $\lambda$ in Theorem 2. Larger $\lambda$-values correspond to smaller budgets $B$, incentivizing one to search less of the haystack. Crucially, the optimal acquisition function $\alpha^\star$ behaves differently for different $\lambda$-values, and therefore for different budgets: roughly, smaller budgets cause $\alpha^\star$ to explore less. We will return to this in Section 3.3.3 and Figure 3.

## 3.3 An acquisition function class for cost-aware Bayesian optimization

To adapt $\alpha^\star$ to the Bayesian optimization setting, we need to handle two differences: (i) $X$ does not need to be discrete, and (ii) a general probabilistic model is used for $f$. Since Theorem 1 ostensibly

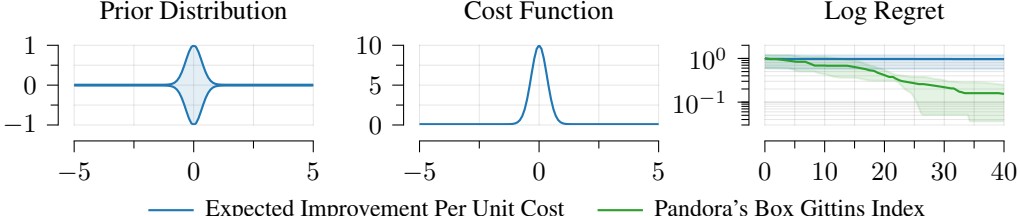

Figure 2: A Bayesian optimization problem with varying costs on which LogEIPC—a numerically stable implementation of EIPC, see Section 4 and Appendix D.1—has poor performance, inspired by Astudillo et al. [4], Section A. The domain is $X = [-500, 500]$, which we visualize on the subinterval $[-5, 5]$. Left: illustration of the non-uniform prior variance, which is given by a Matérn-5/2 kernel scaled by a narrow bump function. Center: the cost function, which is a narrow bump-shaped function. Right: median regret curves and quartiles for LogEIPC and PBGI. Legend refers only to regret curves.

requires $f(x)$ to be independent of $f(x')$ for all $x \neq x'$, the key question is *how to incorporate data and spatial correlations* into $\alpha^\star$. We propose to do so in the simplest and most obvious way: namely, at each time $t$, we plug the posterior distribution $f \mid x_{1:t}, y_{1:t}$ in place of $f$. This yields three variants, depending on the precise cost-aware setting one is interested in:

1. *Budget-constrained*: define *Pandora's Box Gittins index (PBGI)* acquisition function
$$\alpha_t^{\mathrm{PBGI}}(x) = g \qquad \text{where } g \text{ solves} \qquad \mathrm{EI}_{f \mid x_{1:t}, y_{1:t}}(x; g) = \lambda c(x) \qquad (6)$$
and $\lambda$ is a hyperparameter that should be tuned to match the evaluation budget $B$.

2. *Cost-per-sample*: we can directly apply $\alpha_t^{\mathrm{PBGI}}$ in this setting as well, but now $\lambda$ is instead interpreted as unit-conversion factor which ensures costs and rewards have the same units, and the Pandora's Box stopping rule is used for deciding when to terminate the optimization procedure and return the best observed value.

3. *Adaptive decay*: here, we do not have a pre-defined budget or cost-based stopping rule. Define the *Pandora's Box Gittins index with adaptive decay (PBGI-D)* acquisition function $\alpha^{\mathrm{PBGI\text{-}D}}(x)$ analogously to $\alpha^{\mathrm{PBGI}}(x)$, but with a time-dependent $\lambda_t$ schedule, set according to the Pandora's Box stopping rule. Specifically, $\lambda_0$ is initialized to a given value, then set
$$\lambda_{\tau+1} = \begin{bmatrix} \dfrac{\lambda_\tau}{\beta} & \text{if } f_\tau^* \geq \alpha_\tau^{\mathrm{PBGI}}(x_\tau^*; \lambda_\tau) \\[2ex] \lambda_\tau & \text{otherwise} \end{bmatrix} \qquad (7)$$
where $\beta > 1$ is a decay factor that is used to decrease $\lambda_\tau$ at any time $\tau$ when the Pandora's Box stopping rule triggers. The advantage of this variant is that it can be more robust to different ranges of the policy hyperparameters: we discuss this further in Appendix D.3.

To understand this acquisition function class, one can think of it via the following approximation: for the general cost-aware Bayesian optimization problem, we (a) correctly incorporate observed data into the prior to obtain the posterior, but then (b) pick new samples according to the rule that would have been Bayesian-optimal if the posterior had no correlations. Said differently, $\alpha^{\mathrm{PBGI}}$ arises from exactly solving a simplified dynamic program, where the simplification is of a spatial nature, rather than the usual temporal lookahead. One can therefore expect this acquisition function to work best in situations where correlations are not the decisive factor for determining performance.

In what problems does this happen? In stationary kernels, correlations encode local dependence. Therefore, one can expect $\alpha^{\mathrm{PBGI}}$ to be approximately-optimal in settings where the key decisions involve choosing between different far-away data points. One can intuitively expect this to occur more often in high-dimensional problems, where the volume of the search space is large and most points are far away from each other. We will examine this point empirically in the sequel.

**3.3.1 Computation.** To compute $\alpha^{\mathrm{PBGI}}$ efficiently, note that $y \mapsto \mathrm{EI}_\psi(x; y)$ is monotone. As a result, the value $g$ in (6) can be computed efficiently via bisection search. In Appendix B.2, we show that (i) the gradient of $\alpha^{\mathrm{PBGI}}$ can be computed straightforwardly via an explicit analytical expression without any additional optimization, and (ii) the resulting computational costs are closer to those of expected improvement than those of expensive multi-step-lookahead-based approaches.

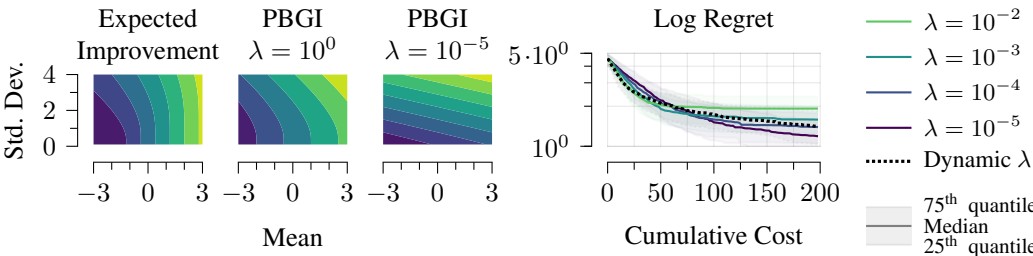

Figure 3: Left: contour plots showing how EI (left) and PBGI (center-left, center-right) depend on the posterior mean and standard deviation at a given point (lighter colors indicate higher values). We see that PBGI values high standard deviation more than EI. Right: PBGI performance across values of $\lambda$, under the setup of the Bayesian regret experiment of Section 4 with $d = 8$. We plot the median of a set of samples using $n = 256$ random seeds, along with quartiles to show variability. We see that large $\lambda$-values decrease regret sooner, but eventually lose out to smaller $\lambda$-values.

**3.3.2 Extension to stochastic and non-automatically-differentiable costs.** One can show that Theorem 1 holds in the more general case where the cost function is random, as long as one substitutes $c(x)$ with its mean. The same holds for Theorem 2, as shown in Appendix B.5. In this setting, therefore, stochasticity of $c(x)$ affects performance, but does not change the optimal strategy. Building on these observations, we propose extensions of $\alpha_t^{\mathrm{PBGI}}$ to the more general setting where costs are modeled using a log-normal process in Appendix B.4.

**3.3.3 Qualitative behavior and comparisons.** Compared to cost-unaware acquisition functions such as EI, PBGI can be more risk-averse if costs are large or more risk-seeking if costs are small. In varying-cost budget-constrained settings, this tradeoff is mediated by $\lambda$, and the obtained decisions can differ significantly from those of widely-used baselines such as EIPC. In particular, PBGI can make qualitatively different decisions on problems where there is a high-variance point with a large cost, among a set of many low-variance low-cost points. In Figure 2, we adapt the construction of Astudillo et al. [4], Section A into a one-dimensional Bayesian optimization problem with a non-stationary prior, and observe that EIPC—even in its numerically stable logarithmic form described in Section 4—has substantially worse performance than PBGI.

The PBGI acquisition functions depends on $f \mid x_{1:t}, y_{1:t}$ through its mean and standard deviation at each point. We plot this in Figure 3. This shows for large $\lambda$ that PBGI can resemble EI, whereas for small $\lambda$ it is nearly linear. Specifically, in the $\lambda \to 0$ limit, we have

$$\alpha_t^{\mathrm{PBGI}}(\cdot) \approx \mu_t(\cdot) + \sigma_t(\cdot)\sqrt{2\log\frac{\sigma_t(\cdot)}{\lambda c(\cdot)}} \tag{8}$$

where $\mu_t$ and $\sigma_t$ are the mean and standard deviation of $f \mid x_{1:t}, y_{1:t}$. This expression is similar to the upper confidence bound (UCB) acquisition function whose dependence is exactly linear, with heterogeneous learning rate parameter set to $\eta_t = \sqrt{2\log\frac{\sigma_t(x)}{\lambda c(x)}}$. A derivation is given in Appendix B.3. For small $\lambda$, one can thus view PBGI as giving a way to automatically tune UCB's confidence parameter in a careful way depending on $c(x)$.

## 4 Experiments

We now empirically evaluate the Gittins-index-based acquisition function on cost-aware problems. We also evaluate on the same problems with a spatially-constant cost function, a setting we term *uniform costs*—this facilitates comparisons with classical, cost-unaware baselines. In both cases, mirroring practical settings, we work with a deterministic, algorithm-independent evaluation budget.

We implement all methods in BoTorch [5] using Matérn Gaussian processes with smoothness $\nu = 5/2$. For Bayesian regret experiments where objectives are sampled from the prior, we fix the length scale to be $\kappa = 10^{-1}$ for both the prior and the posterior. For synthetic and empirical experiments, we apply maximum marginal likelihood optimization to dynamically adjust the length scale every

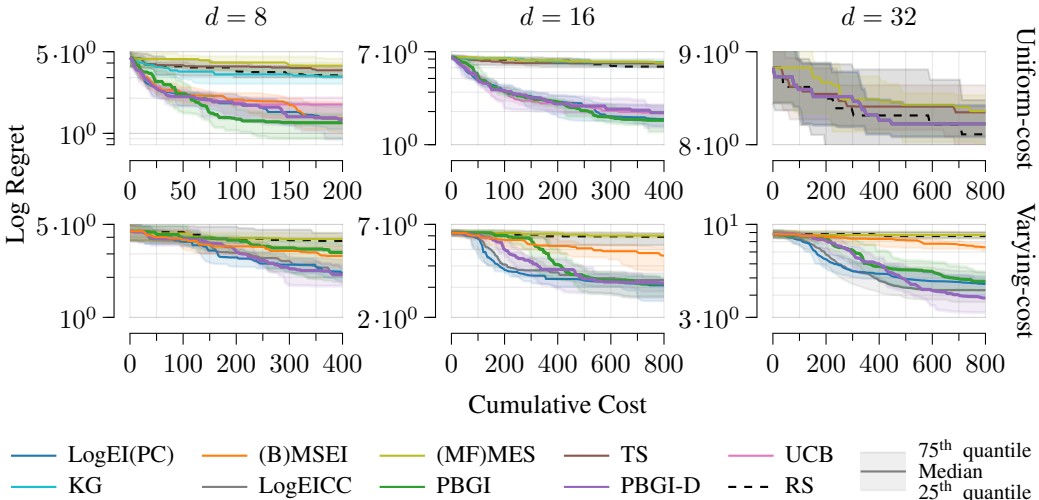

Figure 4: Regret curves for objective functions sampled from the prior, shown using medians, as well as quartiles to indicate experiment variability. We see in the cost-aware setting that both PBGI variants usually exhibit comparable performance to LogEIPC and LogEICC, with PBGI-D decisively outperforming other baselines. In the uniform-cost setting, the story is similar for $d = 8$ and $d = 16$: PBGI and PBGI-D perform comparably to the best baselines, which are LogEI and UCB, as well as MSEI for $d = 8$. However, for $d = 32$, all methods perform comparably to random search, with PBGI, PBGI-D, LogEI, and UCB having near-identical median performance. See Figure 14 for an alternative visualization using mean and standard error.

iteration. To ensure that our results are not sensitive to these and other hyperparameter choices, all experiments were repeated with alternatives given in Appendix D.6. Each experiment was repeated for 16 seeds to assess variability, unless stated otherwise. Experimental details are in Appendix C.

We evaluate both PBGI variants from Section 3.3, namely $\alpha^{\mathrm{PBGI}}$ with $\lambda = 10^{-4}$, and $\alpha^{\mathrm{PBGI-D}}$ with $\lambda_0 = 10^{-1}$ and $\beta = 2$. To assure ourselves that performance differences are not primarily due to tuning, we deliberately use the same $\lambda$-values for PBGI and the same $(\lambda_0, \beta)$-values for PBGI-D on all problems. Comparisons with other $(\lambda_0, \beta)$-values can be found in Appendix D.3.

For varying-cost problems—that is, those with spatially non-constant cost—we compare with *(log) expected improvement per unit cost (LogEIPC)* [37, 38], *(log) expected improvement with cost cooling (LogEICC)* [27], *multi-fidelity max-value entropy search (MFMES)* [40] and *budgeted multi-step expected improvement (BMSEI)*, [4], which was shown in that work to have state-of-the-art cost-aware performance. For uniform-cost problems—that is, those with constant costs—we compare with *log expected improvement (LogEI)* [2], *Thompson sampling (TS)*, *upper confidence bound (UCB)* [39], *max-value entropy search (MES)* [42], *knowledge gradient (KG)* [17], and *multi-step expected improvement (MSEI)* [24]. We choose these because (i) they are standard, and (ii) acquisition function optimization succeeds for them on our problems, reducing confounding. We work with logarithmic expected improvement variants (LogEI, LogEIPC, LogEICC), as they have recently been shown to have substantially better numerical and optimization properties than their traditional counterparts [2]—see Appendix D.1 for details.

## 4.1 Bayesian regret

For our first experiment, we examine how well the proposed acquisition functions perform on random functions sampled from the prior. To quantify the effect of problem difficulty, we vary the dimension of the domain $X = [0, 1]^d$, and consider $d \in \{8, 16, 32\}$. Results, in terms of empirical regret curves and their associated quartiles, are shown in Figure 4. Additional results for $d = 4$, demonstrating comparable performance between both PBGI variants and all baselines, as well as sensitivity comparisons involving the Gaussian process prior with varying kernel hyperparameters such as different smoothness values and length scales, are included in Appendix C.

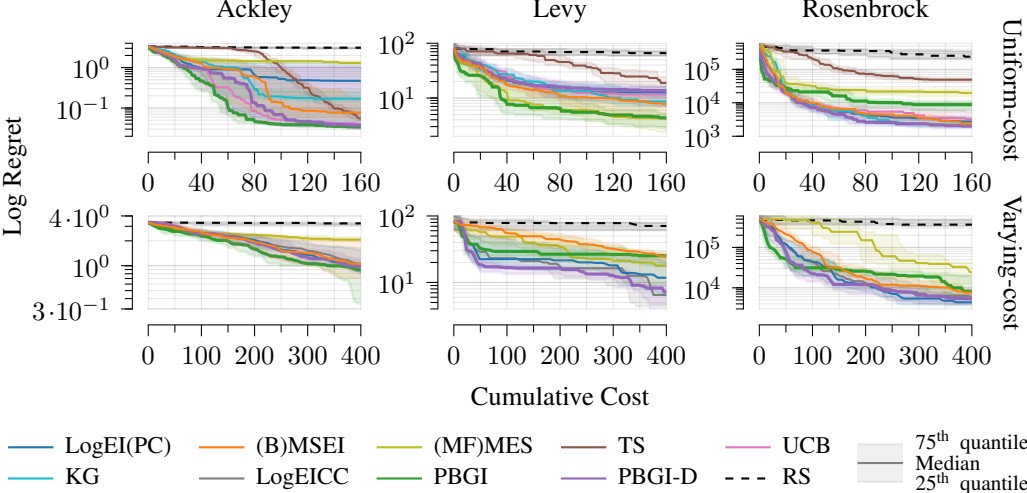

Figure 5: Synthetic benchmark regret curves, shown using medians, as well as quartiles to assess variability. All objective functions are defined with dimension $d = 16$. We see in the cost-aware setting that PBGI, PBGI-D, LogEIPC, and LogEICC all perform similarly on the heavily-multimodal Ackley function, matching or outperforming the non-myopic BMSEI baseline. On the Levy and Rosenbrock functions, PBGI-D matches—and for some cost budgets outperforms—all baselines, including the non-myopic BMSEI. Under uniform costs, PBGI performs well on Ackley and Levy, but is outperformed by PBGI-D and most baselines on Rosenbrock. See Figure 14 for an alternative visualization using mean and standard error.

In the low-dimensional case of $d = 8$, PBGI and LogEI variants achieve similar or better performance to the non-myopic (B)MSEI baseline. Once we increase dimension to $d = 16$, we see bigger differences, with PBGI variants, LogEI variants, and UCB now decisively outperforming (B)MSEI. Notably, the PBGI variants are also competitive in the uniform-cost setting—in spite of being designed for cost-aware problems. This can be explained via the curse of dimensionality: as dimension increases, the problem begins to look more like the uncorrelated Pandora's Box problem where using Gittins index is Bayesian-optimal. Eventually, however, the problem becomes too difficult for meaningful progress to be made within our computational budget, as seen for the uniform-cost problem with $d = 32$, where all deterministic methods perform near-identically and no method outperforms random search.

## 4.2 Synthetic benchmarks

Next, we consider standard synthetic global optimization benchmark functions. To represent a variety of geometric properties, we examine the *Ackley*, *Levy*, and *Rosenbrock* functions. A visualization of the two-dimensional versions of these functions is given in Appendix A.

Figure 5 presents results for $d = 16$. Additional results for $d \in \{4, 8\}$ showing that PBGI and all baselines perform similar, are given in Appendix C. We see that the behavior of different acquisition functions varies according to the the function. On the Ackley function, PBGI outperforms PBGI-D and the baselines. In contrast, on the Levy function, PBGI is best in the uniform-cost setting, but PBGI-D, LogEIPC, and LogEICC are best in the varying-cost setting. We conclude that PBGI can in principle offer stronger performance than PBGI-D, as long as $\lambda$ is not-too-suboptimal, while PBGI-D tends to be less-performant but is more robust to this hyperparameter choice.

We also examine performance on the banana-shaped Rosenbrock-function, which has a small number of local optima [35]. In this case, PBGI-D performs the strongest, matching the LogEI variants, while outperforming PBGI and (B)MSEI. This can intuitively be explained by the one-step optimality of expected improvement, which better-exploits the less-multimodal objective, while PBGI and multi-step-based acquisition functions are more conservative and may therefore require different tuning to perform best. We conclude that PBGI-D may be a better choice in settings where there is a

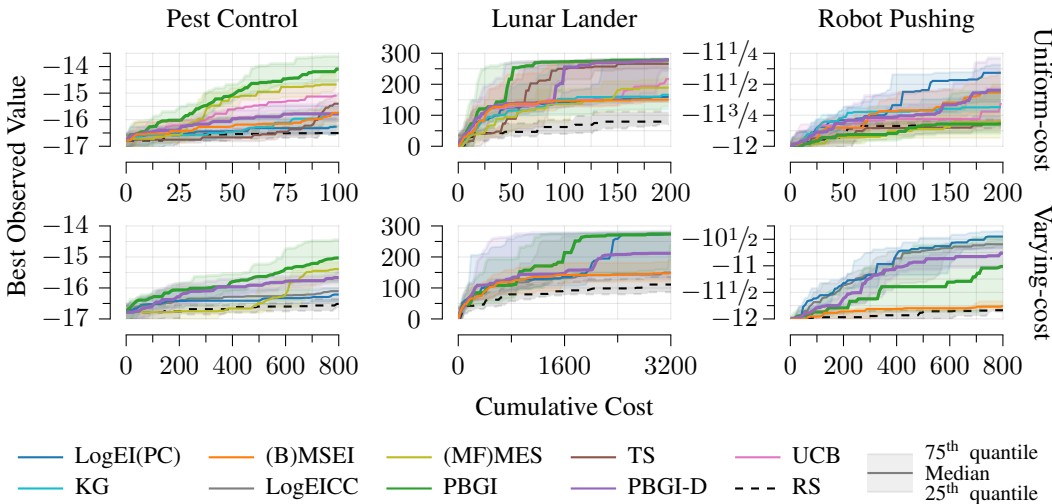

Figure 6: Empirical benchmark regret curves, shown using medians, as well as quartiles to show variability. We see in both the varying-cost and uniform-cost settings that PBGI exhibits strong performance on the Pest Control and Lunar Lander problems. On the Robot Pushing problem, LogEI variants perform strongest, with PBGI-D performing second-strongest and matching or outperforming all other baselines. See Figure 14 for an alternative visualization using mean and standard error.

potential mismatch between the objective and the prior in terms of degree of multimodality, as this can be counteracted in part by its decay behavior, particularly compared to PBGI.

### 4.3 Empirical objectives

Finally, we benchmark PBGI policies on three empirical global optimization problems motivated by applied challenges: *Pest Control* where $d = 25$ [31], *Lunar Lander* where $d = 12$ [15], and *Robot Pushing* where $d = 14$ [42]. Detailed descriptions of these problems and associated cost functions are in Appendix C. Note that, for Lunar Lander and Robot Pushing, the cost functions used are not automatically-differentiable. To handle this challenge and illustrate how our acquisition function can be used when the cost function is unknown, we apply unknown-cost PBGI and baseline variants, where the costs are modeled using a second independent log-Gaussian process: details on this unknown-cost PBGI variant, including its analytic form, are given in Appendix B.4.

From Figure 6, we see that the PBGI matches or outperforms baselines on Pest Control and Lunar Lander, in both the varying-cost and uniform-cost settings. On the other hand, PBGI performs poorly on Robot Pushing, where instead LogEI variants perform best and PBGI-D performs second-best; the non-myopic BMSEI baseline also performs poorly. This mirrors behavior previously seen on the Rosenbrock function, from which we suspect that a mismatch between prior and objective multimodality may be at play here as well. Note also that unlike in the Bayesian regret experiments, UCB's performance is far from strongest. This may be in part because we tune UCB using the schedule of Srinivas et al. [39], which is derived specifically for Bayesian regret, and may be less-ideal for other settings. In comparison, PBGI-D works reasonably well on five of the six cases.

## 5 Conclusion

In this paper, we introduced a new acquisition function class for cost-aware Bayesian optimization, the *Pandora's Box Gittins index*, based on an unexplored connection between Bayesian optimization and the *Pandora's Box* problem from economics. We observed promising performance from two variants of this acquisition function class on both cost-aware problems which are the focus of this work, and, additionally, on classical uniform-cost problems. Performance gains tended to be largest on higher-dimensional and multi-modal problems. Our work constitutes a first step toward integrating ideas from Gittins index theory, including insights from generalizations of Pandora's Box, and related areas such as queueing theory, into Bayesian optimization.

## Acknowledgments

We thank James T. Wilson for his suggestions, which helped us significantly improve the paper's presentation. AT is grateful to Anastasios Angelopoulos for hosting him at UC Berkeley on November 17th, 2022: though the visit and planned talk that day had to be cancelled last-minute due to labor strikes, the cancellation ultimately resulted in an unexpected chain of events that, months later, led to AT and ZS exchanging ideas about Gaussian processes and Gittins index theory, which ultimately led to this work. PF was partially supported by AFSOR FA9550-20-1-0351. ZS was supported by the NSF under grant numbers CMMI-2307008, DMS-2023528, and DMS-2022448. AT was supported by Cornell University, jointly via the Center for Data Science for Enterprise and Society, the College of Engineering, and the Ann S. Bowers College of Computing and Information Science.

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

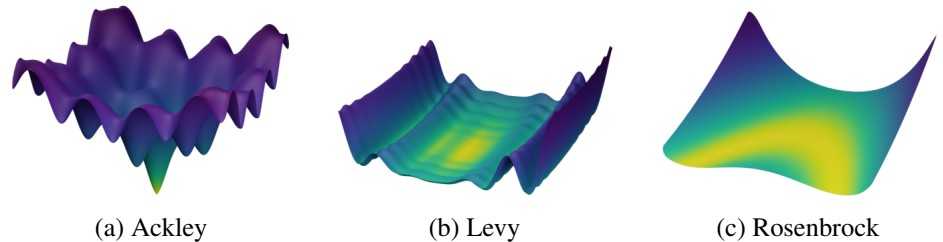

(a) Ackley        (b) Levy        (c) Rosenbrock

Figure 7: An illustration of the two-dimensional Ackley, Levy, and Rosenbrock functions.[1] From the visual, one can see that these functions differ in terms of multimodality and ridge-regions within the optimization landscape.

## A    Illustrations

Here we provide a set of additional illustrations to aid understanding of our results.

### A.1    Visualization of synthetic benchmark functions

To better understand how the behavior of Bayesian optimization algorithms on the three different synthetic benchmark functions might be affected by their geometric shape, Figure 7 provides a visual illustration of their two-dimensional variants.[1] This allows us to visually see the multimodality of the Ackley function, multimodality and ridge-like regions in the Levy function, and banana shape of the Rosenbrock function. While we use higher-dimensional versions of these in our experiments, this illustration provides some intuition for what the resulting the optimization landscape might look like, helping contextualize results.

### A.2    Comparison of behaviors between EI and PBGI

In Figure 8, we see that EI's maximum occurs at a point near $x \approx 0.365$, which is located closer to the observed data and is fairly close to the maximum of the posterior mean. Similarly, large-$\lambda$ PBGI's maximum occurs at a similar point near $x \approx 0.358$, indicating that the algorithm makes similar decisions in spite of the fact that the acquisition function's shape is different. In contrast, small-$\lambda$ PBGI's maximum occurs at a point near $x \approx 0.86$: it favors a riskier approach, preferring a location where the mean is smaller, but the variance is larger.

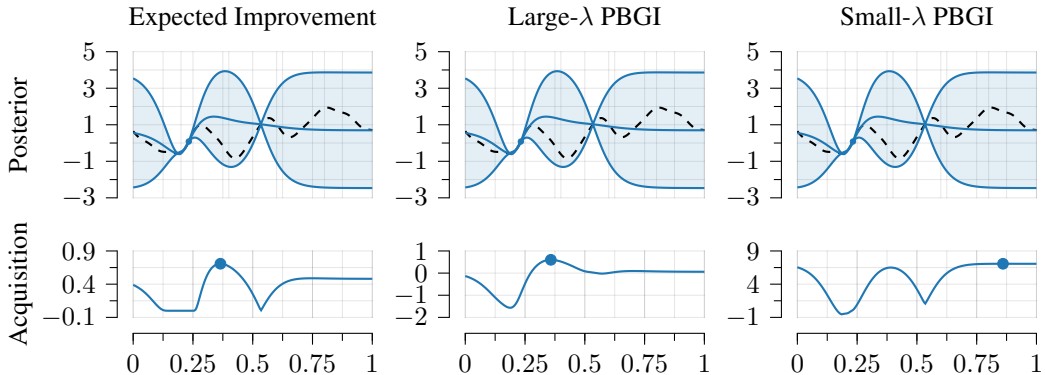

Figure 8: Comparison between three acquisition functions, namely Expected Improvement (EI), Pandora's Box Gittins Index (PBGI) with a large $\lambda$-value of $\lambda = 10^0$, and Pandora's Box Gittins Index with a small $\lambda$-value of $\lambda = 10^{-5}$. All three are computed using the same posterior distribution with four data points. The maximum is shown as a large dot.

---

[1]This visual originally appeared in Terenin [41], and is reproduced here with permission.

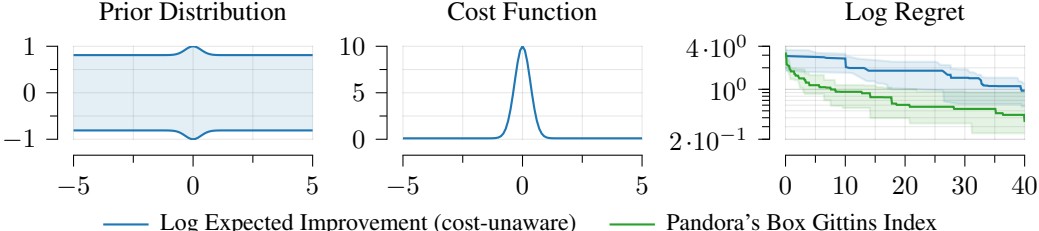

Figure 9: A Bayesian optimization problem with varying costs on which LogEI, a numerically stable implementation of EI—which by construction ignores costs—has poor performance. See Section 4 and Appendix D.1 for additional details. Like Figure 2, this construction also mirrors Astudillo et al. [4], Section A. The domain is $X = [-500, 500]$, which we visualize on the subinterval $[-5, 5]$. Left: illustration of the non-uniform prior variance, which is given by a Matérn-5/2 kernel scaled by a narrow bump function, plus a constant. Center: the cost function, which is a narrow bump-shaped function. Right: regret curves, shown as medians and quartiles.

This example shows potential differences in behavior between EI and PBGI, and illustrates how these differences are mediated by the parameter $\lambda$.

### A.3 An example where expected improvement underperforms

In Section 3.3.3, we showed a cost-aware Bayesian optimization problem on which the logarithmic form of expected improvement per unit cost (LogEIPC) has poor performance. Here, we show that this problem can be modified so that the logarithmic form of ordinary expected improvement (LogEI), which ignores the cost function, also has poor performance—a somewhat obvious, but nonetheless important sanity check that we make to ensure that costs play a sufficiently-important role in problems of this class to merit their consideration. This is shown in Figure 9. It is not hard to construct a less-visualization-friendly variant of these problems on which both expected improvement per unit cost and ordinary expected improvement perform poorly, by considering cost functions which are appropriately-weighted sums of bump functions.

## B   Theory and calculations

Below, we provide additional ideas to help understand the Pandora's Box problem and acquisition function that results from its considerations.

### B.1   Additional intuition on Pandora's Box

In what follows, we sketch a viewpoint from which one can see the key idea behind why Theorem 1 holds. Rather than considering the full Pandora's Box problem with a general set of open and closed boxes, consider first the case where there is exactly one closed and one open box. To slightly simplify notation, let $f$ denote the random reward inside the closed box, let $c$ denote the cost of opening the closed box, assumed deterministic, and let $g$ denote the visible reward of the open box. Our possible actions are as follows:

1. *Open the closed box.* In this case, we pay a cost of $c$, but subsequently get to choose between taking the realized value $f$, or instead taking $g$ from the box that was originally open. In expectation, the total value obtained by taking this action is $\mathbb{E}(\max(f, g)) - c$.

2. Take the reward from the open box. The total value obtained is $g$.

We can therefore analytically solve for the optimal policy of this respective Markov decision process: we open the closed box if $\mathbb{E}(\max(f, g)) - c \geq g$, and take the reward from the open box if $\mathbb{E}(\max(f, g)) - c \leq g$, with both actions optimal in the case of equality. As consequence, if $g$ is such that both actions are optimal, the same value is obtained no matter whether one chooses to open the box or not. Rewriting the preceding expressions slightly, this occurs when

$$\mathbb{E}\max(f - g, 0) = c \tag{9}$$

where in the case of multiple boxes the left-hand-side becomes the expected improvement function. The insight of Weitzman [44]—and indeed of Gittins [19] in a much more general setting—is that one can modify the Pandora's Box Markov decision process by replacing closed boxes with open boxes whose value $g$ satisfies (9) without changing the optimal policy. As a consequence, the optimal policy is precisely the Gittins index policy of Theorem 1.

As a final point, note that the assumption that $c$ is deterministic is made without loss of generality: if $c$ is instead stochastic but has finite expectation, the same reasoning applies, but with the value $c$ in (9) replaced with its expected value. We will make use of this in Appendix B.4.

## B.2 Gradient of the PBGI acquisition function

In most Bayesian optimization setups, gradient-based methods including multi-start stochastic gradient descent, BFGS, and L-BFGS-B, are used to effectively optimize analytical acquisition functions, such as EI and UCB. These methods can also be used to optimize the PBGI acquisition function. To facilitate this, we provide the gradient formula for the PBGI acquisition function. In what follows, mirroring the preceding and following sections, if costs are stochastic then $c(x)$ should be replaced with its respective mean. We also omit time subscripts to ease notation.

**Proposition 3** (Gradient of PBGI). *Let $\mu(x)$ and $\sigma(x)$ be the mean and standard deviation of the posterior Gaussian process $(f \mid x_{1:t}, y_{1:t})(x)$. With this notation, the gradient of the acquisition function $\alpha^{\mathrm{PBGI}}(x)$ is given by*

$$\nabla \alpha^{\mathrm{PBGI}}(x) = \nabla\mu(x) + \frac{\phi\left(\frac{\mu(x)-\alpha^{\mathrm{PBGI}}(x)}{\sigma(x)}\right)\nabla\sigma(x) - \lambda\nabla c(x)}{\Phi\left(\frac{\mu(x)-\alpha^{\mathrm{PBGI}}(x)}{\sigma(x)}\right)}, \tag{10}$$

*where $\phi$ and $\Phi$ denote the density and cumulative distribution function of a standard normal distribution, respectively.*

*Proof.* Recall that when $\psi(x) \sim \mathrm{N}(\mu(x), \sigma(x))$ is Gaussian, the expected improvement with respect to the comparator $y$ is given as

$$\mathrm{EI}_\psi(x; y) = (\mu(x) - y)\Phi\left(\frac{\mu(x) - y}{\sigma(x)}\right) + \sigma(x)\phi\left(\frac{\mu(x) - y}{\sigma(x)}\right). \tag{11}$$

Next, note by definition of $\alpha^{\mathrm{PBGI}}$, we have

$$\mathrm{EI}_{f|x_{1:t}, y_{1:t}}(x; \alpha^{\mathrm{PBGI}}(x)) = \lambda c(x). \tag{12}$$

Differentiating this with respect to $x$ on both sides gives

$$\nabla \mathrm{EI}_{f|x_{1:t}, y_{1:t}}(x; \alpha^{\mathrm{PBGI}}(x)) = \lambda\nabla c(x). \tag{13}$$

Applying the product and chain rule to the left-hand-side gives

$$\nabla \mathrm{EI}_{f|x_{1:t}, y_{1:t}}(x; \alpha^{\mathrm{PBGI}}(x)) = (\nabla\mu(x) - \nabla\alpha^{\mathrm{PBGI}}(x))\Phi\left(\frac{\mu(x) - \alpha^{\mathrm{PBGI}}(x)}{\sigma(x)}\right) \tag{14}$$

$$+ (\mu(x) - \alpha^{\mathrm{PBGI}}(x))\phi\left(\frac{\mu(x) - \alpha^{\mathrm{PBGI}}(x)}{\sigma(x)}\right)\nabla\left(\frac{\mu(x) - \alpha^{\mathrm{PBGI}}(x)}{\sigma(x)}\right) \tag{15}$$

$$+ \nabla\sigma(x)\phi\left(\frac{\mu(x) - \alpha^{\mathrm{PBGI}}(x)}{\sigma(x)}\right) \tag{16}$$

$$+ \sigma(x)\phi'\left(\frac{\mu(x) - \alpha^{\mathrm{PBGI}}(x)}{\sigma(x)}\right)\nabla\left(\frac{\mu(x) - \alpha^{\mathrm{PBGI}}(x)}{\sigma(x)}\right). \tag{17}$$

Recall the identity for the derivative of the Gaussian density, namely

$$\phi'(x) = -x\phi(x). \tag{18}$$

Applying this identity to (17) gives

$$\sigma(x)\phi'\left(\frac{\mu(x) - \alpha^{\mathrm{PBGI}}(x)}{\sigma(x)}\right)\nabla\left(\frac{\mu(x) - \alpha^{\mathrm{PBGI}}(x)}{\sigma(x)}\right) \tag{19}$$

$$= -(\mu(x) - \alpha^{\mathrm{PBGI}}(x))\phi\left(\frac{\mu(x) - \alpha^{\mathrm{PBGI}}(x)}{\sigma(x)}\right)\nabla\left(\frac{\mu(x) - \alpha^{\mathrm{PBGI}}(x)}{\sigma(x)}\right) \tag{20}$$

which is equal to the negation of (15), hence (15) and (17) cancel: we get

$$(\nabla\mu(x)-\nabla\alpha^{\mathrm{PBGI}}(x))\Phi\left(\frac{\mu(x)-\alpha^{\mathrm{PBGI}}(x)}{\sigma(x)}\right)+\nabla\sigma(x)\phi\left(\frac{\mu(x)-\alpha^{\mathrm{PBGI}}(x)}{\sigma(x)}\right)=\lambda\nabla c(x). \quad (21)$$

Rearranging this gives the expression in the claim. $\qquad\square$

### B.3 Small-cost limit of PBGI acquisition function

We now derive the small-$\lambda$ limiting expression for PBGI which was presented in Section 3.3.3. According to (11) and (12), we have

$$\alpha_t^{\mathrm{PBGI}}(\cdot)=\mu_t(\cdot)-\sigma_t(\cdot)u(\cdot) \quad (22)$$

where $\mu_t$ and $\sigma_t$ are the mean and the standard deviation of $f\mid x_{1:t},y_{1:t}$, and $u(\cdot)$ is the solution of

$$u(\cdot)\Phi(u(\cdot))+\phi(u(\cdot))=\frac{\lambda c(\cdot)}{\sigma_t(\cdot)}. \quad (23)$$

When $\lambda\to0$, the solution satisfies $u(\cdot)\to-\infty$. This implies that both $\Phi(u(\cdot))\to0$ and $\phi(u(\cdot))\to0$. Since $\phi(u(\cdot))$ dominates $\Phi(u(\cdot))$, we can approximate the solution $u(\cdot)$ by $\phi(u(\cdot))\approx\frac{\lambda c(\cdot)}{\sigma_t(\cdot)}$. By substituting the probability density function of a standard normal distribution for $\phi$, we obtain

$$u(\cdot)\approx-\sqrt{2\log\left(\frac{\sigma_t(\cdot)}{\lambda c(\cdot)}\right)}. \quad (24)$$

Thus, we can conclude that in the limit $\lambda\to0$, the Gittins index becomes

$$\alpha_t^{\mathrm{PBGI}}(x;\lambda)\approx\mu_t(x)+\sigma_t(x)\sqrt{2\log\left(\frac{\sigma_t(x)}{\lambda c(x)}\right)}. \quad (25)$$

As consequence, we expect PBGI-D, whose $\lambda$-parameter eventually decays to zero, to potentially inherit consistency and other properties known for UCB. Specifically, since PBGI-D decreases the value of $\lambda$ to zero over time by dividing it with a constant every time the Gittins stopping rule triggers, its behavior should eventually be well-described by the above limit. As a result of this limit, unexplored regions where $\sigma(x)\gg0$ should eventually have acquisition values that are larger than regions where $\sigma(x)\approx0$. We believe this should hold regardless of whether the costs are known—our main focus—or unknown, especially if the cost function $c(x)$ is uniformly bounded above and below. This because (25)'s dominant term in the $\lambda\to0$ limit is $\sigma(x)\sqrt{2\log\frac{1}{\lambda}}$, which does not depend on the cost $c(x)$.

### B.4 Closed-form expression for the PBGI unknown-cost variant

In the Lunar Lander and Robot Pushing empirical examples of Section 4, the cost function does not admit an analytic, automatically-differentiable form, and can only be evaluated in a black-box manner. To handle this, we model the logarithm of the costs as a Gaussian process, and condition this process on the costs observed at locations evaluated so far. This mirrors how unknown costs are handled in Astudillo et al. [4] for other acquisition functions, such as EIPC and BMSEI.

From the viewpoint of the Pandora's Box problem, stochastic costs make little difference: following the discussion in Appendix B.1, the optimality results of Weitzman [44] continue to hold even if costs are stochastic, so long as the costs in the formula for $\alpha^\star$ are replaced with expected costs. Mirroring this, if we plug in the mean of a log-normal random variable into the definition of $\alpha^{\mathrm{PBGI}}$, we obtain the following acquisition function.

**Definition 4.** *Let $c(x)$ be log-normal for all $x$. For a dataset $(x_1,y_1),..,(x_t,y_t)$, let $\mu_{\ln c}$ and $\sigma_{\ln c}$ be the posterior mean and posterior standard deviation of the log-costs. Define the* UNKNOWN-COST PANDORA'S BOX GITTINS INDEX ACQUISITION FUNCTION *by*

$$\alpha_t^{\mathrm{PBGI\text{-}U}}(x)=g \quad \text{where } g \text{ solves} \quad \mathrm{EI}_{f\mid x_{1:t},y_{1:t}}(x;g)=\lambda\exp\left(\mu_{\ln c}(x)+\frac{\sigma_{\ln c}(x)^2}{2}\right). \quad (26)$$

The interpretation of $\lambda$, namely as a hyperparameter that determines the expected budget the algorithm will use before reaching its respective stopping time, is the same as in the known-cost setting. One can define cost-per-sample and decay variants in the same manner as well.

## B.5 Relationship between expected budget-constrained and cost-per-sample problems

We now prove Theorem 2. In what follows, recall that the assumptions of Theorem 1 are (i) $X$ is discrete, (ii) $\mathbb{E}\,|f(x)| < \infty$ for all $x$, (iii) $f(x)$ and $f(x')$ are independent for $x \neq x'$, and (iv) the budget satisfies $B > \min_{x \in X} c(x)$. Further, Theorem 1 was stated for deterministic costs: more generally, we allow for stochastic costs satisfying $0 < \mathbb{E}\,c(x) < \infty$, and in such cases instead define $\alpha^\star$ using the expected costs.

We begin by defining the Markov decision process (MDP) under study and stating the properties of it that we will use. Define:

1. States: let $\mathcal{S} = \mathcal{S}^\circ \cup \partial\mathcal{S}$ be the union of two disjoint sets, namely the set $\mathcal{S}^\circ$ of non-terminal states and set $\partial\mathcal{S}$ of terminal states. These are described below:

   (a) Non-terminal states: let $\mathcal{S}^\circ$ consist of all finite sequences of length $|X|$ taking values in $\mathbb{R} \cup \{\boxtimes\}$, where numbers represent the reward inside an open box, and $\boxtimes$ represents a closed box, along with terminal states described below.

   (b) Terminal states: let $\partial\mathcal{S} = \mathbb{R}$ represent the reward of the box chosen by the learner.

2. Actions: let $\mathcal{A} = X$, where actions represent either opening a closed box, or taking a reward from an open box.

3. Costs: define the non-terminal cost function $c : \mathcal{S} \times \mathcal{A} \to \mathbb{R}$ by $c(s, a) = c(a)$.

4. Rewards: define the terminal reward function $\partial r : \partial\mathcal{S} \to \mathbb{R}$ by $\partial r(s) = s$.

5. Transition kernel: define a Markov transition kernel such that for a non-terminal state $s$ and an action $a$:

   (a) If box $i$ is closed, that is, $s_i = \boxtimes$, and $a$ corresponds to opening this closed box, then the MDP transitions to a new state $s'$, where $s'_i$ represents a random draw of the value in box $i$ and all other components $s'_j$ for $j \neq i$ remains the same as $s_j$;

   (b) If box $i$ is open, that is, $s_i \in \mathbb{R}$, and $a$ corresponds to taking the reward $s_i$ from this open box, then the MDP deterministically transitions to a terminal state $s' = s_i$.

This defines a class of time-homogeneous undiscounted Markov decision processes, parameterized by the reward distribution $f$, of bounded expected value, and cost function $c$. We consider Markov policies, which are probability kernels mapping states to probability measures over actions. For this class of MDPs, all such policies are guaranteed to terminate in finite time, because at most one can open all the boxes before being forced to select one and thereby enter a terminal state. As consequence, by standard MDP theory:

1. The value function $V^{(\pi,c)} : \mathcal{S} \to \mathbb{R}$ is well-defined, where the superscripts denote the policy and the cost with respect to which the MDP is defined.

2. The optimal value function $V^{(*,c)} : \mathcal{S} \to \mathbb{R}$ is also well-defined.

3. There exists an optimal policy $\pi^{(*,c)}$ which achieves the optimal value $V^{(*,c)}$.

4. The map $\mathbb{R}^{|X|}_+ \to \mathbb{R}$ defined by $c \mapsto V^{(\pi,c)}$ is affine for all $\pi$.

5. The map $\mathbb{R}^{|X|}_+ \to \mathbb{R}$ defined by $c \mapsto V^{(*,c)}$ is convex, since it is a supremum of affine functions.

We immediately note that this formulation extends to cover two variations of interest:

1. *Stochastic costs.* One can handle this by replacing $c$ with a probability kernel. In this case, letting $m$ be the mean costs, we have $V^{(\pi,c)} = V^{(\pi,m)}$. Using this, we henceforth work with deterministic costs without loss of generality.

2. *Simple regret as a terminal reward.* One can also consider an MDP with a stochastic terminal reward which subtracts the best-in-hindsight term $\sup_{x \in X} f(x)$ from the terminal rewards $\partial r$ given above—this gives an objective equal to simple regret up to a minus sign. Since this only changes the rewards up to a random constant which is independent of the chosen actions, by linearity of expectation, the value function of this modified MDP is equal to those of our MDP up to a constant. We therefore omit this term without loss of generality.

The expected budget-constrained setting does not directly incorporate costs into the MDP itself: more precisely, it takes $c(s,a) = 0$ within the MDP formulation, which means the value function of this modified MDP is $V^{(\pi,0)}$. Instead, costs are incorporated as a constraint set on the policy class one considers. With this notation, the values of the optimization problems for Bayesian-optimal policy in the expected budget-constrained and cost-per-sample problems are

$$V_{\mathrm{ebc}}^{(*,c)} = \sup_{\pi \in \Pi_B^{(c)}} V^{(\pi,0)} \qquad\qquad V^{(*,c)} = \sup_{\pi \in \Pi} V^{(\pi,c)} \qquad (27)$$

where $\Pi$ is the set of all Markov policies, and the feasible set for the expected budget-constrained problem is

$$\Pi_B^{(c)} = \left\{ \pi \in \Pi : \mathbb{E} \sum_{t=1}^{T^{(\pi)}} c(x_t) \leq B \right\} \qquad (28)$$

and $T^{(\pi)}$ is policy $\pi$'s stopping time, that is

$$T^{(\pi)} = \inf\{t \geq 1 : s_t \in \partial S\} \qquad (29)$$

which is bounded above by $|X| < \infty$. This defines the optimization problems under study. Define the set of maximizers

$$\Pi^{(*,c)} = \left\{ \pi \in \Pi : V^{(\pi,c)} = V^{(*,c)} \right\} \qquad (30)$$

which is non-empty, since, as said above, by MDP theory the supremum defining $V^{(*,c)}$ is achieved. Define also the set of feasible policies which satisfy the constraints in a tight manner, namely

$$\Pi_{B,\mathrm{eq}}^{(c)} = \left\{ \pi \in \Pi_B^{(c)} : \mathbb{E} \sum_{t=1}^{T^{(\pi)}} c(x_t) = B \right\}. \qquad (31)$$

We are now ready to prove the claim in question. For this, we employ a Lagrangian duality argument: loosely speaking, this will reveal the cost-per-sample $\lambda$ to be the Lagrange multipliers associated with the expected budget constraint. We prove the main claim via a series of lemmas, starting from handling the Lagrange-multiplier-part of the argument.

**Lemma 5.** *Define the function*

$$\mathcal{A} : [0,\infty) \to \mathbb{R} \qquad\qquad \mathcal{A}(\lambda) = V^{(*,\lambda c)} + \lambda B. \qquad (32)$$

*Suppose that the infimum of $\mathcal{A}$ is achieved, and denote it by $\lambda_B^* \in [0,\infty)$. Suppose further that there exists an optimal policy for the cost-per-sample MDP for which the expected budget constraint is tight, namely $\widehat{\pi} \in \Pi^{(*,\lambda_B^* c)} \cap \Pi_{B,\mathrm{eq}}^{(c)}$. Then we have*

$$V_{\mathrm{ebc}}^{(\widehat{\pi},c)} = V_{\mathrm{ebc}}^{(*,c)}. \qquad (33)$$

*Proof.* We start by expressing the expected budget-constrained optimization problem in an unconstrained form via Lagrange multipliers, obtaining

$$V_{\mathrm{ebc}}^{(*,c)} = \sup_{\pi \in \Pi_B^{(c)}} V^{(\pi,0)} = \sup_{\pi \in \Pi} \inf_{\lambda \geq 0} \left( V^{(\pi,0)} - \lambda \left( \mathbb{E} \sum_{t=1}^{T^{(\pi)}} c(x_t) - B \right) \right) \qquad (34)$$

$$= \inf_{\lambda \geq 0} \sup_{\pi \in \Pi} \left( V^{(\pi,0)} - \lambda \left( \mathbb{E} \sum_{t=1}^{T^{(\pi)}} c(x_t) - B \right) \right) \qquad (35)$$

$$= \inf_{\lambda \geq 0} \left( V^{(*,\lambda c)} + \lambda B \right) = \inf_{\lambda \geq 0} \mathcal{A}(\lambda) \qquad (36)$$

where the second line follows from the Lemma 10 since the respective Lagrangian equals $V^{(\pi,\lambda c)}$ up to a constant, and the third line follows by definition of $V^{(*,\lambda c)}$ is by definition the terminal reward minus the cumulative costs.

Now, suppose the infimum of $\mathcal{A}$ is achieved, and let $\lambda_B^* \in [0, \infty)$ be any minimizer. Then $V_{\text{ebc}}^{(*,\lambda_B^* c)} = V^{(*,\lambda_B^* c)} + \lambda_B^* B$. Using this, and the fact that the policy $\widehat{\pi}$ by assumption achieves the supremum over $\sup_{\pi \in \Pi} V^{(\pi, \lambda_B^* c)}$ and satisfies $\widehat{\pi} \in \Pi_B^{(c)}$, we get

$$\sup_{\pi \in \Pi_B^{(c)}} V^{(\pi, 0)} = \sup_{\pi \in \Pi} V^{(\pi, \lambda_B^* c)} + \lambda_B^* B = \sup_{\pi \in \Pi_B^{(c)}} V^{(\pi, \lambda_B^* c)} + \lambda_B^* B. \tag{37}$$

Thus, the optimization objectives, defining the expected budget-constrained problem and the cost-per-sample problem with costs $\lambda_B^* c$, are equal up to a constant. Therefore, their minimizer sets coincide, and the claim follows. $\qquad\square$

Lemma 5 reveals that as long as the optimization problems arising from Lagrange multipliers are achieved, and the resulting policy $\widehat{\pi}$ is feasible, the expected budget-constrained problem will admit the same optimum as its associated cost-per-sample problem. By MDP theory, we know the supremum over $\Pi$ is achieved: we now show the infimum involving $\lambda$ is achieved as well, which essentially amounts to ruling out arbitrarily large $\lambda$ values.

**Lemma 6.** *The map $\mathcal{A}$ is convex. Moreover, the infimum $\inf_{\lambda \geq 0} \mathcal{A}(\lambda)$ is achieved.*

*Proof.* First, note that convexity follows straightforwardly from convexity of $c \mapsto V^{(*,c)}$ and the fact that the sum of convex functions is convex. Next, since the feasible set is $\lambda \in [0, \infty)$ and the objective is convex, either the infimum is achieved, or the objective is non-increasing. We prove the latter property cannot hold. For this, first consider the policy $\pi'$ that deterministically opens some box $x' \in X$ whose cost is $c(x') < B$—note that our assumptions guarantee the existence of at least one such box—and selects the value in it. Therefore if we define $\mathcal{A}'(\lambda) = V^{(\pi', \lambda c)} + \lambda B$, we have

$$\mathcal{A}'(\lambda) = V^{(\pi', \lambda c)} + \lambda B \leq V^{(*, \lambda c)} + \lambda B = \mathcal{A}(\lambda). \tag{38}$$

At the same time, we have $V^{(\pi', \lambda c)} = \mathbb{E}\, f(x') - \lambda c(x')$ therefore

$$\mathcal{A}'(\lambda) = \mathbb{E}\, f(x') - \lambda \underbrace{(c(x') - B)}_{\text{negative}} \tag{39}$$

which means the map $\mathcal{A}'$ is affine and strictly increasing with respect to $\lambda$. Thus, the map $\mathcal{A}$ is lower-bounded by a strictly increasing affine function, and therefore cannot be non-increasing. We conclude that the infimum is achieved. $\qquad\square$

The next part is to show that the optimal cost-per-sample policy is feasible for the expected budget-constrained problem. Then to do so, we need to verify a certain convergence criterion, given below. In the following, note that $\lambda'$—which can be negative—is never used in the definition of any optimal policy, only to compute the value of a given policy, which still makes sense even with negative costs.

**Lemma 7.** *For any monotone sequence $\lambda_n \to \lambda$ where $\lambda_n > 0$ and $\lambda > 0$, and any $\lambda' \in \mathbb{R}$, there exist policies $\pi_n^* \in \Pi^{(*,\lambda_n c)}$ and $\pi^* \in \Pi^{(*,\lambda c)}$ such that*

$$V^{(\pi_n^*, \lambda' c)} \to V^{(\pi^*, \lambda' c)}. \tag{40}$$

*Proof.* First, note that

$$V^{(\pi_n^*, \lambda' c)} = \underbrace{V^{(\pi_n^*, \lambda c)}}_{(a)} + (\lambda' - \lambda_n) \underbrace{\mathbb{E} \sum_{t=1}^{T^{(\pi_n^*)}} c(x_t)}_{(b)}. \tag{41}$$

Using this, by passing limits through the respective sums and products, it suffices to prove convergence of (a) and (b) separately, with the same sequence choice $\pi_n^*$ in both cases. By Weitzman [44]—see Kleinberg et al. [25], Theorem 1, for an alternative proof—any policies $\pi_n^*, \pi^*$ which maximize the respective Gittins indices $\alpha_n^*, \alpha^*$ are optimal: we will therefore choose $\pi_n^*, \pi^*$ from this set of policies. This choice is not unique due to the possibility of ties, both in terms of which boxes to open, and when to stop: we will show that tie-breaking choices do not affect convergence of (a), and will make a suitable choice of tie-breaking rules, depending on the sequence $\lambda_n$ in the claim's assumptions, to prove convergence of (b).

*Part I: convergence of (a).* Define $\kappa_n(x) = \min(f(x), \alpha_n^*(x))$, and define $\kappa$ analogously. By Kleinberg et al. [25], Theorem 1, we have

$$V^{(\pi^*, \lambda_n c)} = \mathbb{E} \max_{x \in X} \kappa_n(x) \tag{42}$$

and analogously for $\lambda$, $\alpha^*$, and $\kappa$. We now argue that $\alpha_n^* \to \alpha^*$ monotonically pointwise. Since $\lambda_n \to \lambda$ converges monotonically, consider

$$\mathrm{EI}_f(x, g) = \lambda_n c(x). \tag{43}$$

Recall that $\mathrm{EI}_f(x, g)$ is continuous and strictly decreasing in $g$: hence, its inverse in $g$ exists and is also strictly decreasing and continuous. We conclude that $\alpha_n^* \to \alpha^*$ monotonically pointwise. From this, convergence of the respective expectations $\mathbb{E} \max_{x \in X} \kappa_n(x)$, and therefore convergence of (a), follows by the Monotone Convergence Theorem.

*Part II: convergence of (b).* We will need an identity involving the order in which boxes are opened: for this, we first prove that once a tie-breaking rule is chosen, for large enough $n$, $\pi_n^*$ and $\pi^*$ open boxes in the same order. Since the number of boxes is finite, and $\alpha_n^* \to \alpha^*$ monotonically pointwise from (a): it follows for $n$ large enough that, if there are no ties in $\alpha^*(x)$, then $\pi_n^*$ opens boxes in the same order as $\pi^*$. If there are ties in $\alpha^*$, we choose a tie-breaking rule so that $\pi_n^*$ and $\pi^*$ open boxes in the same order. It follows that, once this choice is made, for $n$ large enough, all policies open boxes in the same order $x_1, .., x_{|X|}$, with different $\pi_n$ and $\pi^*$ possibly stopping at different times.

The identity we seek will differ depending on how tie-breaking rules regarding when to stop are handled: recall that ties can occur when opening the box $x_t \in X$ at time $t$, we have $\alpha^*(x_t) = f_t^*$, where $f_t^*$ is the best observed value up to time $t$. We adopt the following tie-breaking rule for stopping, depending on whether or not $\lambda_n$ is an increasing or decreasing sequence:

- If $\lambda_n$ is increasing: let $\pi^* = \pi_+^*$ be the policy that opens the best closed box in the event of a tie.

- If $\lambda_n$ is decreasing: let $\pi^* = \pi_-^*$ be the policy that stops in the event of a tie.

Note also that if $\lambda_n$ is both increasing and decreasing, it is constant, and the claim we want to prove holds: therefore, we suppose it is not, which makes the choice between $\pi_+^*$ and $\pi_-^*$ uniquely determined. We make the same tie-breaking choices for $\pi_n^*$, letting $\pi_{n,+}^*$ and $\pi_{n,-}^*$ be the respective policies. Let $x_1, .., x_{|X|}$ denote boxes in the order they are opened. Then the expected total costs are

$$\mathbb{E} \sum_{t=1}^{T^{(\pi_+^*)}} c(x_t) \stackrel{\text{(i)}}{=} \sum_{i=1}^{|X|} \mathbb{P}(T^{(\pi_+^*)} \geq i) c(x_i) \stackrel{\text{(ii)}}{=} \sum_{i=1}^{|X|} \mathbb{P}(f_j^* \leq \alpha^*(x_j), 1 \leq j \leq i) c(x_i) \tag{44}$$

$$\stackrel{\text{(iii)}}{=} \sum_{i=1}^{|X|} \mathbb{P}(f(x_j) \leq \alpha^*(x_i), 1 \leq j \leq i) c(x_i) \stackrel{\text{(iv)}}{=} \sum_{i=1}^{|X|} \prod_{j=1}^{i-1} \mathbb{P}(f(x_j) \leq \alpha^*(x_i)) c(x_i) \tag{45}$$

where (i) follows by writing the probability as an expectation of an appropriate indicator, (ii) follows by noting that under the policy $\pi_+^*$, the event $T^{(\pi_+^*)} \geq i$ occurs if and only if at each time $j = 1, .., i$, the best observed value $f_j^*$ is not higher than the Gittins index of $x_j$, (iii) follows by expanding the maximum which defines $f_j^*$ and using monotonicity of $\alpha^*(x_j)$ in $j$ to simplify the resulting events, and (iv) follows by independence of $f(x_j)$ and $f(x_{j'})$ for $j \neq j'$. Using this, it suffices to show

$$\mathbb{P}(f(x_j) \leq \alpha_n^*(x_i)) \to \mathbb{P}(f(x_j) \leq \alpha^*(x_i)) \qquad \text{if } \pi^* = \pi_+^*, \text{ and} \tag{46}$$
$$\mathbb{P}(f(x_j) < \alpha_n^*(x_i)) \to \mathbb{P}(f(x_j) < \alpha^*(x_i)) \qquad \text{if } \pi^* = \pi_-^*. \tag{47}$$

The former equals the cumulative distribution function of $f(x_j)$, which is right-continuous, evaluated at $\alpha_n^*(x_i)$. The latter is similar but is instead left-continuous. For $\pi_{n,+}^*$, since $\lambda_n$ is increasing, we have that $\alpha_n(x)$ is decreasing, and the claim follows by right-continuity. For $\pi_{n,-}^*$ since $\lambda_n$ is decreasing, we have that $\alpha_n(x)$ is increasing, and the claim follows by left-continuity. $\qquad\square$

We are now ready to prove the key property needed to apply Lemma 5.

**Lemma 8.** *If $\lambda_B^* > 0$, then there exists a $\widehat{\pi} \in \Pi^{(*, \lambda_B^* c)} \cap \Pi_{B,\mathrm{eq}}^{(c)}$.*

*Proof.* By the Envelope Theorem—specifically, Lemma 13—we have for $\lambda \in [0, \infty)$ that

$$\partial_v \mathcal{A}(\lambda) = \partial_v \left( V^{(*,\lambda c)} + \lambda B \right) \tag{48}$$

$$= \partial_v \left( \sup_{\pi \in \Pi} \left( V^{(\pi,0)} - \lambda \left( \mathbb{E} \sum_{t=1}^{T^{(\pi)}} c(x_t) - B \right) \right) \right) \tag{49}$$

$$= \max_{\pi' \in \Pi^{(*,\lambda c)}} \partial_v \left( V^{(\pi,0)} - \lambda \left( \mathbb{E} \sum_{t=1}^{T^{(\pi)}} c(x_t) - B \right) \right) \Bigg|_{\pi = \pi'} \tag{50}$$

$$= vB - \min_{\pi \in \Pi^{(*,\lambda c)}} v \mathbb{E} \sum_{t=1}^{T^{(\pi)}} c(x_t) \tag{51}$$

where all Gâteaux derivatives—see Definition 11—are taken with respect to $\lambda$ and exist by convexity, and the pointwise convergence condition of Lemma 13 follows by first noting that

$$V^{(\pi,0)} - \lambda \left( \mathbb{E} \sum_{t=1}^{T^{(\pi)}} c(x_t) - B \right) = V^{(\pi,\lambda c)} + \lambda B \tag{52}$$

and applying Lemma 7. By the first-order optimality conditions, we have

$$\partial_v \mathcal{A}(\lambda_B^*) \geq 0. \tag{53}$$

Combining this with the above, we conclude

$$\min_{\pi \in \Pi^{(*,\lambda c)}} v \mathbb{E} \sum_{t=1}^{T^{(\pi)}} c(x_t) \leq vB. \tag{54}$$

Since $\lambda_B^* > 0$, this expression holds for $v$ equal to $\pm 1$: plugging this in, we get

$$\min_{\pi \in \Pi^{(*,\lambda_B^* c)}} \mathbb{E} \sum_{t=1}^{T^{(\pi)}} c(x_t) \leq B \qquad B \leq \max_{\pi \in \Pi^{(*,\lambda_B^* c)}} \mathbb{E} \sum_{t=1}^{T^{(\pi)}} c(x_t). \tag{55}$$

Now, let $\widehat{\pi}_-$ be a policy from the minimizer set, and let $\widehat{\pi}_+$ be a policy from the maximizer set. Define a third policy $\widehat{\pi}_\alpha$ which randomizes between the two, choosing $\widehat{\pi}_-$ with probability $\alpha$ and $\widehat{\pi}_+$ with probability $1 - \alpha$. Since $\widehat{\pi}_+$ and $\widehat{\pi}_-$ are optimal, they achieve the same value, so by convexity $\widehat{\pi}_\alpha$ also achieves the same value: therefore, $\widehat{\pi}_\alpha \in \Pi^{(*,\lambda_B c)}$. On the other hand, the expected total costs of $\widehat{\pi}_\alpha$ are a convex combination of the expected costs of $\widehat{\pi}_-$ and $\widehat{\pi}_+$: since the former lower-bounds $B$ and the latter upper-bounds $B$, there exists an $\alpha \in [0, 1]$ for which the costs of $\pi_\alpha$ are exactly $B$. Taking $\widehat{\pi} = \widehat{\pi}_\alpha$ for this value of $\alpha$, we obtain $\widehat{\pi} \in \Pi_{B,\text{eq}}^{(c)}$ and the claim follows. $\square$

To complete the proof, we show the minimizer set of $\mathcal{A}$ does not contain zero.

**Lemma 9.** *Let $\lambda_B^* \in [0, \infty)$ be any minimizer of $\mathcal{A}$, then $\lambda_B^* > 0$.*

*Proof.* From the derivative calculation used in proof of Lemma 8, plugging in $v = 1$ into the respective Gâteaux derivative, we obtain

$$\frac{\mathrm{d}}{\mathrm{d}\lambda} \mathcal{A}(\lambda) = B - \min_{\pi \in \Pi^{(*,\lambda c)}} \mathbb{E} \sum_{t=1}^{T^{(\pi)}} c(x_t). \tag{56}$$

If $\lambda = 0$, then the costs are zero: thus, any policy that opens every box, then selects the highest reward among opened boxes, is optimal. Therefore, we have $T^{(\pi)} = |X|$, and every optimal policy's cost is equal to the total cost of all the boxes. This gives

$$\frac{\mathrm{d}}{\mathrm{d}\lambda} \mathcal{A}(0) = B - \sum_{x \in X} c(x) < 0 \tag{57}$$

where the inequality follows by the assumption that the expected budget constraint is active. The claim follows. $\square$

With these results at hand, we are now ready to prove the main claim.

**Theorem 2.** *Consider the expected budget-constrained problem, with the assumptions of Theorem 1. Assume the problem is feasible and the constraint is active, namely $\min_{x \in X} c(x) < B < \sum_{x \in X} c(x)$. Then there exists a $\lambda > 0$ and a tie-breaking rule such that the policy defined by maximizing the Gittins index acquisition function $\alpha^\star(\cdot)$, defined using costs $\lambda c(x)$, is Bayesian-optimal.*

*Proof.* Combine Lemma 5 with Lemmas 6, 8 and 9. □

We conclude by comparing this claim and proof with the results of Aminian et al. [3]:

1. The objective of their expected budget-constrained problem is more general: while we focus on minimizing simple regret, Aminian et al. [3] study how to maximize utility, which is defined in a broader context—one example being the difference between the best observed value and cumulative costs, which we study here. Our expected budget constraint is similarly a special case, which, in their terminology, takes all weights on indicators denoting inspection to be identical, and takes all selection weights to be zero.

2. We do not assume that $f(x)$ has finite support for all $x$. This results in a significant technical difference: we must apply a sharp envelope theorem with an explicit supremum to conclude that the expected budget constraint is satisfied. More-straightforward results such as those presented by Milgrom and Segal [30], or their subdifferential-formulated analogues as found in the proof of Aminian et al. [3], Proposition 1 (iii), would not suffice: due to an explicit counterexample involving upside-down-absolute-value functions, without suitable structure, one cannot conclude that the supremum is achieved and the resulting inequality is tight. As an alternative to our approach, one could instead appeal to finiteness of the class of deterministic policies—which Aminian et al. [3] assume, leading to piecewise-linear value functions. We avoid these assumptions, leading to a more technical but more general argument.

We conclude by noting that one can also consider variants of the problems we have studied under an almost-sure budget constraint, for instance

$$\Pi_{B,\text{as}}^{(c)} = \left\{ \pi \in \Pi : \mathbb{P}\left( \sum_{t=1}^{T^{(\pi)}} c(x_t) \leq B \right) = 1 \right\} \qquad V_{\text{asbc}}^{(*,c)} = \sup_{\pi \in \Pi_{B,\text{as}}^{(c)}} V^{(\pi,0)}. \qquad (58)$$

For this problem, this optimal value is upper-bounded by the optimal value of the expected budget-constrained optimization—specifically, $V_{\text{asbc}}^{(*,c)} \leq V_{\text{ebc}}^{(*,c)}$, since $\Pi_{B,\text{as}}^{(c)} \subseteq \Pi_B^{(c)}$.

### B.6 Auxillary results from optimization theory

Below we state and prove three results from optimization theory: a Lagrange Multiplier Theorem for optimization problems with inequality constraints, and two Envelope Theorems. These results are not new, but are difficult to find at the level of generality we need them at: most references begin by assuming the domain of optimization is $\mathbb{R}^d$, or, if not that, that it is a Banach space, whereas we need results that hold when the domain of optimization is an arbitrary set, without a linear structure or a topology. In light of this, we have found it easier to simply prove the claims we need.

**Lemma 10.** *Let $X$ be an arbitrary set, let $f : X \to \mathbb{R}$ and let $g : X \to \mathbb{R}$. Then defining $\mathcal{L}(x, \lambda) = f(x) - \lambda(g(x) - y)$ for $\lambda \geq 0$, we have*

$$\sup_{\substack{x \in X \\ g(x) \leq y}} f(x) = \sup_{x \in X} \inf_{\lambda \geq 0} \mathcal{L}(x, \lambda). \qquad (59)$$

*Moreover, suppose there exist $x^*, \lambda^*$ which satisfy $\mathcal{L}(x^*, \lambda^*) = \inf_{\lambda \geq 0} \sup_{x \in X} \mathcal{L}(x, \lambda)$ and $g(x^*) = y$. Then we have*

$$\sup_{x \in X} \inf_{\lambda \geq 0} \mathcal{L}(x, \lambda) = \inf_{\lambda \geq 0} \sup_{x \in X} \mathcal{L}(x, \lambda). \qquad (60)$$

*Proof.* First, we show that for all $x$, we have

$$\inf_{\lambda \geq 0} \mathcal{L}(x, \lambda) = \begin{cases} f(x) & g(x) \leq y \\ -\infty & g(x) > y. \end{cases} \qquad (61)$$

To show this, suppose first that $g(x) \leq y$. Then $\lambda(g(x) - y) \leq 0$, hence its negation is non-negative, and minimized at $\lambda = 0$. Now, suppose the converse. Then $\lambda(g(x) - y) > 0$, so its negation is negative, and the objective can be made arbitrarily close to $-\infty$ by scaling $\lambda$. Using this, write

$$\sup_{x \in X} \inf_{\lambda \geq 0} \mathcal{L}(x, \lambda) = \max \left( \sup_{\substack{x \in X \\ g(x) \leq y}} \inf_{\lambda \geq 0} \mathcal{L}(x, \lambda), \ \sup_{\substack{x \in X \\ g(x) > y}} \inf_{\lambda \geq 0} \mathcal{L}(x, \lambda) \right) \tag{62}$$

$$= \max \left( \sup_{\substack{x \in X \\ g(x) \leq y}} f(x), -\infty \right) = \sup_{\substack{x \in X \\ g(x) \leq y}} f(x). \tag{63}$$

We now argue that, under the claim's additional assumption, one can swap the order of the supremum and infimum. Let $x^*, \lambda^*$ be a pair for which the constraints are tight. Then

$$\inf_{\lambda \geq 0} \sup_{\substack{x \in X \\ g(x) \leq y}} \mathcal{L}(x, \lambda) \overset{(i)}{=} \sup_{\substack{x \in X \\ g(x) \leq y}} \mathcal{L}(x, \lambda^*) \overset{(ii)}{=} \sup_{\substack{x \in X \\ g(x) = y}} \mathcal{L}(x, \lambda^*) \tag{64}$$

$$\overset{(iii)}{=} \sup_{\substack{x \in X \\ g(x) = y}} \inf_{\lambda \geq 0} \mathcal{L}(x, \lambda) \overset{(iv)}{\leq} \sup_{\substack{x \in X \\ g(x) \leq y}} \inf_{\lambda \geq 0} \mathcal{L}(x, \lambda) \tag{65}$$

where (i) follows by definition of $\lambda^*$, (ii) follows by the fact that $x^*$ achieves the supremum and satisfies $g(x^*) = y$, (iii) follows by the fact that, when restricted to the set $\{x \in X : g(x) = y\}$, $\mathcal{L}(x, \lambda)$ is constant in $\lambda$ for all $x$-values, hence the infimum is taken over constant functions, (iv) follows by making the feasible set larger. Combining this with the sup-inf inequality [10]

$$\sup_{\substack{x \in X \\ g(x) \leq y}} \inf_{\lambda \geq 0} \mathcal{L}(x, \lambda) \leq \inf_{\lambda \geq 0} \sup_{\substack{x \in X \\ g(x) \leq y}} \mathcal{L}(x, \lambda) \tag{66}$$

gives the claim. $\qquad \square$

It is easy to see that this argument holds even if one lets $y$ take values in an infinite-dimensional vector space, as long as $\lambda$ takes values within a convex cone which is suitably paired with the aforementioned vector space, but we will not need this level of generality. The arguments we employ will be cleanest if we work with directional derivatives in the sense of Gâteaux, as opposed to left-derivatives and right-derivatives of real-valued functions, which in our setting are equivalent but require more management of minus signs. For this, we adopt the following notation.

**Definition 11.** *Let $\mathcal{X}$ be a topological vector space, let $X \subseteq \mathcal{X}$, and let $f : X \to \mathbb{R}$ be a function. The GÂTEAUX DERIVATIVE $\partial_v f(x) \in \mathbb{R}$ of a function $f$ at a point $x \in X$ in the direction $v \in \mathcal{X}$, if it exists, is defined as*

$$\partial_v f(x) = \lim_{\varepsilon \to 0^+} \frac{f(x + \varepsilon v) - f(x)}{\varepsilon}. \tag{67}$$

Note that we require no properties of our Gâteaux derivatives: in particular, $\partial_v f(x)$ can be non-linear in $v$, though one can easily see that it will always satisfy $\partial_{\alpha v} f(x) = \alpha \partial_v f(x)$ for $\alpha \geq 0$. If $f$ is a convex function, one can show by monotonicity of finite differences that $\partial_v f(x)$ always exists—in the non-extended-valued sense defined above—on the relative interior of the effective domain of $f$.

Our arguments need an appropriate Envelope Theorem. The first statement we need is essentially equivalent to Milgrom and Segal [30], Theorem 1, which is formulated in the language of left-derivatives and right-derivatives: Milgrom and Segal [30] state in a footnote that their claim also holds in a general normed vector space, provided one works with directional differentiation. In fact, the claim is even more general than that, and does not require a norm. Since we find this variant to be particularly clean, elegant, and instructive, we prove the claim in full generality below.

**Lemma 12.** *Let $\Theta$ be a subset of a topological vector space, and let $X$ be an arbitrary set. Let $f : X \times \Theta \to \mathbb{R}$ be bounded above in its first argument, and define $\mathcal{V}$ to be*

$$\mathcal{V}(\theta) = \sup_{x \in X} f(x, \theta) \tag{68}$$

*Suppose that, for every $\theta \in \Theta$, the supremum is achieved, and let $X^*(\theta)$ be the maximizer set. For any $v$, suppose that the Gâteaux derivatives $\partial_v \mathcal{V}(\theta)$ and $\partial_v f(x, \theta)$ exist and are finite-valued, with the convention that the Gâteaux derivative of $f$ is taken in its second argument. Then*

$$\sup_{x^* \in X^*(\theta)} \partial_v f(x^*, \theta) \leq \partial_v \mathcal{V}(\theta). \tag{69}$$

*Proof.* Fix $\theta \in \Theta$, and let $x^*(\theta) \in X^*(\theta)$ be an arbitrary maximizer. Note that, for all $\theta' \in \Theta$ and all $x^*(\theta') \in X^*(\theta')$, by optimality we have

$$f(x^*(\theta), \theta') \leq f(x^*(\theta'), \theta') = \mathcal{V}(\theta') \tag{70}$$

with equality for $\theta' = \theta$. Letting $\varepsilon > 0$ be sufficiently small, taking $\theta' = \theta + \varepsilon v$, subtracting $f(x^*(\theta), \theta) = \mathcal{V}(\theta)$ from both sides, and dividing by $\varepsilon$, we get

$$\frac{f(x^*(\theta), \theta + \varepsilon v) - f(x^*(\theta), \theta)}{\varepsilon} \leq \frac{\mathcal{V}(\theta + \varepsilon v) - \mathcal{V}(\theta)}{\varepsilon} \tag{71}$$

thus taking limits as $\varepsilon \to 0$ gives

$$\partial_v f(x^*(\theta), \theta) \leq \partial_v \mathcal{V}(\theta). \tag{72}$$

This holds for all choices $x^*(\theta) \in X^*(\theta)$, and the claim follows. $\qquad \square$

For our arguments to go through, we need a sharper version of this result, with the inequality replaced with an equality. Results like this go back at least to Danskin [12], and are proven by Bonnans and Shapiro [9], Proposition 4.12, Başar and Bernhard [6], Theorem 10.1, and Bertsekas [8]: however, all of their claims require topological assumptions on the domain of optimization. Moreover, it is easy to see—for instance by considering an envelope made up of upside-down absolute value functions, where $\mathcal{V}(\theta)$ is constant but $f(x, \theta)$ is not—that the inequality cannot be tight without similar assumptions. Below, we show that being affine in $\theta$ and a certain convergence criterion suffice, even if $X$ is an arbitrary set. The argument is similar to that of the aforementioned references.

**Lemma 13.** *With the assumptions and notations in Lemma 12, suppose further that $f(x, \theta)$ is affine in $\theta$ for all $x$, and that for any monotone $\theta_n \to \theta$ there exist $x^*(\theta_n) \in X^*(\theta_n)$ and $x^*(\theta) \in X^*(\theta)$ such that $f(x^*(\theta_n), v) \to f(x^*(\theta), v)$. Then we have*

$$\sup_{x^* \in X^*(\theta)} \partial_v f(x^*, \theta) = \max_{x^* \in X^*(\theta)} \partial_v f(x^*, \theta) = \partial_v f(x^*(\theta), \theta) = \partial_v \mathcal{V}(\theta). \tag{73}$$

*Proof.* For any $\varepsilon$, note that

$$\frac{\mathcal{V}(\theta + \varepsilon v) - \mathcal{V}(\theta)}{\varepsilon} = \frac{f(x^*(\theta + \varepsilon v), \theta + \varepsilon v) - f(x^*(\theta), \theta)}{\varepsilon} \tag{74}$$

$$= \frac{f(x^*(\theta + \varepsilon v), \theta + \varepsilon v) - f(x^*(\theta + \varepsilon v), \theta)}{\varepsilon} \tag{75}$$

$$+ \frac{\overbrace{f(x^*(\theta + \varepsilon v), \theta) - f(x^*(\theta), \theta)}^{\text{negative}}}{\varepsilon} \tag{76}$$

$$\leq \frac{f(x^*(\theta + \varepsilon v), \theta + \varepsilon v) - f(x^*(\theta + \varepsilon v), \theta)}{\varepsilon} \tag{77}$$

$$= f(x^*(\theta + \varepsilon v), v) \tag{78}$$

where the respective term is negative because $f(x^*(\theta + \varepsilon v), \theta) \leq f(x^*(\theta), \theta)$, which holds by optimality of $x^*(\theta)$. Taking limits gives

$$\partial_v \mathcal{V}(\theta) \leq f(x^*(\theta), v) = \partial_v f(x^*(\theta), \theta) \tag{79}$$

where the final equality follows from the expression for the Gâteaux derivative of a linear function. Combining this expression with the Lemma 12 shows that the respective supremum is achieved, and gives the claim. $\qquad \square$

# C   Experimental setup

We implement all experiments in BOTORCH [5]. Following standard practice, we initialize each optimization algorithm with $2(d+1)$ values drawn using a quasirandom Sobol sequence, where $d$ is the dimension of the domain. All computations were run on CPU, with individual experiments ran in parallel on various nodes of the Cornell G2 cluster, each allocated up to 4GB of memory, except for certain exceptions. These exceptions include KG, MSEI, and BMSEI for higher dimensions, which required substantially more memory, up to 32GB. Most individual runs took several minutes at most, with exception of the more-expensive KG, MSEI and BMSEI baselines: more information with a direct runtime comparison is given in Appendix D.2.

**Gaussian process models.**    In the Bayesian regret experiments shown in Figure 4, we use Matérn kernels with identical fixed hyperparameters—namely smoothness $5/2$ and length scale $10^{-1}$—for both the Gaussian process prior used to sample the objective function, and the Gaussian process used for Bayesian optimization. To maintain consistency, we do not standardize data in this variant. For the synthetic and empirical experiments shown in Figure 5 and Figure 6, we use Matérn kernels with smoothness $5/2$ and length scales learned from data via maximum marginal likelihood optimization, and standardize input variables to be in $[0, 1]^d$ along with output variables to be zero-mean and unit-variance, following BoTorch defaults. In the unknown-cost experiments, we model the objective and the logarithm of the cost function using independent Gaussian processes. Additional Bayesian regret experiment results, showing the effect of varying the kernel smoothness and length scale hyperparameters, are provided in Appendix D.6.

**Acquisition function optimization.**    This is done as follows. We begin by computing acquisition values at $200d$ points spread across the domain $X$, where $d$ is the dimension of $X$. For all acquisition functions except MSEI and BMSEI, which use a modification described below, the initial $200d$ points are generated using a Sobol sequence design. From these, $10d$ points are selected according to the initialization heuristic used by BoTorch, detailed in Balandat et al. [5], Appendix F.1. We then use multi-start L-BFGS-B to optimize the acquisition function from each selected point. The point with highest acquisition value among the $10d$ optimized points is chosen as the next evaluation point.

We now detail the modified strategy used for MSEI and BMSEI: here, the initial $200d$ points are selected using the warm-start initialization strategy described in Jiang et al. [24], Appendix D and Astudillo et al. [4], Appendix F. This strategy uses the optimal solution from the previous iteration, targeting the branch that originates from the tree's root and whose fantasy sample most closely matches the actual observed value of the previously suggested candidate on the true function. This modification favors MSEI and BMSEI, slightly disadvantaging PBGI and other baselines.

**Acquisition function hyperparameters.**    For PBGI, we choose the constant hyperparameter to be $\lambda = 10^{-4}$. For PBGI-D, we choose the initial value to be $\lambda_0 = 10^{-1}$ and the constant decay factor $\beta = 2$. For both variants, we compute the Gittins indices using 100 iterations of bisection search without any early stopping or other performance and reliability optimizations.

For LogEICC, we follow Lee et al. [27] to use $\alpha_t = \frac{B-t}{B}$ as the cost exponent where $B$ is set to be the cumulative cost cap shown on our plots. For UCB, we follow the schedule of Srinivas et al. [39] given by $\eta_t = 2\log(dt^2\pi^2/6\delta)$, where $d$ is the dimension. We also adopt the choice of $\delta = 10^{-1}$ and a scale-down factor of 5, as used in that work's experiments. For MSEI and BMSEI, we use 4 lookahead steps, each with a batch size of 1 and a single fantasy point.

**Omitted baselines.**    We omit MSEI from the Bayesian regret plots for $d = 16$ with $\kappa = 10^{-1}$ and for $d = 32$ with all length scale choices because we were unable to get it to work reliably in these settings: the implementation of Jiang et al. [24] results in frequent crashes due to running out of memory and related issues when used on higher-difficulty problems. We also omit KG from $d = 32$, BMSEI from the cost-aware Pest Control and Robot Pushing experiments, and MES from the Bayesian regret experiment with $d = 32$, for the same reasons.

In addition to the baselines mentioned in Section 4, we also implemented the *predictive entropy search (PES)* acquisition function of Hernández-Lobato et al. [23], but could not get its computations to run reliably in an automatic-differentiation-based environment without resulting in NaN gradients. Hernández-Lobato et al. [23] document this behavior, and suggest using finite-differencing in situa-

tions where it occurs: however, from initial examination, we found this to decrease performance on higher-dimensional problems. We therefore opted to restrict ourselves to automatically-differentiable baselines and omit PES, to ensure that performance differences seen can reliably be attributed to the acquisition functions used, and not to gradient computation.

**Objective functions: Bayesian regret.** For Bayesian regret, this is straightforward: the objective is simply a draw from a Fourier feature approximation of the respective Gaussian process prior, drawn in such a way that different baselines with the same random number seed share the same objective, but objectives for different seeds are different draws from the same prior. We use a total of 1024 Fourier features.

**Objective functions: synthetic benchmarks.** The synthetic benchmark functions we use are as follows. We use variants with dimension $d = 4, 8, 16$ in our experiments.

*Ackley*: this is

$$f_A(x_1, .., x_d) = 20 - 20 \exp\left(-0.2\sqrt{\frac{1}{d}\sum_{i=1}^{d} x_i^2}\right) - \exp\left(\frac{1}{d}\sum_{i=1}^{d} \cos(2\pi x_i)\right) + e \quad (80)$$

with search domain $X = [-1, 1]^d$.

*Levy*: this is $f_L = 100 \widehat{f}_L$ where

$$\widehat{f}_L(x_1, .., x_d) = s_1 + \sum_{i=1}^{d-1} (w_i - 1)^2 \left(1 + 10\sin(\pi w_i + 1)^2\right) + (w_d - 1)^2 \left(1 + \sin(2\pi w_d)^2\right) \quad (81)$$

with $w_i = 1 + \frac{x_i - 1}{4}$ and $s_1 = \sin(\pi w_1)^2$, and search domain $X = [-10, 10]^d$.

*Rosenbrock*: this is $f_R = 10^5 \widehat{f}_R$ where

$$\widehat{f}_R(x_1, .., x_d) = \sum_{i=1}^{d-1} \left(100(x_{i+1} - x_i^2)^2 + (x_i - 1)^2\right) \quad (82)$$

with search domain $X = [-5, 10]^d$.

**Cost function.** In the cost-aware Bayesian regret and synthetic benchmark experiments, we use the cost function

$$c(x) = 20\|S(x)\|_1 + 1 \quad (83)$$

where $S$ is an affine map used to standardize the input domain: specifically, $S(x) = Ax + b$ where $A$ is a diagonal matrix and $b$ is a vector, both chosen so that the image of $X$ under $S$ is $[0, 1]^d$.

**Objective functions: empirical.** We now detail the empirical objective functions.

*Pest Control (d = 25).* The pest control problem, as described in Oh et al. [31] aims to minimize the spread of pests as well as the costs of prevention treatment. We adopt the experiment setup from Li et al. [28], framing this as a categorical optimization problem with 25 variables, each representing a stage of intervention with 5 values reflecting different treatments. The objective function combines the spread of pests and the costs of prevention.

In our cost-aware experiment, we use the cost of prevention as the cost function. This can be computed in an automatically-differentiable manner, thus this problem is a known-cost problem.

*Lunar Lander (d = 12).* Following the setup in Eriksson et al. [15], we consider a reinforcement learning problem optimizing a controller for the lunar lander as implemented in OpenAI Gym, which includes 12 continuous input dimensions for engine throttle adjustments. The state space captures the lander's position, angle, time derivatives, and leg contact status. The controller's actions allow for directional booster firings or inaction. The objective is to maximize the average final reward over 50 randomly generated environments.

For cost-aware experimentation, we choose the cost to be the average number of simulation time steps, assuming batch processing in groups of 16. This assumption is based on the implementation

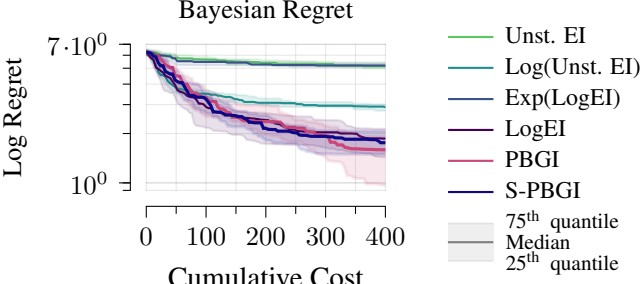

Figure 10: Experimental results comparing the impact of log-scaling and numerical stability on the performance of EI and PBGI, tested on Bayesian regret with $d = 16$ and shown using quartiles. We see that Exp(LogEI) is close to Unst. EI, and similarly for S-PBGI vs PBGI. However, Log(Unst. EI) improves significantly over EI, and LogEI improves even more.

found in the code associated with Li et al. [28]. Note that this objective involves the number of actual simulation steps used, and is therefore not automatically-differentiable: we thus consider this problem to be an unknown-cost problem.

*Robot Pushing ($d = 14$).* In this work, we adapt one of the three versions of the robot pushing problem designed by Wang and Jegelka [42], where two robots work to push two objects to their specified targets. The problem's complexity is captured through 14 control parameters, including each robot's initial placement and motion settings. The objective minimizes the sum of the distances from the final position of each object to its respective target.

In our cost-aware experiments, we test both known-cost and unknown-cost variants. The known-cost variant is the maximum of the two robots' operational duration representing the evaluation time, and the unknown-cost variant is the sum of their traversal distances representing the total energy use, similar to the cost function used for energy-aware robot pushing benchmark of Astudillo et al. [4], but with one modification: we use the distance traversed by the robot arms instead of the distance the objects are moved. To understand the effect of this difference, we include the results for both the unknown-cost and known-cost versions in Appendix D.4.

## D  Additional experimental results

Here, we provide additional experimental results to better understand performance differences and other aspects of policy behavior, including the effect of various problem hyperparameters.

### D.1  Impact of log-scaling and numerical stability on EI and PBGI

Ament et al. [2] introduce the LogEI variant of Expected Improvement (EI), which is equal to the logarithm of EI but incorporates improved numerical properties. Their work demonstrates that LogEI significantly outperforms EI in practice. In this appendix, we aim to investigate two key questions:

1. What specific properties of LogEI contribute to its improved performance over ordinary EI?

2. Can the stable numerical primitives proposed by Ament et al. [2] also benefit PBGI?

The LogEI acquisition function, denoted as $\alpha^{\text{LogEI}}(x)$, is defined as the logarithm of the EI acquisition function, $\alpha^{\text{EI}}(x)$, but is computed using a numerical helper function $\log\_h$ designed to provide better numerical stability. This introduces two potential sources of improvement for LogEI over EI: (a) the enhanced numerical stability due to the use of $\log\_h$, and (b) the effect of logarithmic scaling, which alters the gradients of the acquisition function and may lead to more effective optimization of $\alpha^{\text{LogEI}}(x)$. To disentangle these two factors, we compare not only LogEI and ordinary EI— the latter of which we refer to as *unstable EI (Unst. EI)* in this section for disambiguation—but also two additional variants: Log(Unst. EI), which directly applies a logarithmic transformation to unstable EI ($\log(\alpha^{\text{EI}}(x))$) without using $\log\_h$ or other numerical stability tricks, and Exp(LogEI) ($\exp(\alpha^{\text{LogEI}}(x))$), which applies the numerical helper function from LogEI, but then undoes the

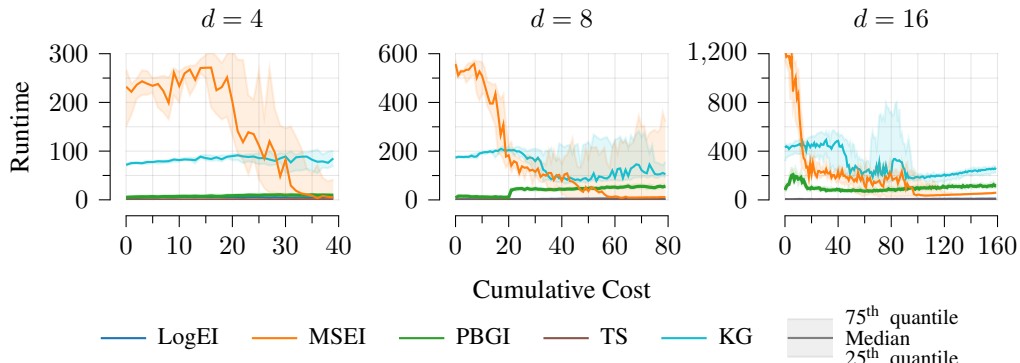

Figure 11: Runtime comparison of PBGI against baselines for computing the acquisition function on the Ackley benchmark across different dimensions ($d = 4, 8, 16$). We see that runtime of PBGI is slower than LogEI and TS, but faster than KG and MSEI.

logarithmic transformation. For Log(Unst. EI), we compute $\log(\alpha^{\mathrm{EI}}(x) + 10^{-6})$ to avoid infinite floating-point evaluations.

To test whether PBGI can benefit from the improved numerical stability, we also test PBGI and the following variant, which we call *stable PBGI (S-PBGI)*, which uses the following gradient formula:

$$\nabla_x \alpha^{\mathrm{S\text{-}PBGI}}(x) = \nabla_x \mu(x) + \frac{\nabla_x \sigma(x) - \sigma(x) \nabla_x \log c(x) - z \log\_\mathrm{h}'(z) \nabla_x \sigma(x)}{\log\_\mathrm{h}(z)} \qquad (84)$$

where $\log\_\mathrm{h}(z)$ is given by (9) of Ament et al. [2]. We do not consider logarithmic scaling of the PBGI acquisition function value because it can be negative.

From Figure 10 we see that the improved performance of LogEI over EI appears to be mostly attributable to the log-scaling, though the numerically stable computation provides significant benefit, too. In contrast, there appears to be little-to-no difference between PBGI and S-PBGI.

## D.2    Runtime comparison

Here, we provide a runtime comparison between PBGI with 100 bisection iterations and various baselines, including inexpensive baselines such as LogEI and TS, and expensive ones such as MSEI. We do so in the Ackley function setting, using the same hyperparameter settings as the main experiments. We do not perform any explicit caching in computing $\mu$ or $\sigma$ or their gradients, and simply rely on standard BoTorch routines to obtain these at each iteration. We measure the CPU time (in seconds) to compute and optimize the acquisition function.

Results are in Figure 11. We see that PBGI is slightly slower than LogEI and TS, but significantly faster than either KG or MSEI, though the runtime of the latter decreases substantially as it accumulates more data. Overall, we conclude that PBGI's runtime is closer to that of classical acquisition functions than sophisticated lookahead-based variants. If necessary, PBGI's runtime can potentially be reduced further by optimizing the computation of $\nabla \mu$ and $\nabla \sigma$, or using fewer bisection iterations.

## D.3    Hyperparameter choice for PBGI-D

Next, we examine the behavior of different hyperparameter choices for $\lambda_0$ and $\beta$ in PBGI-D. Recall that this variant sets $\lambda_t = \frac{\lambda_{t-1}}{\beta}$ at times when the Gittins stopping rule triggers, and $\lambda_t = \lambda_{t-1}$ at all other times. To understand this, we examine regret curves in the Bayesian regret setting of Section 4 with $d = 8$ under different choices. Results are in Figure 12. We see that behavior of different $\lambda_0$-values is qualitatively similar to that of different $\lambda$-values in PBGI, but with substantially smaller differences between variants, which are detectable primarily due to the relatively-large number of seeds in each experiment. Smaller values of $\lambda_0$ tend to lead to very slightly higher regret on small time scales, but very slightly lower regret on larger time scales. Behavior for $\beta$ is similar: larger values lead to a more gradual decay of $\lambda$, but with minimal overall impact on performance, especially in

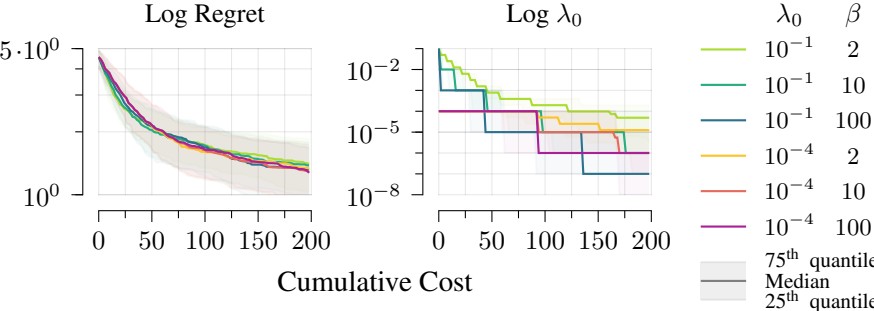

Figure 12: Behavior of PBGI-D under different choices of the initial value $\lambda_0$, and the coefficient of decay $\beta$, on the Bayesian regret experiment with $d = 8$ and other parameters set the same way as in Section 4. We show medians under $n = 256$ samples, along with quartiles to assess variability. We see that smaller $\lambda_0$-values act similarly to the behavior seen in Figure 3 for PBGI, trading off risk-seeking vs. risk-averse behavior, but with a substantially smaller gap between the two variants. Larger $\beta$-values lead to more-abrupt decay curves for $\lambda$, with an effect qualitatively similar to $\lambda_0$. In both cases, though our relatively-large sample size allows us to see small differences between methods, their overall impact on performance is close-to-nonexistent on the scale of variability.

comparison to between-seed variability. We conclude that, in terms of regret, PBGI-D is less sensitive to hyperparameter choice compared to PBGI, whose behavior was previously shown in Figure 3.

### D.4 Effect of unknown costs

The Robot Pushing empirical benchmark involves two cost functions: a known-cost variant based on the maximum operational duration, reflecting the evaluation time, and an unknown-cost variant based on the total distance traversed, a proxy for energy use similar to the variant considered by Astudillo et al. [4]. One can therefore ask: is the resulting algorithm behavior similar in these two settings? Figure 13 shows that the answer is *yes*: for both the known-cost and the unknown-cost settings, LogEIPC, LogEICC, and PBGI-D outperform MFMES and the non-myopic BMSEI baseline.

### D.5 Alternative visualization of experimental results: means and standard errors

Figure 14 gives an alternative version of Figures 4 to 6: instead of showing the median and quartiles, it shows the mean and standard error under 95% confidence, without Bonferroni correction. This can be used, for instance, to assess the influence of the number of random seeds used on our results.

### D.6 Kernel and problem hyperparameters

**Choice of kernel.** To check whether our results are sensitive to the kernel used for the Gaussian process model, we replicated the Bayesian regret experiments with Matérn kernels with smoothness parameters $\nu = 3/2, 5/2$, as well as the squared exponential kernel, which is the limit of Matérn kernels as $\nu \to \infty$ [33].

Similar to the original results of Figure 4, we can clearly see from Figure 15 and Figure 16 that behavior splits into three regimes:

1. *Easy:* $d$ sufficiently-small, most policies achieve similar performance.

2. *Medium-hard:* $d$ moderate-to-large, both PBGI variants match or outperform baselines.

3. *Very hard:* $d$ sufficiently large, no policy outperforms random search.

We also see that $d = 32$ lands in the very-hard regime for the uniform-cost case but not for the cost-aware case: intuitively, this occurs because costs can reduce the effective volume of the search space, since high-cost regions without promising points can be excluded from search.

This behavior is consistent among different kernels, but where the exact threshold at which regimes switch differs. In particular, for the squared exponential kernel, the separation between the medium

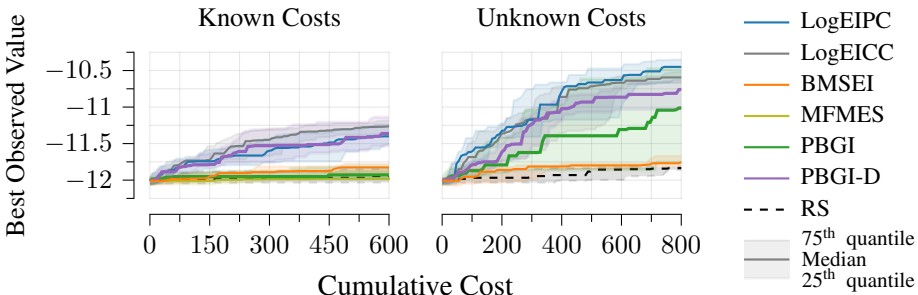

Figure 13: Experimental results for the Robot Pushing empirical benchmark, with the known-cost variant (left) and unknown-cost variant (right). We see that LogEIPC, LogEICC, and PBGI-D performing best in both cases.

and the hard regime appears earlier than for the other variants: all policies there have similar performance to random search when $d = 16$.

**Choice of length scale.** To check whether our results are sensitive to the Gaussian process model's length scale, we compare $\kappa = 10^{-1}$ with $\kappa = 5 \cdot 10^{-1}$ and $\kappa = 10^0$. From Figure 17 and Figure 18, which show uniform-cost and cost-aware results, respectively, we can see that PBGI variants' performance tends to improve relative to most baselines as dimension increases in both scenarios. Since $\kappa = 10^0$ and $\kappa = 5 \times 10^{-1}$ result in easier problems than $\kappa = 10^{-1}$, in the uniform-cost case under $d = 32$, these two length scales land into the medium-hard regime rather than the very-hard regime.

**Synthetic benchmark dimension.** To better understand the effect of problem dimension in settings outside of Bayesian regret, we repeat the synthetic benchmark experiments with $d = 4$, $d = 8$ and $d = 16$. Results in Figure 20. Since $d = 4$ and $d = 8$ are less-difficult, PBGI-D usually performs comparably to baselines, while PBGI either under-performs or over-performs, depending on the objective. This behavior might be explainable by PBGI's sensitivity to hyperparameters in comparison with PBGI-D, whose performance is more-uniform.

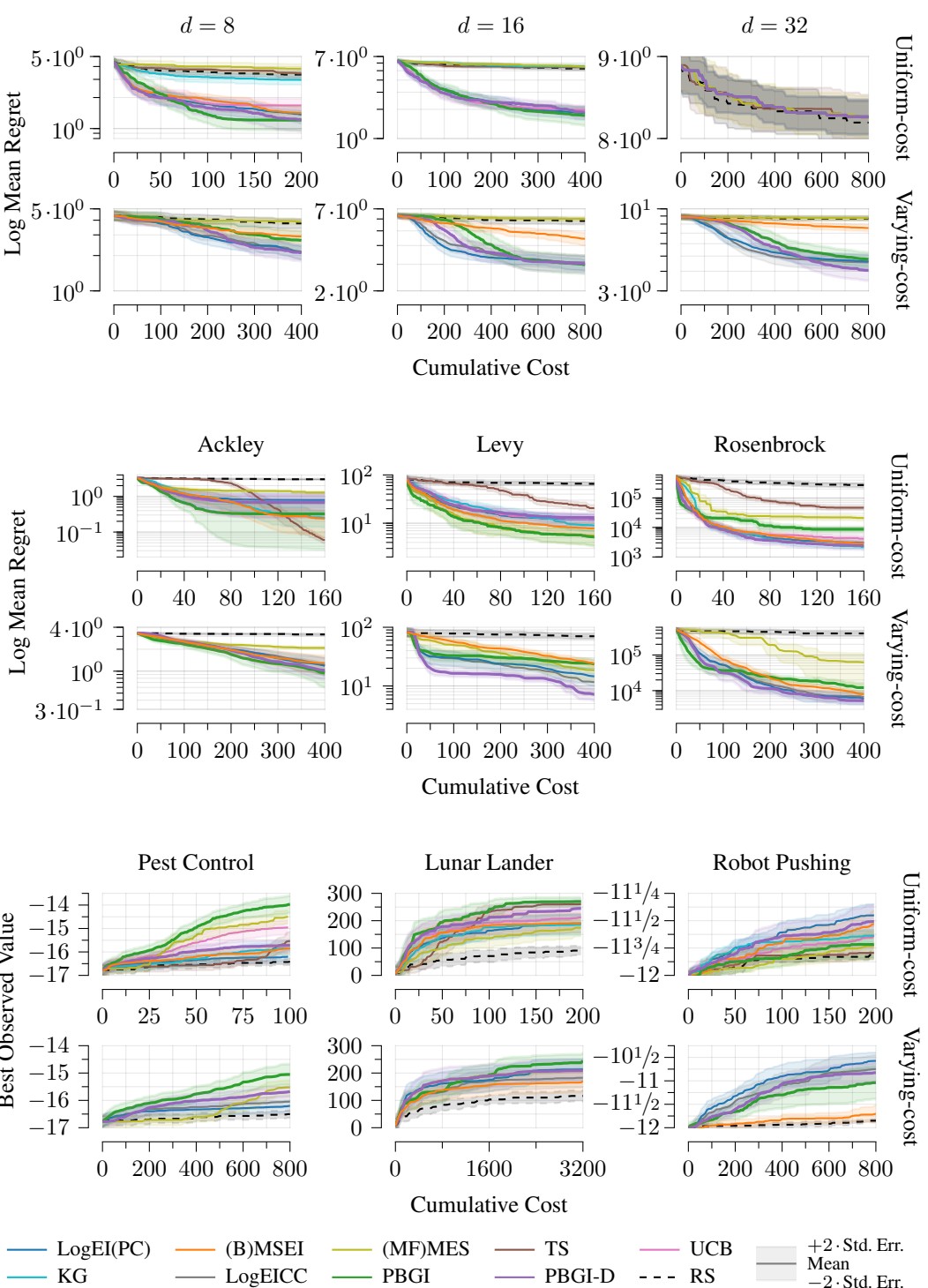

Figure 14: Regret curves for the Bayesian regret and synthetic benchmark experiments, as well as best observed values for the empirical experiments, showing using means (solid lines) and two times standard errors (shaded regions). See Figures 4 to 6 for details.

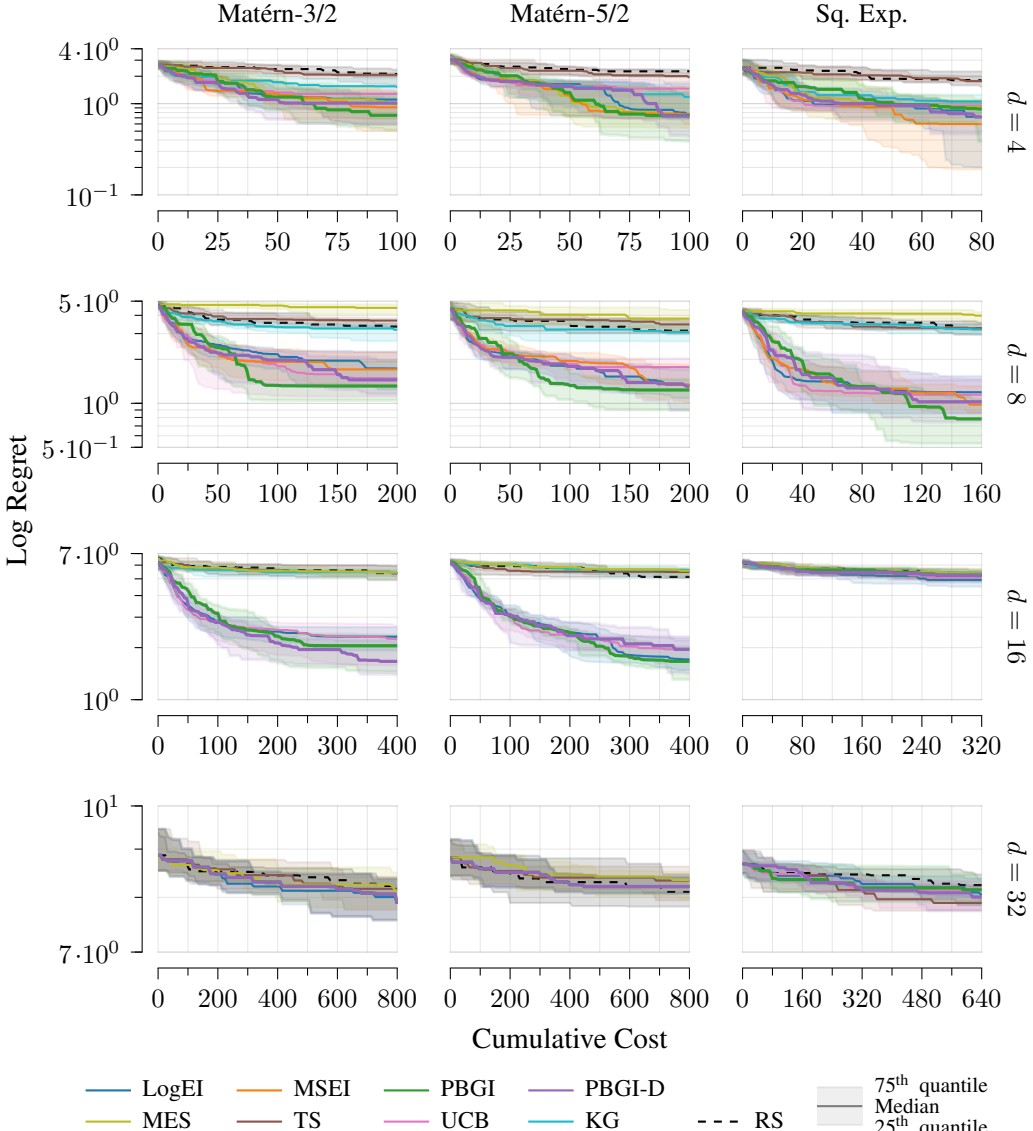

Figure 15: Comparison of Bayesian regret across Gaussian process priors with different kernels over different dimensions, in the uniform-cost setting. All length scales are $\kappa = 10^{-1}$. We see that overall behavior is similar, but the precise thresholds at which each example switches between the easy, medium-hard, and very hard regimes differ.

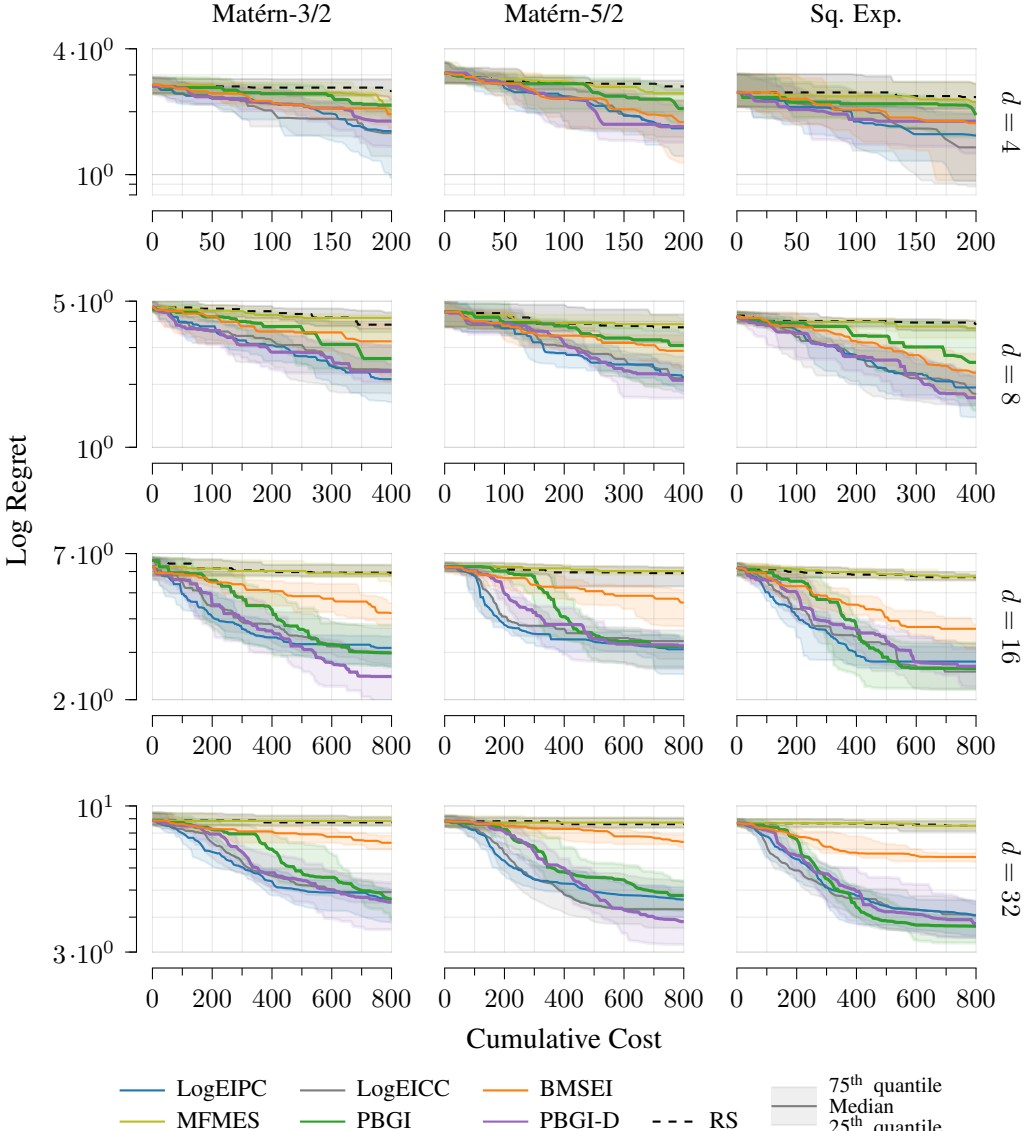

Figure 16: Comparison of Bayesian regret across Gaussian process priors with different kernels over different dimensions, in the cost-aware setting. All length scales are $\kappa = 10^{-1}$. We see that overall behavior is similar, but the precise thresholds at which each example switches between the easy, medium-hard, and very hard regimes differ.

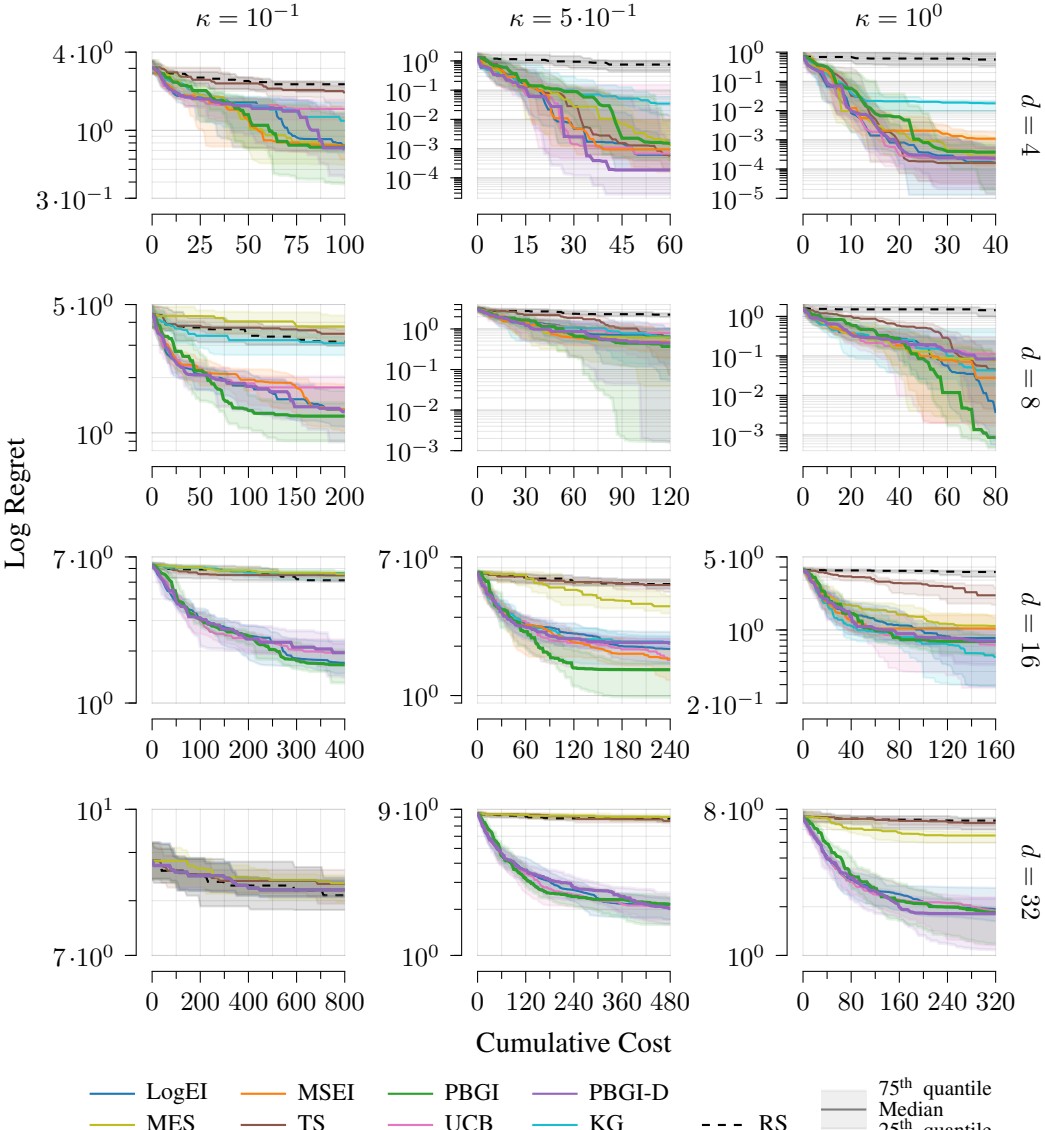

Figure 17: Comparison of Bayesian regret across different length scales and dimensions, with a Matérn-5/2 kernel, in the uniform-cost setting. We see similar overall behavior, but each example switches between the easy, medium-hard, and very hard regimes at different precise thresholds.

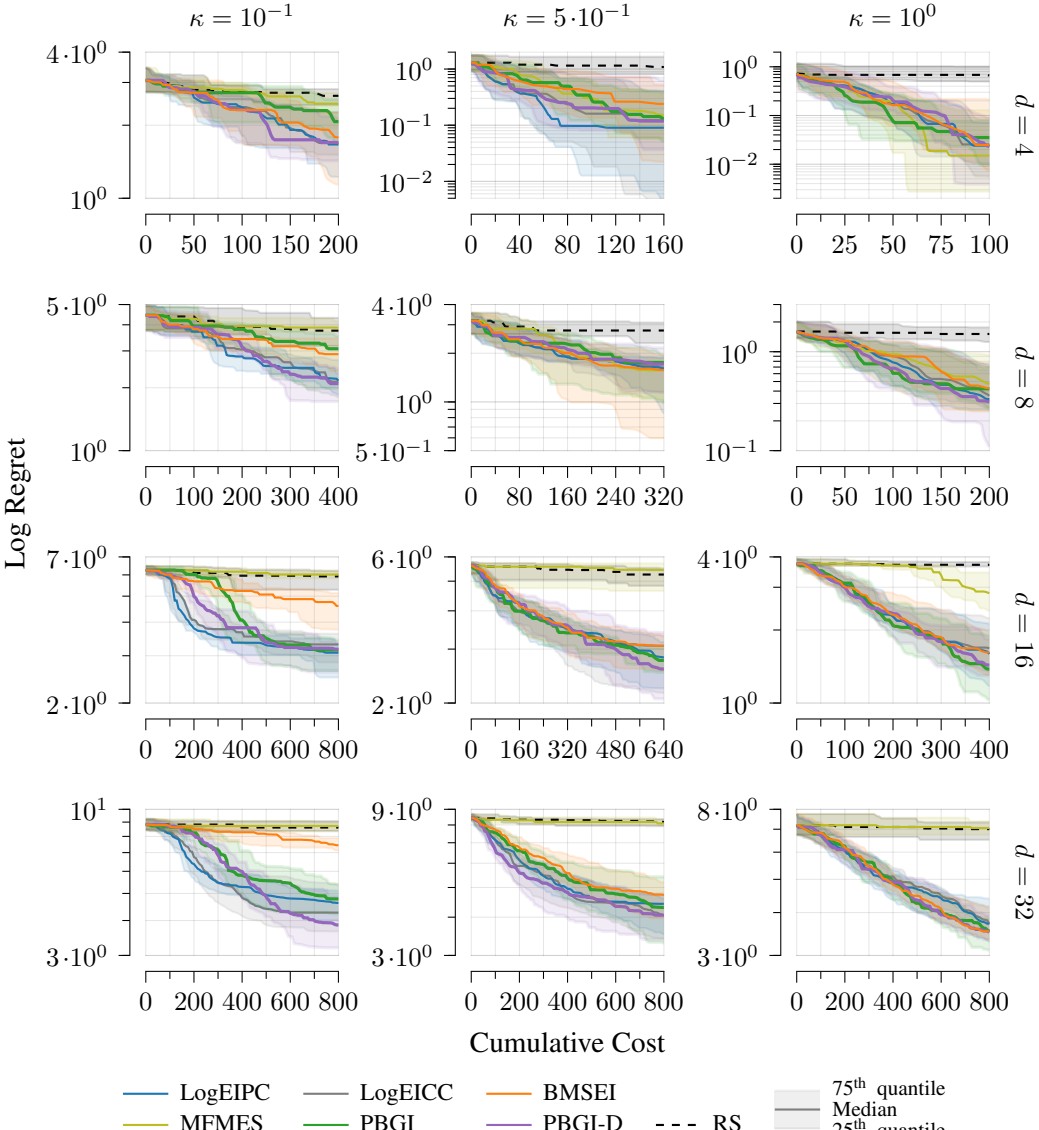

Figure 18: Comparison of Bayesian regret across different length scales and dimensions, with a Matérn-5/2 kernel, in the cost-aware setting. We see similar overall behavior, but each example switches between the easy, medium-hard, and very hard regimes at different precise thresholds.

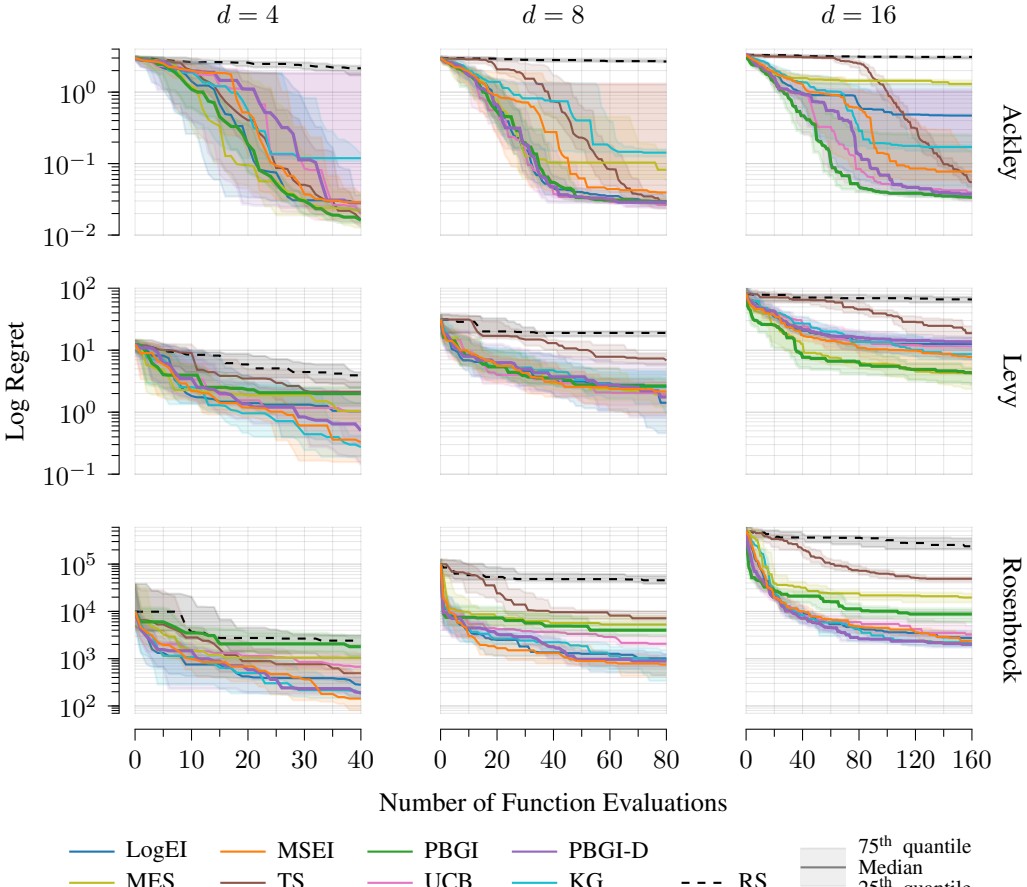

Figure 19: Comparison of regret for synthetic benchmark functions under different dimensions, in the uniform-cost setting. We see that most methods perform similarly for $d = 4$, with differences between the most-competitive methods emerging as dimension increases to $d = 8$ and $d = 16$.

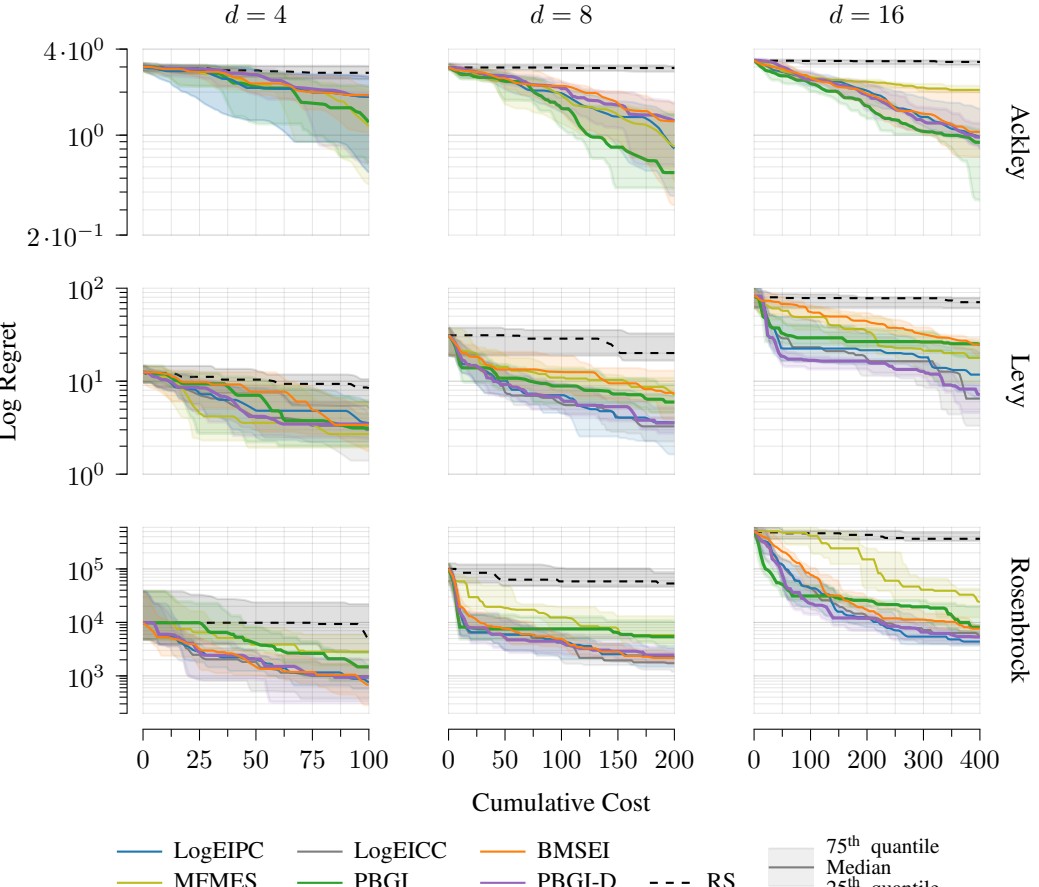

Figure 20: Comparison of regret for synthetic benchmark functions under different dimensions, in the cost-aware setting. We see that most methods perform similarly for $d = 4$, with differences between the most-competitive methods emerging as dimension increases to $d = 8$ and $d = 16$.

