# OpenReview forum: "Cost-aware Bayesian Optimization via the Pandora's Box Gittins Index"
_NeurIPS.cc/2024/Conference — NeurIPS 2024 poster_

### Official Review · Reviewer_q648 · 2024-06-29

**Soundness:** 4
**Presentation:** 2
**Contribution:** 4
**Rating:** 8
**Confidence:** 5

**Summary:**

The paper proposes the Pandora's Box Gittens Index, a cost-aware acquisition function. Moreover, it questions the behavior of previous cost-aware acquisition functions, and motivates the behavior of PBGI, through connections to the Pandora's Box problem and the behavior under different, uniform costs. Theoretical results are presented to support the claims that are made. Results under both uniform and cost-varying setups display strong performance.

**Strengths:**

Firstly, I am generally positive towards the paper. However, I believe the presentation is in part a substantial strength of the paper, but also its biggest drawback (see "Why does it work well?" and "Constant Cost Experiments"). I will provide the specifics under each section.

__Presentation__: The connection to the Pandora's Box problem is handled very nicely. As someone unfamiliar to the problem, the introduction to it, the connection to BO, and the resulting algorithms were clear, pedagogical and had the right level of detail. I appreciate the effort on the authors' end to do this, specifically in 3.1 and 3.2.

__Relevant Problem(s)__: Both cost-aware and budgeted BO are highly relevant and practical problems, that are commonly treated as distinct. An acquisition function which accounts for both is enticing. Moreover, the obvious flaws in existing methods enhances the relevance further.

__Thorough Experimental Evaluation__: Experimental results are provided on a wide range of tasks, from highly controlled to real-world. The benchmarked algorithms appear to be the relevant ones. Some experiments would do well with additional replications to reduce the error bar widths, but this is a _very_ minor remark.

__Theoretical Results__: Meaningful theoretical results on the behavior of the algorithm are presented.

__Visuals__: The plots are mostly very well made.

**Weaknesses:**

__Why does the algorithm work well?__ This is mostly related to 3.3, and the results section. After reading up until 3.3, I believe I have understand what I need to know up until that point. However, I am still left wondering how the algorithm will behave, and why this is beneficial (specifically for fixed-cost). PBGI appears to be a more exploratory variant of EI (which I believe is desperately needed, but only from anecdotal evidence). It should be visualized more precisely what PBGI will do compared to EI for a simple, either 1 or 2D example to see the differences - ideally under different costs or budgets.

__Constant Cost Experiments__: If we are effectively tuning the exploratory behavior of EI or UCB (the latter is perfectly tunable as-is) by making up a constant cost objective on Ackley or Levy, what is there to learn? Are we saying that EI should intrinsically be more exploratory? Can we equally well find costs that are highly detrimental to PBGI? Should we only local search if costs are consistently high? Do we get drastically different behavior if we  multiply all Ackley output by 100, and if so, why is this sensible?

__Acquisition Computation Details__: The acquisition computation requires a solve for $\alpha^*$ involving the cost and EI. This seems like a rather tedious excercise, so is it computationally expensive? 3.3 provides explanation, but it would be nice to see empirical evaluation on this, too. Moreover, can gradients still be computed through autodiff? How does this affect acquisition optimization? Since there is no longer a simple forward pass, this should be expanded on.

__Normalization of cost__: How are costs and objective value normalized in PBGI? Ackley ranges from 0 to 24 (if in the -32 to 32 range) whereas Rosenbrock ranges from 0 to $10^6$, roughly. Does that mean that PBGI is vastly more exploratory for Rosenbrock, since the cost would be way lower relative to the objective? Since I found this to (unfortunately) be rather important and not clear, I am putting it as a weakness.

With that said, I believe these weaknesses are what separates a currently good paper from a very good one.

**Questions:**

__Details on cost__: For unknown costs, the cost is modeled as a GP. Is the mean cost used for acquisition computation, and why do the authors consider the log cost? While I am aware that $\log c$ is convention, I would have thought that the authors' framework did not need transformations of the metrics to work well. Why is log cost the right metric to consider?

__Lack of normalization__: The examples that are considered in the plots do not normalize the input space. Is this meaningful for the behavior of the algorithm?

**Limitations:**

Limitations have been properly addressed.

---

> ### Author Rebuttal · Authors · 2024-08-06
>
> > Strengths:
> > [...]
> > Presentation: [...]
> > Relevant Problem(s): [...]
> > Thorough Experimental Evaluation: [...]
> > Theoretical Results: [...]
> > Visuals: [...]
>
> Thank you for the kind feedback - we're glad you appreciated the paper!
>
> > Weaknesses:
> > Why does the algorithm work well?  [...] wondering how the algorithm will behave, and why this is beneficial [...] It should be visualized more precisely what PBGI will do [...]
>
> This is a great idea! **Please see Figure A of the PDF attached to the main response, which includes an additional visual** that we will add to the appendix in the next draft.
>
> > Constant Cost Experiments: [...] what is there to learn? [...] EI should intrinsically be more exploratory? Can we [...] find costs [...] detrimental to PBGI? [...] Should we only local search [...]? [...] behavior if we multiply all Ackley output by 100 [...]?
>
> We largely agree, but the devil is in the details. In our view, a good way to think about this is via a "needles-in-haystacks" metaphor: some haystacks might have better needles than others, but **exploring a large haystack is only worth it if one expects to actually find the needle in it before running out of budget**. In the discrete expected-budget-constrained problem, Proposition 2 tells us that tuning a risk-seeking-vs.-risk-averse parameter is necessary to handle this tradeoff optimally for each budget, and EI includes no such parameter. In the correlated case, we expect such tradeoffs to loosely map to local-vs.-global tradeoffs, but we believe mainly that **it is the "haystack-size" that matters, not locality** in the sense of whether that haystack comes from a nearby or far-away location.
>
> We have **not yet been able to find costs that are detrimental to PBGI**, though some objective functions - such as the unimodel Rosenbrock objective - can be detrimental. We think because other acquisition functions' behavior - which otherwise serve as a weakness - in this specific case counterbalance model mismatch.
>
> Since **multiplying outputs is equivalent to scaling costs and normalizing**, and our examples do use normalization (see below), we expect this behavior to be **comparable to that shown in Figure 3** which studies the effect of varying $\lambda$: we expect large-$\lambda$ values to reach better objective values faster at first, but eventually lose-out to small-$\lambda$ values which start off slow but eventually find a better location.
>
> > Acquisition Computation Details: [...] is it computationally expensive? [...] can gradients still be computed through autodiff? [...] Since there is no longer a simple forward pass, this should be expanded on.
>
> This involves solving a one-dimensional convex optimization problem: we do so very efficiently using bisection search. Moreover, Appendix A2 shows that **no additional optimization is needed to compute gradients using autodiff - one forward pass is enough**, in contrast with other situations. Finally, **Figure 9 in Appendix C provides a runtime comparison** which shows that the solution is a little bit more expensive than EI, but significantly less expensive than non-myopic and other more-computationally-expensive baselines.
>
> > Normalization of cost: [...]
>
> This is an extremely important question: we accidentally omitted certain details regarding normalization from our appendix, which we have now added. In short, **we normalize our inputs and outputs using standard practices** in all cases where this makes sense. Specifically:
> * We **always normalize input values to be in $[0,1]^d$**, except for the illustration of the EIPC counterexample in Figure 2, where this is omitted in order to present easier-to-visualize numbers. There, normalization is equivalent to changing the length scale, so this change does not affect algorithmic behavior.
> * We **normalize output values** in all cases except where the objective functions are sampled from the prior (Bayesian regret), for exactly the reasons you mentioned. For synthetic benchmark functions specifically, we scale the Ackley, Levy, and Rosenbrock functions by constants given in the script `synthetic.py`. This results in **objective values roughly in $[0,10]$** which matches the prior's scale in most regions of space, except a small volume of outlier-regions the algorithms learn to avoid.
> * For Bayesian regret, we omit normalization because **this particular setting by definition has no normalization-mismatch**, as the prior and objective come from the same distribution, with the same kernel, amplitude and length scale parameters.
> * To empirically test the effect of scaling on Rosenbrock, where our scaling constant was perhaps not ideally chosen, we **include an additional plot in Figure B attached to the PDF of our main referee response**. These show slight improvement for PBGI and PBGI-D, with the latter now competitive with the best baselines.
> * Very regrettably, we accidentally forgot to document normalization details in the submission manuscript draft, though it can be seen in our code. We have **added the normalization information in full detail in an updated appendix** in the current manuscript draft.
>
> > Questions:
> > Details on cost: [...] why do the authors consider the log cost? [...]
>
> We consider log-costs because **this ensures that costs are positive**. If negative costs are allowed, they can create undesired behavior where the algorithm repeatedly evaluates a location with negative costs over and over again, because the costs are effectively interpreted as a deterministic reward.
>
> > Lack of normalization: [...] plots do not normalize the input space. [...]
>
> This is a critically important question: we **do normalize the input space to be $[0,1]^d$** in all cases where it makes sense, which can be seen in the code in but was accidentally omitted from the manuscript draft's appendix. This has been explained in detail in the current version. Please see the preceding answer with bullet-points for more details on normalization.

---

> ### Comment · Reviewer_q648 · 2024-08-10
> **Response to Rebuttal**
>
> Thanks to the authors for their effort in producing a nice rebuttal. Having read the other reivewers' reviews, I feel more confident in my my own. The algorithm is novel, addresses an important niche and is well-supported both theoretically and empirically. Moreover, it is carefully presented.
> While there is certainly room for relevant follow-up (i.e. setting $\lambda$, noise treatment), that in itself suggests that the paper can be impactful. Moreover, I am personally curious about its application in a multi-fidelity context.
> ____________
> ## Why does the algorithm work well?
> Thank you, this is exactly the type of plot I think the paper needs. To me, it explain what the purpose of the algorithm is, and combined with the haystack argument, captures the essence of it nicely.
>
> On this topic, it would seem like PBGI achieves a very similar trade-off to the rarely used improvement threshold in EI, e.g. https://automl.github.io/SMAC3/main/_modules/smac/acquisition/function/expected_improvement.html#EI (see the xi hyperparameter). Are the authors aware of any similarities between the two?
>
> __Note:__ I do _not_ think this restricts the novelty of the method.
>
> ## Normalization and Constant Cost
> Thanks for the clarification. I still do not perfectly grasp this, as values in [0, 10] are not exactly conventional, either (rather, standardizing the current set of observations). As such, it does still seem to me like the algorithm is dependent on the output range of the function, and perhaps this is unavoidable in the cost-aware setting.
>
> ______
> I strongly feel that this paper should be accepted. As such, I will happily vouch for it.

---

> ### Author Response · Authors · 2024-08-11
>
> We sincerely appreciate your thoughtful and encouraging feedback! We are delighted to hear that you find our method novel and appreciate your interest in potential follow-up work! Below, we aim to clarify further the points raised in your review. Thank you once again for your valuable feedback and support.
>
>
>
> # Similarities between PBGI and the rarely-used EI
> Thanks for bringing up this point. The $\xi$ parameter in the variant of EI you mention is different from but related to the Gittins index. In particular, for the uniform-cost maximization problem, **the Gittins index should be [the value of $\xi$ such that the threshold-adjusted EI equals $\lambda$] - [the current best observed value]**. For the minimization problem which the EI in your reference link is used for, the minus sign should be replaced with a plus sign.
>
> However, **$\xi$ and $\lambda$ exhibit similarity in monotonicity when controlling the exploration vs exploitation trade-off** -- both larger $\xi$ and larger $\lambda$ imply higher exploitation. This can be illustrated using a toy Pandora's Box example involving a choice between a closed box and an open box with the current best-observed value inside: as $\xi$ increases, you are more likely to take the open box, which is a sign of exploitation. Similarly, as $\lambda$ (representing the cost to open the closed box in this setting) increases, you are also more likely to take the open box. **Note that this similarity is only qualitative, not quantitative!**
>
>
> # Normalization and Constant Cost
> You are correct that our algorithm relies on the output range of the objective function when the costs are not scaled. To clarify, we normalize the outputs not merely to adhere to standard practice, but primarily for two reasons. First, theoretically, for a fixed GP prior, a fixed budget, and fixed costs, the best value of $\lambda$ should be consistent across problems, assuming a well matched prior. Second, in practice, normalization allows us to treat different problems, such as Rosenbrock and Ackely, which may have varied output ranges, as if they are drawn from the same fixed GP prior. This approach enables us to apply the same $\lambda$ value across different problems: normalization makes using such a value at least reasonable. The above reasoning also applies to the uniform-cost setting.

---

> > ### Comment · Reviewer_q648 · 2024-08-11
> > **Acknowledgement**
> >
> > Thanks to the authors for their further response, The responses make sense so me, I appreciate the clarification.

---

### Official Review · Reviewer_7Tfe · 2024-07-02

**Soundness:** 2
**Presentation:** 3
**Contribution:** 3
**Rating:** 5
**Confidence:** 3

**Summary:**

This paper draws connections between cost-aware Bayesian Optimization (BO) and a problem from the economics literature, called the Pandora's Box problem. Based on these connections, the paper proposes a new cost-aware acquisition function, called Pandora's Box Gittins Index (PBGI). Numerical experiments show that PBGI achieves competitive performance against usual acquisition functions.

**Strengths:**

Viewing cost-aware BO under the lens of the Pandora's box problem is interesting. This work suggests that promising solutions might lie at the intersection between economics and black-box optimization.

The paper is also well-written.

**Weaknesses:**

**Impact of the lengthscale $\kappa$ (1)**. In the experiments, the hyperparameter $\kappa$ is fixed arbitrarily. Although an ablation study about $\kappa$ is given in Appendix C, it seems unlikely that a GP with an arbitrary lengthscale will properly fit to the observed data. Because it is difficult to compare the performance of optimization algorithms when they use models that are unfit to the observed data, I think this weakens the experimental part of the paper.

**Impact of the lengthscale $\kappa$ (2)**. Furthermore, as far as I understand the paper, the performance of PBGI seems to heavily rely on a trade-off between the magnitude of $\kappa$ and the dimensionality of the function domain. Roughly speaking, PBGI should also have good performance on low-dimensional spaces if $\kappa$ is low enough. Conversely, it should perform poorly on high-dimensional spaces if $\kappa$ is large enough. Studying the relation between $\kappa$ and $d$ would strengthen the contribution.

**Calibration of $\lambda$ and $\beta$**. In the numerical experiments, $\lambda$ and $\beta$ are also set arbitrarily. I understand that calibrating them is less obvious than calibrating $\kappa$, but it is nevertheless a crucial part of the PBGI. As far as I know, the paper does not provide a principled way to calibrate these parameters. I believe providing one would strengthen the contribution.

**Noisy setting**. The experiments seem to have been conducted in a noiseless setting. However, in zeroth-order optimization tasks, objective black-boxes often cannot be observed without an observational noise. Therefore, it would have been interesting to evaluate PBGI's performance within a noisy setting. The results, including the ranking of the acquisition functions, may differ.

**Notation**. The notation for the posterior distribution $f | y_1, \cdots, y_t$ is incorrect and should be replaced by $f | \mathcal{D}$, with $\mathcal{D} = \\{(x_1, y_1), \cdots, (x_t, y_t)\\}$.

**Questions:**

Here are some questions to spark the discussion with the authors. My reservations are mainly about the experimental part of the paper, I will be happy to increase my score if these are addressed during the discussion period.

(1) Why not use MLE to find the appropriate lengthscale for each experiment?

(2) Have you studied the relation between $\kappa$ and the dimensionality of the function domain $d$ w.r.t. PBGI performance?

(3) Do you have any insights on how to calibrate $\lambda$ and $\beta$?

(4) Do you consider noise in your experiments? If not, do you have any insights on the behavior of PBGI in a noisy setting?

**Limitations:**

Yes they did, and I believe the discussion period will shed even more light on the limitations of PBGI.

---

> ### Author Rebuttal · Authors · 2024-08-06
>
> > Strengths:
> > [...] interesting. [...]
> > [...] well-written.
>
> Thank you very much for your review! We are delighted that key strengths - specifically, **novelty, via a brand-new technical perspective on Bayesian optimization** - are recognized. We would like to draw your attention to these points: **we think our strengths have significant merit**, and that they may outweigh some of the weaknesses and points of improvement discussed below. In light of a certain unfortunate typo, we also believe that our manuscript already addresses at least some of your concerns about the length scale mismatch (see below), though we will make additional effort to make this clearer in the next draft. On these bases, we would like to gently ask you to reconsider your score.
>
> > Weaknesses:
> > Impact of the lengthscale $\kappa$ (1). In the experiments, the hyperparameter $\kappa$ is fixed arbitrarily. Although an ablation study about $\kappa$ is given in Appendix C, it seems unlikely that a GP with an arbitrary lengthscale will properly fit to the observed data. [...]
>
> There was a typo in the paper on Line 245: please let us apologize and clarify how our experiments treate length scales:
> * In the Bayesian regret experiments (Sec. 4.1), **the objective function is drawn from the prior, which has a fixed length scale,** and all of the algorithms use a kernel with this same lengthscale.
> * In other experiments, all kernel hyperparameters - including length scales - are **learned using max marginal likelihood**.
>
> We believe that this concern is primarily a result of this typo. In both types of experiments, each algorithm's GP either **has the correct lengthscale from the beginning or learns the correct lengthscale from data**.
>
>
> > Impact of the lengthscale $\kappa$ (2). [...] relation between $\kappa$ and $d$ [...]
>
> This is a good point: what seems to matter is the "number of effective boxes" induced by $\kappa$, $d$, and the volume of $X$. We include results on this in the appendix. We suspect the best way to get a more precise picture would be to prove a suitable regret bound. While we are **extremely interested in studying this question**, doing so is likely to **involve enough new theoretical machinery for a whole separate paper**, so we prefer to defer it to a follow-up work.
>
> > Calibration of $\lambda$ and $\beta$. [...]
>
> We agree: studying how to set these parameters automatically is a great research question! In principle, proving a regret bound and setting these parameters to minimize the bound could provide for an avenue by which to do so. For the same reasons as in the prior answer, we believe this would make for **a great follow-up to our work**.
>
>
> > Noisy setting. [...]
>
> The primary reason we don't study this is that **in the presence of noise, the analog of the Pandora's Box model and Gittins index machinery is significantly more complicated**. One needs to create a "noisy Pandora's box" model, but this requires one to potentially inspect a noisy Pandora's box multiple times to better estimate its value. There is Gittins index machinery that can handle this type of multi-step inspection in principle, but computation becomes significantly harder because one will not obtain an analytic, expected-improvement-like optimization objective. One must also consider how to translate to the Bayesian optimization setting, for instance by inspecting many nearby points instead of reinspecting the same point many times. We believe this is one of several interesting potential extensions that our work opens the door to.
>
> > Notation. [...]
>
> Thank you for drawing our attention to this point. While it is reasonably-standard in the Gaussian process literature to omit dependence on $x$-values (for example, the paper *Efficiently Sampling Functions from Gaussian Process Posteriors*, Wilson et al., ICML 2020, also does this), in retrospect we agree that it could be made clearer and will examine this in the next manuscript draft.
>
> > Questions:
> > [...]
> > (1) Why not use MLE to find the appropriate lengthscale [...]
>
> **We do this:** our synthetic and empirical experiments include length scale optimization. We apologize again for the typo which incorrectly stated this was absent.
>
> > (2) [...] relation between $\kappa$ and the dimensionality $d$ [...] w.r.t. PBGI performance?
>
> **In short: yes, our Apdx. C5 and C6** study this, but developing a more principled evaluation would make for a well-motivated follow-up paper. Please see the preceding answers on length scales.
>
> > (3) [...] how to calibrate $\lambda$ and $\beta$?
>
> **We deliberately did not tune these parameters for each problem in order to make our claims stronger: we get good performance even if they are not set optimally**. Theorem 2 and the right-hand-side plot in Figure 3 tell us that $\lambda$ should be tuned to match the expected budget: a natural way to do so is to (a) run the algorithm, (b) look at whether the regret first quickly goes down, then levels-off too soon, and if so, (c) make $\lambda$ smaller. Our PBGI-D variant is a first attempt at doing this automatically, though it introduces the problem of calibrating $\beta$). We believe there is more to explore regarding calibrating the parameters, e.g. through a theoretical regret analysis, but doing so is beyond the scope of this work.
>
> > (4) Do you consider noise [...]
>
> No, our experiments are all noise-free. As we discussed in a previous answer, we have ideas about how one could create a Pandora's box model and Gittins index that handles noise, but it would require enough novel technical development that it could well be a follow-up paper in its own right.

---

> > ### Comment · Reviewer_7Tfe · 2024-08-09
> > **Rebuttal Ack**
> >
> > Thank you for the detailed rebuttal and interesting discussions.
> >
> > I'm glad to see the lengthscales are optimized with MLE. I am positive about the paper, but I still believe finding an automatic way to calibrate $\lambda$ and $\beta$ would strengthen the contribution.
> >
> > Furthermore, taking noise into account is quite important as many (most) applications of BO do involve a noisy objective function.
> >
> > I've increased my score to 5.

---

> ### Author Response · Authors · 2024-08-10
>
> # Automatically Tuning Lambda
>
> Thank you very much for your response! In our response, we had originally included an idea for how to automatically tune $\lambda$ in a setting where we can run Bayesian optimization multiple times, but this got cut due to rebuttal space limitations. We expand on this below.
>
> Our automatic tuning idea below **applies in settings where evaluating the true objective function is the key computational bottleneck** (for example, running a large physics simulation, training a large neural network). In this setting, it is (a) important to choose $\lambda$ well to ensure high-quality acquisitions, and (b) worthwhile to spend some degree of computation on choosing $\lambda$.
>
> The key to our proposal comes from Figure 3 (which is an 8-dimensional Bayesian regret experiment), from which we can make two observations:
>
> * **As one varies $\lambda$, PBGI's regret curve varies in a qualitatively predictable way.** Specifically, large-$\lambda$ regret curves first drop sharply, then flatten out, whereas small-$\lambda$ regret curves drop less-sharply, but eventually reach a better value.
> * **Testing a relatively small number of $\lambda$ values is sufficient to find a good choice.** Figure 3 uses a small number of geometrically spread $\lambda$ values, namely $10^{-n}$ for $n \in $ {$2, 3, 4, 5$}.
>
> Motivated by this - and by theoretical considerations given below - we propose the following tuning scheme: every few iterations, **run a Bayesian regret experiment comparing PBGI across a range of geometrically spread $\lambda$ values, then set $\lambda$ to the one with best performance at the desired budget**. For the prior GP in the Bayesian regret experiments, use the current posterior GP from the "real" Bayesian optimization loop, and draw the objective function for the experiment from that same GP. For an expensive-enough true objective function, running such as experiment can be computationally viable, and it outputs a $\lambda$ value that is very likely to be good, if not optimal, for the real experiment.
>
> **This approach is theoretically principled**, in the following sense.
> One of the lemmas used in our analysis of Theorem 2 shows that **the optimal $\lambda$-value in Pandora's Box satisfies an optimization problem whose gradient relates the Gittins index policy's expected costs with the budget.** Setting this gradient to zero, we obtain an equation which balances the two terms, which the scheme above directly targets.
>
> Furthermore, **using PBGI-D with initial value set to the tuned $\lambda$-value for the real selections provides an additional layer of safety against getting stuck with a too-large $\lambda$-value**. When PBGI-D decides to decrease $\lambda$, one could use that as a signal that another Bayesian regret experiment to tune $\lambda$ is warrented: this may be more efficient than simply running the Bayesian regret experiments every few iterations.
>
> In summary: **the above idea gives a clean and principled way to automatically tune $\lambda$**. One can also apply a similar Bayesian-regret-based scheme to automatically tune $\beta$, but **our empirical results reveal that performance is not very sensitive to the $\beta$ parameter**. Thank you for bringing this point back to our attention! We will add these ideas to the next manuscript draft.
>
>
> # Noisy Observations
>
> Let us also expand on what happens in the noisy observation case, to provide **justification for why this should be out of scope of the current work**.
>
> The optimality claim in Pandora's Box arises by comparing one closed box with one open box.
> In the case where we have observations with noise, including the ability to repeat observations to reduce the effect of noise, the one-closed-one-open box model now consists of a **Nested Pandora's Box** - a Pandora's Box which contains another Pandora's Box inside of it, repeated recursively.
> Modulo certain technical details, we think that one could show using abstract Gittins-index-theoretic machinery that (in the discrete case) this setup also admits an optimal Gittins index policy.
>
> The key technical question becomes: **what is the objective of the convex optimization problem defining the Gittins index policy?** In the Pandora's Box case, this is analytic, given by expected improvement. In the nested analog of Pandora's Box, **we cannot expect an analytic formula, and would need to rely on novel numerical methods yet to be developed** to compute the respective optimization objective function.
> We therefore believe that this situation, while - as you say - extremely relevant, important, and interesting, would be **too complex to describe within the 9-page NeurIPS limit and would therefore best be presented in a follow-up paper.**

---

### Official Review · Reviewer_iB7T · 2024-07-12

**Soundness:** 2
**Presentation:** 2
**Contribution:** 2
**Rating:** 4
**Confidence:** 4

**Summary:**

The paper introduces the Gittins index, a novel perspective from the pandora box problem, to address the cost-aware optimization problem on unknown rewards. It offers a theoretical justification for adapting the Gittins index as an acquisition function and offers empirical results against previous works, demonstrating the effectiveness of the proposed method.

**Strengths:**

1. A novel perspective connecting previously independent works on different types of cost-aware optimization problems.
2. The figures help illustrate the connection between the pandora box problem and cost-aware BO.
3. The performance seems robust in various tasks. And it makes sense that a myopic cost-aware acquisition behaves well when only aiming at minimizing simple regret.

**Weaknesses:**

1. The justification for the algorithm poses a major concern. The conversion to BO lacks sufficient discussion, especially when the c(x) surrogate could be constantly updated. One potential problem with the adopted conversion is that the original Gittins index ignores posterior dynamics. When the cost function is unknown, the noisy rewards and noisy costs let the posterior converge to the underlying functions at the picked candidate. Then the PBGI at that location could remain superior when the cost functions posterior converge much faster relatively, meaning the acquisition is trapped locally. I believe more discussion is crucial for the soundness of the proposed method.

2. There are several problems with clarity. Most outstanding is the availability of the cost function. It seems that most of the paper regards the cost function as given, while in the experiment section and some of the appendix, the cost function is unknown and extracted from another Gaussian process in some cases. This could be confusing, especially given that the unknown cost function typically requires corresponding treatment, and it differentiates from the original Pandora's box problem. The discussion should highlight the differences from the beginning. Also, the paper lacks discussion over the connection to previous work, especially on how the proposed acquisition function avoids arbitrary performance deterioration, as shown in Astudillo et al. [3].  Another problem lies in the Bayesian optimality. Is that defined with respect to formula (4)? Then how is that connected to the optimality in cost-aware Bayesian optimization? The optimality in cost-aware Bayesian optimization should be agnostic to the choice of $\lambda$.

3. There are also known practical concerns (Wu et al. 2020) that different magnitudes of costs could amplify the misspecification in GP, which has not been discussed, at least in the related work section.

Reference:
- Wu, Jian, Saul Toscano-Palmerin, Peter I. Frazier, and Andrew Gordon Wilson. "Practical multi-fidelity Bayesian optimization for hyperparameter tuning." In Uncertainty in Artificial Intelligence, pp. 788-798. PMLR, 2020.

**Questions:**

1. What are the expectations in (4) with respect to?
2. Could the author clarify the discussion in line 104 to line 109. I'm not sure if the context assumes constant cost or not.
3. Could the author clarify the optimality claimed in line 103? I'm not aware that EI bears the tightest BCR or BSR bound.

**Limitations:**

Discussed above.

---

> ### Author Rebuttal · Authors · 2024-08-06
>
> > Strengths:
> > [...] novel perspective [...]
> > [...] figures [...]
> > [...] performance seems robust [...]
>
> Thank you very much for your review! We are delighted that these key strengths - including **novelty, specifically a brand-new technical perspective on Bayesian optimization** - are recognized.
> We would like to draw your attention to these points: **we think our strengths have significant merit**, and that they may outweigh some of the weaknesses and points of improvement discussed below.
> We would thus like to gently ask you to reconsider your score.
>
> > 1. The justification for the algorithm poses a major concern. [...] I believe more discussion is crucial for the soundness of the proposed method.
>
> Thank you very much for this insightful comment, which has led us to **carefully think about consistency properties and related phenomena in a new way**. In short: the finite-$\lambda$ version of our algorithm is designed for finite budgets. Asymptotic consistency requires one to think about cases where the budget is infinite. We think PBGI-D may be a better algorithm for this setting, and your comment has led us to make some preliminary theoretical investigations here that we describe in more detail in the **follow-on comment that we will post as soon as this is enabled on OpenReview.**
>
> > 2. [...] availability of the cost function. [...] most of the paper regards the cost function as given, while in experiment section and some of the appendix, the cost function is unknown [...]
>
> Thank you for pointing this out! We will clarify that:
> * We consider both known and unknown costs.
> * **The primary focus of our work, and the default assumption throughout, is known costs**.
> * When needed, we model unknown costs as log-normal processes, and show how this integrates into our algorithms.
> * Potential limitations of our strategy for handling unknown costs.
>
> Our motivation for focusing on known deterministic costs is that, given we develop novel techniques, **it makes sense to thoroughly understand the simpler case first**. Thoroughly understanding how to handle unknown costs, as well as theoretical analysis (such as regret bounds), are a promising direction for future work that our work opens the door to.
>
> > [...] previous work [...] avoids arbitrary performance deterioration [...] Astudillo et al. [3].
>
> **The key is in Figure 2**: in this example, there is a high-variance high-cost point, and many low-variance low-cost points. The **correct decision (in terms of simple regret) is to pick the high-variance high-cost point**: PBGI does this, while EIPC does not. We discuss this in the paragraph titled *Qualitative behavior and comparisons* on lines 225-233. We also discuss a similar example regarding EI in Figure 8, and have added discussion on these points to the appendix.
>
> > [...] Bayesian optimality. Is that defined with respect to formula (4)? [...] optimality [...] should be agnostic to the choice of $\lambda$.
>
> You're (mostly) correct: Bayesian-optimality in the **cost-per-sample setting, where there is no $\lambda$**, is defined with respect to (4). Bayesian-optimality in the **expected budget-constrained setting**, which requires a choice of $\lambda$ given budget $B$, is defined in Section 2 using the same objective as (4) but with two modifications: (i) costs are not part of the objective, and (ii) only policies whose sum of costs does not exceed some budget $B$ *in expectation* are allowed. The relationship between the two settings, particularly the existence of an optimal $\lambda$, is given by Proposition 2.
> We have **clarified this relationship in full detail in an updated technical appendix** which has been substantially reworked and expanded in the time since this work's submission.
>
> > 3. [...] practical concerns (Wu et al. 2020) that different magnitudes of costs could amplify the misspecification in GP [...] related work [...]
>
> As requested, we'll discuss Wu et al. 2020 in the related work. **The concern in that paper (see their Section 2.4) is specific to the multi-fidelity setting with continuously-varying fidelity**, where one can measure fidelities with near-zero value at near-zero cost. This concern does not typically arise in the single-fidelity setting we study.
>
>
> > Questions:
> > What are the expectations in (4) with respect to?
>
> These are as follows:
> * The random function values $f(1), \dots, f(N)$ for each of the $N$ points in the discrete domain.
> * Choices made by the algorithm, namely the stopping time $T$ and the inspected points $x_1, \dots, x_T \in$ {$1, \dots, N$}. These choices can depend on function values observed so far, and they may also depend on external randomness (though, by standard MDP theory, external randomness is not necessary to solve the problem optimally). Specifically, after inspecting at time $t$, the algorithm chooses whether to stop and, if not, the next point to inspect $x_{t + 1}$, based on the data $\mathcal{D}_t = \{(x_1, f(x_1)), \dots, (x_t, f(x_t))\}$.
>
> We will clarify these details in the next revision.
>
> > [...] clarify the discussion in line 104 to line 109. [...] assumes constant cost or not.
>
> This discussion assumes constant cost, but an analogous discussion could be made for heterogeneous costs too. In particular, the discussion also applies to ordinary EI, not just EIPC. We will reorganize the discussion so that its scope is clearer, and expand on appropriate details, such as the idea behind the counterexample of Astudillo et al. (2021).
>
> > Could the author clarify the optimality claimed in line 103? [...]
>
> We mean EI is the one-step-lookahead greedy algorithm.
> That is, if we consider an arbitrary set of data observed so far $\mathcal{D} = \{(x_1, f(x_1)), \dots, (x_t, f(x_t))\}$, then EI chooses the point $x_{t+1}$ maximizing $\mathbb{E}(\max(f(x_1), \dots, f(x_t), f(x_{t + 1})) \mid \mathcal{D})$.
> One can thus view EI as optimal for a one-step-truncation of the original problem.
> We will clarify this in the next revision.

---

> ### Author Response · Authors · 2024-08-08
>
> # Reviewer iB7T - follow up comment regarding posterior dynamics
>
> > The justification for the algorithm poses a major concern. The conversion to BO lacks sufficient discussion, especially when the c(x) surrogate could be constantly updated. One potential problem with the adopted conversion is that the original Gittins index ignores posterior dynamics. When the cost function is unknown, the noisy rewards and noisy costs let the posterior converge to the underlying functions at the picked candidate. Then the PBGI at that location could remain superior when the cost functions posterior converge much faster relatively, meaning the acquisition is trapped locally. I believe more discussion is crucial for the soundness of the proposed method.
>
> We wanted to start by thanking Reviewer iB7T again for these insightful questions about soundness.
>
> In our view, the most important soundness properties are performance bounds - that is, bounds on simple or cumulative regret. **We view regret bounds as an important direction for follow up work, but we expect it to be a hard enough analysis to be outside the scope of the present paper.**
>
> Nevertheless, the comment inspired us to think about **(a) whether there are weaker soundness properties which we could investigate** in our current submission, such as consistency; and **(b) whether or not the type of example you describe could occur**, with PBGI getting "trapped", and how to mitigate if so. This led to two outcomes:
>
> * **We now have a proof sketch that our PBGI-D algorithm is consistent.** We plan to incorporate this into our next revision - either as a fully fleshed out proof, or as a high-level discussion motivating PBGI-D, or possibly both.
> * **We now believe that one might be able to achieve performance comparable to the better between our two algorithms** - namely, that of PBGI with constant $\lambda = 1/10000$, and PBGI-D with initial $\lambda = 1/10$ and $\beta = 1/2$ - by simply using PBGI-D with initial $\lambda = 1/10000$ (and still $\beta = 1/2$). We will evaluate this hypothesis as soon as it is feasible for us to do so comprehensively.
>
> Below, we discuss these in more detail. We note that **the reasoning throughout is similar for known and unknown costs**, so we focus mainly on the known-cost case of primary interest in our paper.
>
> ## Achieving the best of both PBGI and PBGI-D
>
> We believe that **for large time horizons, decreasing $\lambda$ as the algorithm proceeds is important**. Otherwise, at time points which occurs sufficiently-long after the Gittins stopping rule triggers, without decreasing $\lambda$ the algorithm might become too-greedy.
>
> The **purpose of the fixed-$\lambda$ PBGI is to do well in the setting where we know we will eventually stop**: this setting is by definition finite-time, so one need not expect good asymptotic behavior without modifications such as decay.
>
> In our experiments, we examined PBGI-D with large initial values of $\lambda$, while we examined PBGI for a broader range of values. From thinking carefully about your comments, **we suspect PBGI-D with small initial values of $\lambda$ may in some cases outperform PBGI with constant $\lambda$**, particularly if the initial value and constant are set equal.
> * The intuition here is that, from Figure 3, the regret of PBGI with any constant value of $\lambda$ eventually levels off, causing PBGI-D to later catch up in performance.
> * Since, before the first $\lambda$-decrease, PBGI-D and PBGI make the same decisions and achieve the same regret, it might be possiboe **achieve the best of both worlds by mimicking PBGI on short horizons, but removing the eventually-level-off behavior via $\lambda$-decrease** as in PBGI-D. We will run comprehensive experiments to evaluate this possibility for the next manuscript version.
>
> ## Consistency of PBGI-D
>
> **We believe that PBGI-D is consistent**. Recall that every time the Gittins stopping rule triggers, the algorithm decreases $\lambda$. This causes it to eventually explore more in a manner similar to other acquisition functions. **We believe this holds regardless of whether the costs are known - our main focus - or they are unknown.**
>
> Our preliminary calculations show that for small $\lambda$, point $x$'s Gittins index is $\approx \mu(x) + \sigma(x) \sqrt{2\log \frac{\sigma(x)}{\lambda c(x)}}$, where $\mu(x)$ and $\sigma(x)$ are the mean and standard deviation of the objective model, and $c(x)$ either the known cost or, if unknown, the expected value of the cost model's mean. This should prevent any open neighborhood of the domain from being ignored by PBGI-D, because **for sufficiently small $\lambda$, unexplored regions where $\sigma(x) \gg 0$ will always have better Gittins index than explored regions where $\sigma(x) \approx 0$, thus implying consistency**.
> Even unknown costs do not pose a great obstacle, because as $\lambda \to 0$, the dominant term is $\sigma(x) \sqrt{2\log\frac{1}{\lambda}}$, which is independent of the cost.

---

> > ### Comment · Reviewer_iB7T · 2024-08-11
> >
> > I appreciate the authors' comprehensive response, especially in answering my questions about the scope of the paper, focusing on known cost settings and the consistency of the proposed algorithm. The authors actually promise much beyond the current presentation to be available in future revisions, and I place my trust in most of the points. Yet there are several remaining concerns.
> >
> > First is the claim that the unknown cost setting would bear similar property to the known cost setting when analyzing the consistency. This is not immediately clear to me, as for an unknown cost, I'm not sure what would happen if the model suggests that specific candidates' costs are arbitrarily close to zero due to observation noise, and the assumption is that cost has to be above 0. It seems necessary to truncate the scope of the model for the cost function with an additional assumption that is not typical in BO. This also resonates with my remaining concern over the problem formulation. The author claims that the extreme values only trouble the multi-fidelity setting, so what differentiates the assumption of cost in multi-fidelity BO and the paper's assumptions on the cost?
> >
> > Second is the experiment; in the uniform cost setting with a finite budget, UCB and TS reduce to their non-asymptotic version, where the $\beta_t$ controlling the exploration-exploitation trade-off could be much smaller than the asymptotic version, achieving the exact cost-awareness as is discussed in line 104 to line 109. Though it is limited to the uniform cost setting, such straightforward solutions to make UCB and TS cost-aware should not be omitted in both the discussion and experiments. Otherwise, it could be partially misleading.

---

> ### Author Response · Authors · 2024-08-11
>
> # Consistency in the unknown-cost setting
> > I appreciate the authors' comprehensive response [...] several remaining concerns.
>
> Thank you very much for your comments! **While we do promise to discuss consistency of PBGI-D, given the above reasoning and aforementioned sketch - which amounts to reasonably-standard theory - we are confident we can do it.** Let us address your remaining concerns below.
>
> > [...] unknown cost [...] similar property to the known cost [..] analyzing the consistency. [...] costs are arbitrarily close to zero due to observation noise [...] extreme values only trouble the multi-fidelity [...] what differentiates the assumption of cost in multi-fidelity BO and the paper's assumptions [...]?
>
> The key property of single-fidelity Bayesian optimization is that for practical cost functions, **the infimum of costs of inspecting any point is strictly positive**.
> This is assumed implicitly in our consistency sketch from the previous reply: we regret, in light of space, that we had omitted saying this. The specific way this assumption is used is to ensure $\frac{1}{\lambda c(x)}$ approaches $\infty$ uniformly in $x$ as $\lambda \to 0$.
>
> In contrast, **the assumption of a strictly positive minimum cost need not hold in the continuous-fidelity variant of the multi-fidelity setting**. There may be a spectrum of fidelity levels whose costs converge to zero at the low-fidelity end of the spectrum - this is studied by Wu et al. (2020).
>
> You raise a good point about unknown costs: here is the extent to which our proof sketch covers them. **Our consistency sketch requires that the $c(x)$ values being plugged into our algorithm have a strictly positive minimum value.** We believe this should hold, as long as the true costs have a positive minimum and match the smoothness properties of the lognormal cost model's kernel, but other assumptions might work too. For what it's worth, we believe a global lower bound on the cost of an inspection - which need not be tight - is generally known in practice, arising for instance from the minimum amount of time to run an experiment.
>
> Your point about noisy observations of costs possibly complicating this picture is well taken, but **noisy observations - whether of data or of costs - are outside the scope of our work**. Our work shows that building Gittins index machinery that handles observation noise is a promising future direction, but **handling noise properly requires non-trivial technical development on the Gittins index side (see [this official comment's second heading](https://openreview.net/forum?id=Ouc1F0Sfb7&noteId=la5dFZIeR7))**.
>
> # Cost-awareness of UCB and TS
> > Second is the experiment; in the uniform cost setting with a finite budget, UCB and TS reduce to their non-asymptotic version, where the  controlling the exploration-exploitation trade-off could be much smaller than the asymptotic version, achieving the exact cost-awareness as is discussed in line 104 to line 109. Though it is limited to the uniform cost setting, such straightforward solutions to make UCB and TS cost-aware should not be omitted in both the discussion and experiments. Otherwise, it could be partially misleading.
>
> We appreciate your suggestion to extend the discussion of the limitations of EI and EIPC to UCB and TS, and we will include this in the next version of our manuscript. However, we have two points of confusion that we hope you can clarify:
> 1. **Thompson sampling is by definition not a cost-aware algorithm, and includes no tuning parameters (and, in particular, no learning rates) that make it naturally extend to a tunable cost-aware algorithm.** Moreover, (a) we are not aware of any papers proposing cost-aware extensions of Thompson sampling, and (b) we are not sure what you mean by "asymptotic version of Thompson sampling" as this algorithm has no parameters to tune that could be chosen in an asymptotic or non-asymptotic way. As consequence, could you please clarify precisely how you are thinking about making Thompson sampling into a cost-aware algorithm, so that the notions above make sense?
> 2. On the other hand, since the uniform-cost setting is equivalent to the finite-horizon setting studied in bandits, **UCB can be extended naturally to a uniform-cost-only cost-aware algorithm** by tuning the learning rate appropriately. Using this correspondence, **non-asymptotic tuning (that is, tuning which yields a regret bound valid for any finite T) for controlling explore-exploit tradeoffs is given by Srinivas et al. (2009) - we are already using their higher-performing empirical tuning in our experiments**. While this tuning might not be perfect due to gaps in the analysis of Srinivas et al. (2009), **our tuning of lambda is deliberately set to be imperfect to demonstrate some degree of robustness** and therefore indicating that our improved performance - particularly outside the Bayesian regret setting where we improve significantly on UCB - is not due to superior tuning.

---

> > ### Author Response · Authors · 2024-08-12
> >
> > Thank you very much for your prior responses! We very much appreciate your engagement with our submission and your follow-up questions.
> >
> > We were wondering if our replies have adequately addressed some or all of your most significant concerns, both from your original review and your follow-up comment. If there are major concerns you feel have not been sufficiently addressed, could you please highlight them?
> >
> > And, if there are major concerns we have addressed, we would appreciate it if you were willing to reconsider your score, in light of these clarifications and improvements.

---

> > ### Comment · Reviewer_iB7T · 2024-08-13
> >
> > I appreciate the follow-up by the authors regarding the problem formulation and the comparison to existing methods. I'd like to conclude my evaluation with the following.
> >
> > I believe that all reviewers appreciate the idea of this work, which brings a novel perspective to the cost-aware BO. And I also think that this one-step optimal algorithm could possibly bear a strong performance.
> >
> > On the other side, though the high-level idea is conveyed well, I believe the details in the presentation, especially the problem formulation, the scope of the work, and the analysis of the consistency and Bayesian optimality, require non-trivial revision to qualify it for acceptance, from my perspective. Nevertheless, I trust the author on this part.
> >
> > I'm also concerned about the experiments. Regarding the UCB and TS questions I raised in my previous comments. I want to bring the paper by Chowdhury and Gopalan (2017) to the author's attention as TS should also allow applying a $\beta_t$. My previous comments mentioning the discussion in lines 104 to line 109 are intended to say that if there exists theoretical analysis on dealing with different constant costs, it could be misleading. At least to me, the discussion in lines 104 to line 109 leaves an impression that there is no such solution in literature, not limited to EI. In addition, the information-theoretic method is missing in the baselines. Despite the authors' argument that the formulation differentiates from the multi-fidelity setting, the known cost scenarios could naturally be handled by the existing SOTA entropy-based method (MF)BO algorithm, e.g., MF-MES by Takeno et al. (2020). Such an algorithm could efficiently approximate the acquisition using MC estimation while the known cost problem regresses to a special case discussed by the corresponding work. Since the author argues that known cost is the primary focus of the paper, missing such baselines could downgrade the significance of the empirical strength, which is one important merit of the paper, given that the regret rate is not available.
> >
> >
> > ***References***
> >
> > - Chowdhury, Sayak Ray, and Aditya Gopalan. "On kernelized multi-armed bandits." In International Conference on Machine Learning, pp. 844-853. PMLR, 2017.
> >
> > - Takeno, Shion, Hitoshi Fukuoka, Yuhki Tsukada, Toshiyuki Koyama, Motoki Shiga, Ichiro Takeuchi, and Masayuki Karasuyama. "Multi-fidelity Bayesian optimization with max-value entropy search and its parallelization." In International Conference on Machine Learning, pp. 9334-9345. PMLR, 2020.

---

> ### Author Response · Authors · 2024-08-14
>
> Thank you for making us aware of this variant of Thompson sampling that allows for an exploration / learning rate parameter! We will add this to our references, and we will feature UCB and Thompson sampling our discussion on Lines 104-109 accordingly.
>
> We appreciate the suggestion to compare against an information-theoretic acquisition function. In the time since submission, we've benchmarked against predictive entropy search (PES). **Our results indicate that on uniform-cost problems, the performance of PES is significantly worse than other baselines, such as UCB.** We believe this may be in part due to the difficulty of optimizing the PES acquisition function, which requires a Monte Carlo scheme for handling gradients. Given that Takeno et al. (2020) provides a similar acquisition function using the max-value entropy search (MES) divided by the cost (i.e., it is "MES per unit cost"), we believe it would suffer from similar difficulties. Moreover, we expect this acquisition function to suffer from issues similar to those of EIPC.

---

> ### Comment · Reviewer_iB7T · 2024-08-14
>
> I appreciate the authors' responses to my concerns over the literature and experiments. From my perspective, MES (and multi-fidelity MES, which is the cost-aware variant) alleviates the problem of optimizing the PES acquisition function and shows better performance than PES, as was reported in the MES paper. They are widely available as part of the Botorch standard library and are easy to implement. One tutorial is available at the following link: https://botorch.org/tutorials/max_value_entropy. Most importantly, the mutual information-based method bears better theoretical interpretation than EIPC in both the asymptotic and finite-budget scenarios. I don't foresee the problem of EIPC being extended to MF-MES unless the author offers corresponding proof or empirical evidence. Then, I would be pleased to increase my score and trust that the authors could amend the corresponding results in the camera-ready version.

---

> ### Author Response · Authors · 2024-08-14
>
> # Theory sketch: counterexample for MES
>
> **Here is a proof sketch showing that MES or MES/cost can have arbitrarily poor performance compared to the optimal policy.**
> This proof sketch is inspired by the Astudillo et al. 2021, though the construction is slightly different and takes specific advantage of the characteristics of MES.
> While this example is discrete and uses binary-valued arms for simplicity of understanding, **we believe that these ideas can be extended to construct a smooth counterexample where MES variants (with or without cost)** will perform poorly when used with Gaussian processes.
>
> The example has two groups of arms:
>
> * Group 1: $N$ independent arms ("high-value arms")
>   * Cost is $1$
>   * Value is $1$ with probability $\epsilon$ and 0 with probability $1-\epsilon$.
>
> * Group 2: $M$ additional independent arms ("lower-value arms")
>   * Cost is $1$
>   * Value is $1-\delta$ with probability $1/2$ and $0$ with probability $1/2$
>
> The probability distribution over the maximum is:
> * $\max = 1$, with probability $p(N)$, where $p(N) = 1-(1-\epsilon)^N$
> * $\max = 1-\delta$, with probability $(1-p(N))(1-2^{-M})$
> * $\max = 0$, otherwise
>
> We send $M$ to infinity so that these probabilities become:
> - $\max = 1$ has probability $p(N) = 1-(1-\epsilon)^N$
> - $\max = 1-\delta$ has probability $1-p(N)$
> - $\max = 0$ does not happen
>
> Here, $\epsilon$ and $\delta$ are strictly positive values.
>
> The information gain is:
> * Strictly positive for pulling a high-value arm (this can be seen via direct computation of the expected entropy or via an argument based on mutual information)
> * Goes to zero as $M$ goes to infinity for pulling a lower-value arm (since pulling the arm does not change the distribution of the max, in the limit as $M$ goes to infinity)
>
> First, observe the following, holding $\epsilon$ and $\delta$ fixed.
> Suppose we have a budget $B$ less than $N$.
> * For sufficiently small $\delta>0$, it is optimal to pull the lower-value arm repeatedly until one gives value $1-\delta$, after which it is optimal to pull a higher-value arm. This gives value at least $(1-\delta)(1-2^{-B})$, where $B$ is the budget.
> This problem meets the assumptions of the expected budget constrained Pandora's box setting and so, for an appropriately chosen $\lambda$, by optimality, the Gittins index policy chooses this optimal arm.
> * But MES/cost continually pulls the higher-value arm, and obtains value $1-(1-\epsilon)^B$.
>
> Then, we send $\epsilon$ to $0$ (we must ensure that $M$ goes to $\infty$ at a sufficiently fast rate relative to $\epsilon$).
> Under this change, MES/cost obtains value converging to $0$,
> while the value of the optimal policy remains strictly positive and bounded below by $(1-\delta)(1-2^{-B})$.
>
> Thus, the value of MES/cost divided by the value of the optimal policy converges to $0$.
>
> **We will incorporate this argument - together with a Bayesian optimization example in the spirit of Figure 2 - in the next manuscript version. We will also add MES and MF-MES as baselines to all our experiments in the next manuscript version.**

---

> ### Comment · Reviewer_iB7T · 2024-08-14
>
> I sincerely appreciate the expedited response by the authors. I apologize for finding it difficult to understand the sketch proof fully. Here are my questions about this proof.
>
> 1. The proof claims that MES/cost favors the high-value arms. I, in general, understand that the intention is to dilute the mutual information of high-value arms slower through the slow decrease of $\epsilon$ and a fast increase of $M$. Yet the exact rate is unclear to me. The mutual information for both sets is strictly positive unless $M \rightarrow \inf$ for the lower value set and $\epsilon\rightarrow\inf$. I prefer that the authors offer the exact rate as the feasibility of directing MES/cost to the high-value arms is not obvious, especially when the 1/2 distribution for each arm in the lower-value set makes the arms actually attractive to MES. In addition, when $\epsilon\rightarrow\inf$, the mutual information also goes to zero for the higher value set. If the $\epsilon$ does not go to $\inf$, I don't think the MES/cost is arbitrarily worse. Instead, the higher value set is arguably more favorable with a sufficient budget, though the expected reward could possibly be lower.
>
> 2.  The unit cost actually regresses the scenario to standard BO. In this case, the analysis of MES reveals an order optimally simple regret bound, yet the results seemingly contradict that.  Could the author comment on it?

---

### Official Review · Reviewer_ykvm · 2024-07-16

**Soundness:** 3
**Presentation:** 2
**Contribution:** 2
**Rating:** 5
**Confidence:** 4

**Summary:**

This work tackles cost-aware Bayesian optimization using the Pandora's box Gittins index. In particular, in this paper, the authors focus on expected budget-constrained cost-aware Bayesian optimization and cost-per-sample cost-aware Bayesian optimization. Then, they provide some evidence of the proposed method in order to show the validity of their method. Finally, several experimental results are demonstrated to show the performance of the proposed method.

**Strengths:**

- Cost-aware Bayesian optimization is an important research topic in Bayesian optimization.
- The method proposed in this work provides a new perspective on cost-aware Bayesian optimization.

**Weaknesses:**

- Some parts of this work are not clear.
- Experimental results seem not promising.
- Writing can be improved more.

**Questions:**

- For "Thus, EIPC is perhaps best suited to settings where heterogeneity is the main factor at play," what is heterogeneity here? I think it is not clear.
- Line 111, for "is in widespread use," can it be a real reason?
- For the Pandora's boxes, do you assume that there exist infinite boxes? or, there exist a finite number of boxes?
- Line 152, a period is missing.
- How did you optimize Equations (5) and (6) in practice?
- Figures 2, 3, and 4: which function is used to be optimized for each figure?
- Figure 3: I don't understand the effect of $\lambda$. According to Figure 3, it should be as small as possible.
- For dynamic decay, it is just constant decay, right? $\beta$ is a constant.
- Line 248, how did you choose $\lambda$ and $\beta$? Is there any guidance for these selections?
- Line 250: for "even though per-problem tuning could be advantageous," is it guaranteed? Is there any evidence for this?

**Limitations:**

I don't think that there are any particular societal limitations of this work.

---

> ### Author Rebuttal · Authors · 2024-08-06
>
> > Strengths:
> > [...] important research topic [...]
> > [...] new perspective [...]
>
> Thank you for your review! We are delighted the key strengths - (a) **the importance of the topic**, and (b) **novelty, specifically a brand-new technical perspective on Bayesian optimization** - are recognized. We would like to draw your attention to these points: **we think our strengths have significant merit**, and that they may outweigh some weaknesses and points of improvement, particularly given clarifications below. We would like to gently ask you to reconsider your score on this basis.
>
> > Weaknesses:
> > Some parts [...] not clear.
>
> We would be delighted to hear pointers to specific paragraphs so that we can improve our work.
>
> > Experimental results seem not promising.
>
> **We respectfully disagree.** Bayesian optimization is reasonably mature, along with the algorithmic ideas underlying state-of-the-art baselines. **Introducing a simple new algorithmic idea that on most benchmarks matches or outperforms these baselines - sometimes significantly - is, we believe, promising progress.** For instance, our results show strong performance on cost-aware Bayesian regret (objective functions sampled from the prior) in $d=16$ and $d=32$, where both Gittins variants are **significantly better than every competitor**, including the non-myopic and significantly more elaborate MSEI/BMSEI acquisition function. This performance is repeated on synthetic examples (Ackley and Levy) and empircal ones (Pest Control, Lunar Lander), where Gittins variants come out on top, with a small set of exceptions for which we discuss limitations.
>
> Finally, _note that our plots display quartiles, not standard error_. One can thus expect overlap in the interquartile regions even for large sample sizes, and cases where ranges do not overlap - such as the cost-aware variants mentioned above - are especially significant.
>
> > Writing [...]
>
> We plan to clarify the answers to your and other reviewers' questions (see below). We'd be delighted to hear other specific suggestions regarding clarity.
>
> > Questions:
> > For "Thus, EIPC [...] settings where heterogeneity is the main factor [...]" what is heterogeneity here? [...]
>
> _Heterogeneity_ refers to a non-constant cost function $c(x)$. We have edited this to explicitly state that this refers to heterogeneity of costs.
>
> > Line 111, for "is in widespread use," can it be a real reason?
>
> Could you please clarify this? Here we say EIPC is in widespread use in the cost-aware setting. Given that this acquisition function is default in BoTorch, we believe this correct to our best understanding.
>
> > [...] do you assume that there exist infinite boxes? [...]
>
> In the Pandora's Box problem, the number of boxes $|X|$ is finite: this is given on line 126.
> In general cost-aware Bayesian optimization, we allow general compact domains $X$, but no longer refer to it as a "set of boxes".
>
> > Line 152 [...]
>
> Thank you for pointing out this typo!
>
> > How did you optimize Equations (5) and (6) [...]?
>
> This is given in the *Computation* paragraph on Line 220-224, which also refers to Appendix A.1. In short: we solve the one-dimensional convex optimization problem via bisection search, then apply an analytic gradient formula that does not require solving any additional optimization problems.
>
> > Figures 2, 3, and 4: which function is used [...]?
>
> These are in the captions. For Figures 2 and 3, we intentionally defer most details in the appendix as these figures serve primarily as visual illustrations. In Figure 2, the objective and cost are scaled squared exponentials, plus a small constant. For Figures 3 and 4, the objectives are drawn from a Fourier feature approximation of the prior, where Figure 3 is the same as the $d=8$ case of Figure 4.
>
> > Figure 3: I don't understand the effect of $\lambda$. According to Figure 3, it should be as small as possible.
>
> Not necessarily: from the right-hand-side plot of Figure 3, we see that **large $\lambda$-values perform better if the time horizon is small**. For instance, $\lambda = 0.01$ performs best for time horizons less than 50, whereas smaller values perform better on longer horizons. Therefore, we view $\lambda$ as a parameter which controls risk-seeking vs. risk-averse behavior in the algorithm, with larger $\lambda$-values being more suitable for smaller budgets and smaller $\lambda$-values better for larger budgets. This is stated in the caption: "We see that large $\lambda$-values decrease regret sooner, but eventually lose out to smaller $\lambda$-values."
>
> > For dynamic decay, it is just constant decay, right? $\beta$ is a constant.
>
> While the _decay multiplier_ is a constant $\beta$, the _times when the multiplier is applied_ are dynamic. Specifically, the dynamic decay variant update $\lambda$ to $\beta \lambda$ every time the Gittins index stopping rule triggers - that is, when every point's expected improvement is worse than its inspection cost. This is described formally in the list item of Line 202 which introduces it.
>
> > Line 248, how did you choose $\lambda$ and $\beta$? [...]
>
> In contrast with what is done in many other works, we **deliberately chose a non-optimally-tuned value** in order to demonstrate that our algorithm can achieve good performance _even if these parameters are not explicitly tuned, especially on a per-problem basis_. This is important, since some of our baselines such as EIPC contain no hyperparameters. We chose a relatively small $\lambda$ based on the observation from the right-hand-side plot in Figure 3, where PBGI with large $\lambda$-values initially decrease regret faster, but eventually lose out to PBGI with smaller $\lambda$-values.
>
> > Line 250: for "[...] per-problem tuning could be advantageous," [...] evidence for this?
>
> Yes, this is shown in the right-hand-side of Figure 3: there, picking $\lambda$ to match the intended time horizon well is shown to result in stronger performance.

---

> > ### Comment · Reviewer_ykvm · 2024-08-13
> >
> > Thank you for your detailed response.
> >
> > > We would be delighted to hear pointers to specific paragraphs so that we can improve our work.
> >
> > This comment is the summary of my review. I have already described specific things in Questions and you have answered them.
> >
> > > Could you please clarify this? Here we say EIPC is in widespread use in the cost-aware setting. Given that this acquisition function is default in BoTorch, we believe this correct to our best understanding.
> >
> > I don't agree with your statement. How does the fact that the default setting of BoTorch is EPIC support this sentence "In spite of this rather negative theoretical outlook, EIPC is in widespread use"? Do the potential users of BoTorch use this algorithm despite of the negative theoretical outlook? I believe that scientific writing should be humble and respectful to other previous work.
> >
> > > These are in the captions. For Figures 2 and 3, we intentionally defer most details in the appendix as these figures serve primarily as visual illustrations.
> >
> > I don't understand these sentences. I think that they are not in the captions (you also mentioned that you intentionally defer most details). I think it is a minor thing. You can point where the details are described more specifically.
> >
> > > In the Pandora's Box problem, the number of boxes $|X|$ is finite: this is given on line 126.
> >
> > Do you update a set of boxes? If so, when is it updated?
> >
> > > The times when the multiplier is applied are dynamic.
> >
> > I don't think it is dynamic because a decay rate is constant. Imagine learning rate decay. We don't call it dynamic decay if a decay rate is constant. But, if you want to call it dynamic decay, you need to clarify this.
> >
> > > In contrast with what is done in many other works, we deliberately chose a non-optimally-tuned value in order to demonstrate that our algorithm can achieve good performance even if these parameters are not explicitly tuned, especially on a per-problem basis.
> >
> > I trust the authors' argument that a non-optimally-tuned value is used. However, it should be supported by scientific and numerical evidence.
> >
> > > Yes, this is shown in the right-hand-side of Figure 3: there, picking $\lambda$ to match the intended time horizon well is shown to result in stronger performance.
> >
> > It should be also supported by numerical results.

---

> > > ### Author Response · Authors · 2024-08-14
> > >
> > > Thank you for your feedback. Let us clarify further the points raised in your comment.
> > >
> > > > I don't agree with your statement. How does the fact that the default setting of BoTorch is EIPC support this sentence "In spite of this rather negative theoretical outlook, EIPC is in widespread use"? Do the potential users of BoTorch use this algorithm despite of the negative theoretical outlook? I believe that scientific writing should be humble and respectful to other previous work.
> > >
> > > **This comment took us by surprise.** Let us quote the relevant lines in full rather than in part:
> > >
> > > * _"In spite of this rather negative theoretical outlook, EIPC has been shown to work well on many practical problems, is computationally efficient and reliable, and is in widespread use. We therefore ask: can one develop a technically-principled and computationally-straightforward alternative with at-least-comparable empirical performance?"_
> > >
> > > We acknowledge that EIPC being the default setting in BoTorch does not _prove_ that it is in widespread use, however much this may suggest it. The theoretical limitations of EIPC are from rather recent work (Astudillo et al., 2021), so it is possible that the community is not yet aware of them. With this said, **proving widespread use of EIPC is not relevant to our contributions**.
> > >
> > > Finally, **it is absolutely standard to discuss strenghs and weaknesses of a key baseline**, especially in the context of motivating the ideas introduced in a paper. We firmly believe our discussion (quoted above) of EIPC's strengths and weaknesses is fair and respectful.
> > >
> > > > I don't understand these sentences. I think that they are not in the captions (you also mentioned that you intentionally defer most details). I think it is a minor thing. You can point where the details are described more specifically.
> > >
> > > For Figure 2, the prior distribution from which the objective functions are sampled and the cost function are **explicitly displayed in the left and middle column of that figure.**
> > >
> > > For Figure 4, these are **given in lines 261 and 499-501:** these state that the objective functions are random functions sampled from a Gaussian process prior using a Matérn-5/2 kernel.
> > >
> > > To avoid any omissions, **we will ensure detailed versions of the points below are added to the updated experiment description appendix:**
> > >
> > > * Figure 2: The probability density function of the prior distribution and the cost function are both scaled Gaussian densities plus a constant.
> > >
> > > * Figure 3: We believe the question refers to the right plot. The objective functions are the same as those for the $d=8$ uniform-cost plot in Figure 4 (see below).
> > >
> > > * Figure 4: The objective functions are drawn from the respective Gaussian process prior, drawn in such a way that different baselines with the same random number seed share the same objective, but objectives for different seeds are different draws from the same prior. (To be fully precise: sample paths are computed up to a negligible approximation using 1024 Fourier features - this technique is standard, see Section 4 of "Pathwise Conditioning ..." by Wilson et al. JMLR 2020 for a description.)
> > >
> > > > Do you update a set of boxes? If so, when is it updated?
> > >
> > > In the Pandora's Box problem, the indices, values, and costs of the boxes are part of the definition and are therefore fixed. **The only change throughout the decision process is the replacement of the reward distribution with the actual value inside once a box is opened.** This is described in full on lines 125-135.
> > >
> > > > I don't think it is dynamic because a decay rate is constant. Imagine learning rate decay. We don't call it dynamic decay if a decay rate is constant. But, if you want to call it dynamic decay, you need to clarify this.
> > >
> > > We chose to use "dynamic" in the name because **the times at which decay occurs are dynamic:** that is, the decay times depend on the data observed by the algorithm. This is clarified on Lines 205–206 in the same sentence that introduces the decay parameter $\beta$.
> > >
> > > > I trust the authors' argument that a non-optimally-tuned value is used. However, it [the claim that problem-specific tuning is advantageous] should be supported by scientific and numerical evidence.
> > >
> > > **This claim is supported by Figure 3, which shows direct numerical evidence that tuning $\lambda$ according to the evaluation budget is advantageous**. This is because the color of the curve with the smallest regret value depends on the time point.
> > >
> > > > It should be also supported by numerical results.
> > >
> > > Please see our response to the previous point.

---

> ### Comment · Reviewer_ykvm · 2024-08-14
>
> Thank you for your response.
>
> > Proving widespread use of EIPC is not relevant to our contributions
>
> This is my point. The authors cannot prove this and it might not be a real reason. I would like to say that the authors used this sentence in the submission and I just asked if it can be a real reason. Then, the authors' answer was it is because of the default setting of BoTorch, which is not relevant to this proof. Now, the authors stated that it does not prove that it is in widespread use.  I simply recommend to remove this sentence if you cannot prove this.
>
> > Finally, it is absolutely standard to discuss strenghs and weaknesses of a key baseline
>
> No, you wrote the weaknesses you cannot prove. It is not absolutely standard.
>
> > Answer for "I don't understand these sentences. I think that they are not in the captions (you also mentioned that you intentionally defer most details). I think it is a minor thing. You can point where the details are described more specifically"
>
> I don't understand why the authors explicitly explain them to me. My point is that I (including potential readers) cannot know which functions are exactly used based on their captions. You have to improve the presentation of the figures.  I don't matter if their details are described in the appendices or the captions. You need to exactly indicate where they are described, when you prepare for a revision.
>
> > In the Pandora's Box problem, the indices, values, and costs of the boxes are part of the definition and are therefore fixed. The only change throughout the decision process is the replacement of the reward distribution with the actual value inside once a box is opened. This is described in full on lines 125-135.
>
> No, Lines 125-135 don't explain whether boxes are updated or not. Thus, I was curious about it.
>
> > We chose to use "dynamic" in the name because the times at which decay occurs are dynamic: that is, the decay times depend on the data observed by the algorithm. This is clarified on Lines 205–206 in the same sentence that introduces the decay parameter $\beta$.
>
> In Lines 205-206, there is no description why it is dynamic.
>
> > This claim is supported by Figure 3, which shows direct numerical evidence that tuning $\lambda$ according to the evaluation budget is advantageous.
>
> It is only for a single function and also there are no confidence intervals.  It is not enough for the numerical evidence of how we can choose $\lambda$ and $\beta$.
>
> I was just curious about the missing details of the submission, which can be added to improve this work.  The authors didn't want to clarify them, and just mentioned the unclear sentences the details are missing.  I still think that this manuscript should be improved more.

---

> > ### Author Response · Authors · 2024-08-14
> >
> > Thank you again! Your specific comments on individual phrases in our manuscript have been very helpful.
> >
> > **(1)** We have had an extended discussion of the phrase "EIPC ... is in widespread use" on line 111 of our manuscript. We understand your viewpoint. We propose to replace "is in widespread use" with the text below.
> >
> > * _Several factors point to EIPC's continued use: it is the default in the BoTorch software package, it has been used in a recent studies (for instance, Pricopie et al. 2024), and the paper that proposed it continues to receive high citation counts (1588 in 2023 alone)._
> > * Reference: _Pricopie, Stefan, et al. "Bayesian Optimization with Setup Switching Cost." Proceedings of the Genetic and Evolutionary Computation Conference Companion. 2024._
> >
> > **(2)** Regarding our discussion about per-problem tuning, your original comment was:
> >
> > * _"Line 250: for "[...] per-problem tuning could be advantageous," [...] evidence for this?"_
> >
> > The full sentence to which you referred is:
> >
> > * _"To ensure that performance differences are not primarily due to tuning, we deliberately use the same $\lambda$-and-$\beta$-values on all problems, even though per-problem tuning could be advantageous."_
> >
> > On basis re-examining your comments, we will simply remove the phrase "even though per-problem tuning could be advantageous." This is not central to our point.
> >
> > **(3)** You gave several useful pieces of detailed feedback on improving the following: figure captions, clearly explaining the motivation for using the word "dynamic" in the name PBGI-D, and more clearly explaining that the boxes themselves do not change (our updated appendix now includes a formal mathematical presentation, as needed to handle the updated Theorem 2).
> >
> > We will make these updates to the text in our revision. While we previously offered detailed answers to your questions, we now understand that you were not trying to tell us that you were confused about those points, and were simply drawing attention to opportunities to improve the clarity of the manuscript. Thank you for these suggestions.

---

> ### Comment · Reviewer_ykvm · 2024-08-14
>
> Thank you for your prompt response.  Indeed I was confused about some points, which have been already resolved.
>
> Because most of my concerns are answered, I will raise my score to 5.
>
> Please update your manuscript considering all the comments provided by reviewers.

---

### Author Rebuttal · Authors · 2024-08-07

# Summary

We thank all reviewers and the area chair for their time and thoughtful feedback and evaluating this work!
We are delighted that **all four reviewers recognized the work's key strengths**, including:
* **Importance of the topic (Reviewer ykvm)**.
* **Novelty in the form of a brand-new technical perspective on Bayesian optimization (all reviewers)**.
* **Clarify of presentation (Reviewer 7Tfe)**, with some exceptions that we have identified and fixed thanks to the feedback.
* **Thorough experimental evaluation (Reviewer q648)**.

We believe these strengths rank favorably relative to the standard for accepted papers at NeurIPS.

We'd also like to draw all readers to the text content of the reviews, which were quite positive - based on the text alone, we would have predicted scores of weak accept. Many of the reviewer concerns focus on small suggestions for improving clarity. Other questions about the technical aspects of our paper and its significance were smaller in scope and we believe are largely addressed by our detailed response.

We look forward to engaging with the reviewers in discussion. Once the reviewers have a chance to review our response and discuss any remaining concerns they might have, we hope that scores can be changed to be consistent with the text of the reviews.

## Key Reviewer Concerns

*Reviewer ykvm*:
* Reviewer ykvm states that our empirical results seem "not promising" but does not offer further explanation or detail. We point out that our empirical results show us **matching or outperforming a comprehensive set of baselines**, including a pair of state-of-the-art non-myopic acquisition functions (MSEI, BMSEI), on problems of large-enough dimension. Given the importance of Bayesian optimization and the maturity of its literature, we believe introducing a new algorithm with such results is a significant contribution. Also, Reviewers iB7T and q648 listed our algorithm's performance as a strength.

*Reviewer iB7T*:
* Reviewer iB7T notes aspects of our presentation that could be made clearer, particularly on whether the cost function is known or unknown. We appreciate the feedback, and will clarify this in revision.
* Reviewer iB7T also asks technical questions about our algorithm's behavior, involving posterior dynamics and consistency properties, especially in the unknown-cost setting. **Our primary focus is the known-cost setting, and the fact that the Pandora's Box Gittins index performs well here is our main empirical finding.** Still, we found that thinking about the aforementioned unknown-cost setting and consistency properties significantly helped us better understand our algorithm's behavior, potential limitations, and avenues for improvement. We have therefore included an **extended discussion on these points in an OpenReview comment** - which we will post once review comments are enabled - and view theoretical analysis thereof as a **promising direction for follow-up work prompted by our results and Reviewer iB7T's comment.**

*Reviewer 7Tfe*:
* Reviewer 7Tfe is worried that our results might be influenced by situations where the prior's length scale does not match well with the objective. This discrepancy is primarily caused by a certain unfortunate typo: *in the Bayesian regret experiments, the objective function is sampled from the prior with the same length scale*, ensuring perfect match between length scales in that setting. In contrast, our *synthetic benchmark and empirical experiments address the setting of learned length scales*. Taken together, our results demonstrate **good performance with both perfectly-matched and learned length scales**.
* In addition, Reviewer 7Tfe raises a number of interesting technical questions one could study in follow-up, such as calibrating our algorithms' parameters and accounting for noisy observations. Since we deliberately avoid using carefully-tuned hyperparameters for our method, **our results show performance improvement even if these parameters are not perfectly calibrated**. Handling noisy observations properly is an interesting question, but involves handling non-trivial technical details on the side of Gittins index theory: we discuss this in our response. For noise observations, therefore, we see the **opportunity to study interesting but non-obvious follow-up questions as strength rather than weaknesses**, since the creation of well-motivated technical questions is itself a significant source of research impact.

*Reviewer q648*:
* Reviewer q648, who rated our work the highest, asks about normalization and says this is the major reason preventing an even higher score. In short: we **already normalize the way Reviewer q648 wants us to** in all settings where this makes technical sense - namely, all except Bayesian regret. However, our submission **forgot to precisely document this in the appendix**, which we regret and have now fixed.
* Two figures requested by Reviewer q648 is available as PDF attached to this post.

## Additional contribution: Theorem 2

Our submission did not claim Proposition 2 - now called Theorem 2 - which relates the expected budget-constrained and cost-per-sample problems, as a contribution. We discovered Theorem 2 independently, then during literature review discovered the work of Aminian et al. (2024), which appeared before we were able to submit our work. This work contains a related result that we thought at time of submission might imply our Theorem 2. Since submission, we have determined that our paper's Theorem 2 is not implied by Theorem 1 of Aminian et al. (2024) due to differences in assumptions. We therefore now believe that **Theorem 2 with non-discrete-support reward distributions (for instance, Gaussian rewards) is an original contribution of our work**. Our updated manuscript now has a complete proof, which we will reproduce here if requested. We are also happy to comment on detailed technical differences with Aminian et al. (2024) on request.

---

### Decision · Program_Chairs · 2024-09-25

**Decision:**

Accept (poster)

**Comment:**

This manuscript takes a new approach to cost-aware Bayesian optimization by drawing a connection to the so-called Pandora's box problem from economics, a connection the authors exploit to derive a new acquisition function for this setting.

The reviewers were somewhat split in their assessment of this work during their initial reviews, although the rebuttal and author-reviewer discussion periods helped address some of the issues identified. There was a vigorous discussion regarding the work during the ensuing discussion period, where a champion emerged to support the paper in favor of acceptance on the strength of its theoretical and practical contributions.

Some reviewers continued to push for the authors to continue to clarify some issues in the manuscript brought up during the review process. The authors appear willing to address these issues, and I strongly encourage them to do so.